# Inductive Node-split Cross-Validation in Networks

Lin Zhang*

Cotality, Data Science and Platform

and

Bokai Yang

Qiuzhen College, Tsinghua University

and

Yuhong Yang†

Yau Mathematical Sciences Center, Tsinghua University

Beijing Institute of Mathematical Sciences and Applications

## Abstract

In network literature, cross-validation (CV) has been used for community detection via node or edge splitting, but with half-consistency that can only prevent underfitting. While penalized CV can rescue, it is of great interest to understand if traditional CV can achieve consistency and, if not, what structural obstacles fail to prevent overfitting. In this article, we propose a new CV framework, termed Inductive Node-split Cross-Validation (INCV). The key novelty is to combine node-splitting and an induction step, which results in three steps: training (labeling the nodes in the set of training nodes), membership induction for the held-out nodes, and model assessment based on the induced subgraph of the evaluation nodes. We show that our proposed INCV, without any additional penalty, can achieve full consistency for affiliation SBM. For general stochastic block models, we prove half-consistency and provide theoretical insight into the intrinsic difficulty of eliminating overfitting without additional structural assumptions. Numerical experiments and applications to two real network datasets demonstrate the robustness and practical effectiveness of the proposed method.

**Keywords:** cross-validation, consistency, stochastic block models, model selection
**Mathematics Subject Classification (2020):** 62F12, 62H30

## 1 Introduction

Studies of networks have attracted considerable attention in recent years across a wide range of disciplines, including statistics, computer science, social science, biology and psychology. A central topic in network analysis is the identification of community structure, as networks are often composed of heterogeneous subgroups, commonly referred to as communities. One example is a social network such as Facebook, where participants tend to cluster around several small cliques based on their age, profession, education, common interests, or other demographic or

---

*The first two authors contribute equally to this work.

†Corresponding author: yyangsc@mail.tsinghua.edu.cn

social factors. For the sake of simplicity, we assume in this article that the communities (i.e. cliques, caucuses, etc.) are mutually exclusive in the sense that a member can belong to one and only one community, as well as we assume that connections are undirected. A central problem in a network analysis is to determine the community structure, i.e., who belongs to which community. This task, known as community detection, plays a key role in many applications, including improving the recommendation system (Li and Chen, 2011), better segmenting or classifying images (Papadopoulos et al., 2010) or finding genetically related sub-populations (Greenbaum et al., 2016).

In statistics and machine learning research, one of the best-known probabilistic models for community detection is the stochastic block model (SBM), first introduced by Holland et al. (1983). SBM was developed for modeling a random graph with a community structure. Under SBM, the random graph of interest consists of nodes from distinct communities and has independent edges following Bernoulli distributions, where the entire model can therefore be fully characterized by an assignment function $\phi$ that maps each node to its belonging community and a probability matrix $P$ that gives the probabilities of edges within and between every communities.

There are three general types of community detection methods under SBM: greedy algorithms (Clauset et al., 2004), spectral clustering (Von Luxburg, 2007) and likelihood-based methods (Karrer and Newman, 2011; Bickel et al., 2013). While a large variety of community detection methods of these three types have been developed in the past decades, most of these methods rely on a crucial assumption that the number of communities in the network, denoted by $K$, is known. Thus, a fundamental question is how to determine the choice of $K$, since in many real applications there is no presumable $K$.

A substantial body of literature has been devoted to the problem of selecting the number of communities $K$ in stochastic block models, and a variety of methods have been proposed from both statistical and algorithmic perspectives, for example, by likelihood-based approaches (Wang and Bickel, 2017; Hu et al., 2020), spectral techniques (Le and Levina, 2022; Hwang et al., 2024; Wu et al., 2024), sequential testing procedures for SBMs or DCBM (Jin et al., 2023; Ma et al., 2021), or from a goodness-of-fit point of view (Jin et al., 2025). While these methods have achieved considerable success, they are typically constructed under specific modeling assumptions and may require carefully calibrated criteria or tuning parameters. This has motivated increasing interest in cross-validation as a more generic and data-driven tool for selecting the number of communities in network models.

Cross-validation is a widely used framework for model selection, with well-established theoretical foundations dating back to Stone (1974) and Geisser (1975), and further developed in a substantial body of subsequent work (see, e.g., Shao, 1993; Van Der Laan and Dudoit, 2003; Yang, 2007; Arlot and Celisse, 2010; Lei, 2025). To accommodate the dependence structure inherent in network data, cross-validation has been adapted through specialized designs, most notably node-splitting (Chen and Lei, 2018) and edge-sampling schemes (Li et al., 2020), and more recently subsampling-based approaches (Chakrabarty et al., 2025), primarily motivated by computational considerations. Related cross-validation ideas have also been developed for assessing network and graphon models; see, for example, Cheng et al. (2026). Despite these

methodological advances, existing results that rely solely on cross-validation typically establish only half-consistency for community number selection under stochastic block models, in the sense that underestimation of $K^*$ can be ruled out asymptotically, while overestimation remains difficult to prevent. To overcome this limitation, additional mechanisms beyond the conventional cross-validation have been introduced. Yang et al. (2025) incorporated an explicit penalization term to control overfitting and obtained full consistency, although the resulting guarantee no longer stems from cross-validation alone. Therefore, whether cross-validation alone can guarantee full consistency for network model selection, and under what structural conditions such guarantees may be attainable, remains an open question in general.

In this work, we propose an Inductive Node-split Cross Validation (INCV), a new node-splitting based cross-validation framework for selecting the number of communities in stochastic block models. Specifically, we randomly partition the node set into two disjoint subsets, denoted by $\mathcal{N}^{(1)}$ and $\mathcal{N}^{(2)}$. Unlike existing node-splitting CV method proposed in Chen and Lei (2018) which relies on spectral clustering on rectangular matrix to recover community structure, INCV first applies any off-the-shelf community detection method directly on the induced subgraph over $\mathcal{N}^{(1)}$ to obtain an initial labeling. The nodes in $\mathcal{N}^{(2)}$ are then classified based on their connections to $\mathcal{N}^{(1)}$, and model validation is subsequently carried out using edges within $\mathcal{N}^{(2)}$. In spirit, the classification step is similar to the predictive assignment strategy of Bhadra et al. (2026), which uses a subgraph-based community detection step to assign the remaining nodes in large networks. Our focus, however, is different: we study how such an induction step can be incorporated into cross-validation for selecting the unknown number of communities.

From a theoretical perspective, we will show that INCV achieves full consistency in a simplified affiliation SBM, relying solely on cross-validation without additional penalization. For general stochastic block models, we establish half-consistency and further provide insight into the intrinsic difficulty of ruling out overfitting in the absence of additional structural assumptions, as well as possible conditions under which full consistency may potentially be recovered. We also raise an ad-hoc extension of our framework to the more general DCSBM cases in Supplementary Material.

Furthermore, the proposed INCV method has several additional theoretical and numerical advantages compared with existing cross-validation approaches for community detection. Firstly, our proposed method requires clustering only on the subgraph induced from $\mathcal{N}^{(1)}$, avoiding repeated spectral decompositions such as preliminary low-rank approximation, such as in Li et al. (2020) and Chakrabarty et al. (2025). Secondly, since our INCV does not require low-rank approximations, it can in principle accommodate a wide range of community detection algorithms for the training set. Spectral clustering, likelihood-based methods, or other clustering procedures can be applied, whereas existing low-rank based CV methods are effectively restricted to spectral clustering or similar eigenvector-based algorithms. Thirdly, our conducted numerical experiments have shown the robustness of the proposed method under imbalance and various sparsity settings.

The rest of the article is organized as follows. Section 2 introduces the model specification and the proposed INCV procedure. Section 3 contains the main theoretical results for this article, where Section 3.1 establishes full consistency under affiliation SBMs and Section 3.2

takes general SBMs into account. Section 4 presents various simulations to examine practical performances of our proposed method and compare them with existing CV methods. Two real data examples are given in Section 5. Conclusions and possible future directions are discussed in Section 6.

## 2 Inductive Node-split Cross-Validation (INCV) Procedure

### 2.1 Community Detection under the Stochastic Block Model

The Stochastic block model (SBM) (Holland et al., 1983) is one of the simplest but also most useful models to characterize a random network where the probability of a connection between any two members only depends on which communities they belong to.

Let $G = (\mathcal{N}, \mathcal{E})$ be a random graph where $\mathcal{N}$ is the set of $n$ nodes, representing the $n$ members of the network and $\mathcal{E}$ is the set of random edges in the graph. Without loss of generality, we write the node set as $\mathcal{N} = \{1, 2, \ldots, n\} = [n]$, and the edge set as $\mathcal{E} = \{(i, j) : 1 \leq i < j \leq n\}$. We assume the network contains $K$ disjoint communities, indexed by $\{1, 2, \ldots, K\}$, where $2 \leq K \leq K_{\max}$ for some positive integer $K_{\max}$. Let $\phi : \mathcal{N} \to \{1, 2, \ldots, K\}$ denote the community labeling function, which assigns each node to the community it belongs to; that is, $\phi(i) \in \{1, 2, \ldots, K\}$ for all $i \in \mathcal{N}$. The graph $\mathcal{E}$ can be represented by an $n \times n$ symmetric matrix $A = (A_{ij})_{i,j=1}^{n}$, known as the adjacency matrix. Its upper off-diagonal entries $A_{ij}$ ($1 \leq i < j \leq n$) take value one if there is an edge between node $i$ and node $j$, and zero otherwise. Since the graph is undirected, $A_{ij} = A_{ji}$. Self-loops are not allowed in the stochastic block model, and therefore the diagonal entries of $A$ satisfy $A_{ii} = 0$ for all $i \in \mathcal{N}$.

Under the stochastic block model, the off-diagonal entries of $A$ are independent Bernoulli random variables whose success probabilities depend only on the communities to which the two nodes belong. These probabilities are collected in a $K \times K$ symmetric matrix $P = (p_{uv})_{u,v=1}^{K}$, called the connectivity matrix. The entry $p_{uv}$ represents the probability of an edge between a node in community $u$ and a node in community $v$. In particular, the diagonal entry $p_{uu}$ corresponds to the within-community connection probability, while the off-diagonal entry $p_{uv}$ ($u \neq v$) corresponds to the between-community connection probability. Since the graph is undirected, $p_{uv} = p_{vu}$. Using the labeling function $\phi$, we may write

$$\Pr(A_{ij} = 1) = 1 - \Pr(A_{ij} = 0) = p_{\phi(i)\phi(j)}, \quad \text{for } i \neq j.$$

Given an observed adjacency matrix $A$ and a known number of communities $K$, the goal of community detection is to estimate the community labeling function $\phi$ as well as the connectivity matrix $P$. A natural approach is the maximum likelihood estimation under the stochastic block model, which solves

$$\max_{\phi, P} \sum_{1 \leq i < j \leq n} \left\{ A_{ij} \log p_{\phi(i)\phi(j)} + (1 - A_{ij}) \log(1 - p_{\phi(i)\phi(j)}) \right\}. \tag{1}$$

However, it is known that obtaining an exact solution to (1) is NP-hard (Chen et al., 2014). In practice, the optimization problem (1) is typically addressed using heuristic or approximate

methods, including the label switching algorithm (Bickel and Chen, 2009) and tabu search (Zhao et al., 2012).

Besides likelihood-based methods, spectral clustering provides a computationally efficient and widely used approach for community detection. Spectral clustering can be implemented based on either the graph Laplacian or the adjacency matrix. For example, the (unnormalized) graph Laplacian is defined as

$$L = D - A,$$

where $D$ is the degree matrix with diagonal entries

$$D_{ii} = \sum_{j=1}^{n} A_{ij}.$$

When the number of communities is $K$, spectral clustering proceeds by computing the eigenvectors corresponding to the $K$ smallest eigenvalues of $L$, and then applying the $K$-means algorithm to the rows of the resulting eigenvector matrix.

Alternatively, normalized graph Laplacians such as

$$L_{\text{rw}} = D^{-1}L \quad \text{or} \quad L_{\text{sym}} = D^{-1/2}LD^{-1/2}$$

may be used, in which case row normalization is applied before $K$-means. Another commonly used approach is to perform singular value decomposition on the adjacency matrix $A$, and apply $K$-means to the leading $K$ singular vectors.

It has been established that, under suitable conditions on the connectivity matrix $P$, community sizes, and the number of communities $K$, spectral clustering consistently recovers community structure under the stochastic block model; see Von Luxburg et al. (2008), Rohe et al. (2011), and Lei and Rinaldo (2015).

## 2.2  The INCV Procedure

In general, the cross validation procedure will take three steps to determine the community number $K$ for community detection. Step 1 is to randomly partition the network into a training set and a test set. Step 2 is to perform community detection with considered $K$ values on the training set; Step 3 is to evaluate these candidate $K$ values using the log likelihood or the mean squared error (MSE) loss calculated on the test set. Step 1 to Step 3 may be repeated several times through independent samplings or by the use of an $F$-fold cross validation strategy. The $K$ value that achieves the highest log likelihood or the lowest MSE loss will be the chosen community number for the network.

Specifically, under our regime, INCV begins with a random partition of the node set $\mathcal{N}$ into a training set $\mathcal{N}^{(1)}$ and a test set $\mathcal{N}^{(2)}$. We denote the splitting ratio by

$$\tau = \frac{|\mathcal{N}^{(1)}|}{|\mathcal{N}|} \in (0, 1).$$

Under a suitable permutation of rows and columns, the adjacency matrix $A$ can be partitioned

into four blocks as

$$A = \begin{pmatrix} A^{(11)} & A^{(12)} \\ A^{(21)} & A^{(22)} \end{pmatrix},$$

where the two matrix blocks $A^{(11)}$ and $A^{(22)}$ represent the subgraphs induced by the training set $\mathcal{N}^{(1)}$ and the test set $\mathcal{N}^{(2)}$, respectively. The off-diagonal block $A^{(12)} = \left(A^{(21)}\right)^{\top}$ is a $|\mathcal{N}^{(1)}| \times |\mathcal{N}^{(2)}|$ matrix block where the connections between the training and test sets are stored.

Figure 1: Schematic illustration of the INCV procedure. After splitting the node set into a training subset $\mathcal{N}^{(1)}$ and a test subset $\mathcal{N}^{(2)}$, the adjacency matrix is partitioned into four blocks. The block $A^{(11)}$ is used for clustering, $A^{(12)}$ for classification, and $A^{(22)}$ for validation.

Figure 1 illustrates the conceptual core of INCV. Under the stochastic block model, the three matrix blocks $A^{(11)}$, $A^{(12)}$, and $A^{(22)}$ are mutually independent and play distinct roles in cross-validation:

- **Clustering**: $A^{(11)}$ is used to cluster the training nodes $\mathcal{N}^{(1)}$ into $K$ communities;

- **Classification**: $A^{(12)}$ is used to assign the test nodes $\mathcal{N}^{(2)}$ to the $K$ communities learned from $\mathcal{N}^{(1)}$;

- **Validation**: $A^{(22)}$ is used to evaluate the fitted communities on the test nodes.

We explain the above three steps in detail as follows.

**Clustering.** In the clustering step, for a given candidate number of communities $K \in \{2, \ldots, K_{\max}\}$, we apply a standard community detection method to the subgraph $A^{(11)}$ where either likelihood-based methods or spectral clustering are available, then obtain a partition of $\mathcal{N}^{(1)}$ into $K$ disjoint communities, denoted by $\hat{\mathcal{N}}_1^{(1)}, \hat{\mathcal{N}}_2^{(1)}, \ldots, \hat{\mathcal{N}}_K^{(1)}$. This yields a community labeling function $\hat{\phi}_K : \mathcal{N}^{(1)} \to \{1, 2, \ldots, K\}$, where $\hat{\phi}_K(i) = k$ if node $i \in \hat{\mathcal{N}}_k^{(1)}$.

**Classification.** In the classification step, we assign each node in the test set $\mathcal{N}^{(2)}$ to one of the $K$ communities obtained from the clustering step. Specifically, we adopt a one-vs-all

classification rule based on the block $A^{(12)}$. For each node $j \in \mathcal{N}^{(2)}$, its estimated community label is given by

$$\hat{\phi}_K(j) = \arg\max_{1 \leq k \leq K} \left\{ \frac{1}{|\hat{\mathcal{N}}_k^{(1)}|} \sum_{\substack{i \in \mathcal{N}^{(1)} \\ \hat{\phi}_K(i) = k}} A_{ij}^{(12)} \right\}, \tag{2}$$

where $\hat{\mathcal{N}}_k^{(1)}$ denotes the set of training nodes assigned to community $k$ in the clustering step.

Heuristically, the one-vs-all classifier assigns a test node to the community with which it exhibits the strongest affinity, measured by the average number of connections in $A^{(12)}$. After this step, we obtain the estimated labeling function $\hat{\phi}_K$ for all nodes in $\mathcal{N}$.

**Validation.** Given the estimated labeling function $\hat{\phi}_K$, we evaluate its performance on the held-out subgraph $A^{(22)}$. For each pair of community indices $u, v \in \{1, 2, \ldots, K\}$, define

$$\mathcal{E}_{uv} = \{(i, j) : i, j \in \mathcal{N}^{(2)}, \ i < j, \ \hat{\phi}_K(i) = u, \ \hat{\phi}_K(j) = v\},$$

the set of node pairs in the test set whose endpoints are assigned to communities $u$ and $v$, respectively.

Conditioning on $\hat{\phi}_K$, the negative log-likelihood of the connectivity parameters $\{p_{uv}\}_{u,v=1}^K$, evaluated on $A^{(22)}$, is given by

$$\ell(\{p_{uv}\}, \hat{\phi}_K) = -\sum_{u,v=1}^K \sum_{(i,j) \in \mathcal{E}_{uv}} \left\{ A_{ij}^{(22)} \log p_{uv} + (1 - A_{ij}^{(22)}) \log(1 - p_{uv}) \right\}. \tag{3}$$

The maximum likelihood estimator of $p_{uv}$ based on the test data is

$$\hat{p}_{uv} = \frac{1}{|\mathcal{E}_{uv}|} \sum_{(i,j) \in \mathcal{E}_{uv}} A_{ij}^{(22)}. \tag{4}$$

Substituting $\hat{p}_{uv}$ into (3) yields the minimized negative log-likelihood

$$\ell(\{\hat{p}_{uv}\}, \hat{\phi}_K) = -\sum_{u,v=1}^K \sum_{(i,j) \in \mathcal{E}_{uv}} \left\{ A_{ij}^{(22)} \log \hat{p}_{uv} + (1 - A_{ij}^{(22)}) \log(1 - \hat{p}_{uv}) \right\}. \tag{5}$$

For a single node-splitting cross-validation run, the selected number of communities is defined as

$$K_{\text{opt}} = \arg\min_{2 \leq K \leq K_{\max}} \ell(\{\hat{p}_{uv}\}, \hat{\phi}_K). \tag{6}$$

**Alternative Validation: Mean Squared Error.** Instead of the log-likelihood in (3), one may also evaluate the estimated community labels $\hat{\phi}_K$ using the mean squared error between the observed and predicted adjacency entries in $A^{(22)}$, inspired by existing literature such as Chen and Lei (2018) and Li et al. (2020):

$$\text{MSE}(\{\hat{p}_{uv}\} \mid \hat{\phi}_K) = \frac{1}{|\mathcal{E}^{(2)}|} \sum_{u,v=1}^K \sum_{(i,j) \in \mathcal{E}_{uv}} \left( A_{ij}^{(22)} - \hat{p}_{uv} \right)^2, \tag{7}$$

where $\mathcal{E}^{(2)}$ is the subgraph induced by test node set $\mathcal{N}^{(2)}$. The optimal number of communities can then be selected by minimizing $\text{MSE}(\hat{\phi}_K)$ over candidate values of $K$. This MSE-based validation is more general, as it does not rely on the Bernoulli likelihood assumption and may yield consistency under weaker conditions.

**Alternative Probability Estimation.** In the likelihood and MSE criteria above, the block-wise connection probabilities $\{\hat{p}_{uv}\}$ are estimated from the validation block $A^{(22)}$. This choice leads to a natural MLE-based validation rule and is the main version analyzed in our theoretical results.

As a practical alternative, one may estimate the block-wise connection probabilities from the training block $A^{(11)}$ and then plug these estimates into the validation loss on $A^{(22)}$. This is in the same spirit as existing network cross-validation methods such as Chen and Lei (2018) and Li et al. (2020), where the model parameters are estimated from the training part and evaluated on the held-out part.

Specifically, for $1 \le u, v \le K$, define

$$\mathcal{E}_{uv}^{(1)} = \{(i,j) : i, j \in \mathcal{N}^{(1)}, \, i < j, \, \hat{\phi}_K(i) = u, \, \hat{\phi}_K(j) = v\},$$

and let

$$\hat{p}_{uv}^{(11)} = \frac{1}{|\mathcal{E}_{uv}^{(1)}|} \sum_{(i,j) \in \mathcal{E}_{uv}^{(1)}} A_{ij}^{(11)}. \tag{8}$$

The plug-in validation criteria are then obtained by replacing $\hat{p}_{uv}$ in (5) and (7) with $\hat{p}_{uv}^{(11)}$.

This training-based plug-in version uses $A^{(22)}$ only for evaluation rather than for both parameter estimation and evaluation. Hence, in unrestricted general SBM settings where the number of block-wise parameters increases with $K$, it may be less prone to overfitting in finite samples. A formal consistency analysis of this plug-in version would require combining concentration of the training-block probability estimators with concentration of the held-out validation loss, following arguments similar to those in existing network cross-validation analyses. Since the main theoretical focus of this paper is the MLE-based validation rule, we leave a full treatment of the plug-in version outside the present scope and include it as a practical alternative in the supplementary numerical experiments.

In practice, node-splitting cross-validation is implemented using an $F$-fold strategy. The node set $\mathcal{N}$ is randomly partitioned into $F$ subsets of nearly equal sizes. At each fold, one subset is treated as the test set, while the remaining $F - 1$ subsets form the training set. The above clustering, classification, and validation steps are repeated across all folds, and the final selected community number is the value of $K$ that minimized the validation negative log-likelihood $\ell(\{\hat{p}_{uv}\}, \hat{\phi}_K)$ (or MSE) over the $F$ folds. The full algorithm is displayed in Algorithm 1 below.

Note that Algorithm 1 differs from the Blockwise Node-Pair Splitting Network Cross Validation (NCV) proposed by Chen and Lei (2018) in two crucial aspects, although both of them are based on splitting nodes. First, Chen and Lei's method clusters all nodes into $K$ communities in a single step by performing a singular value decomposition on the rectangular matrix block $(A^{(11)}, A^{(12)})$, whereas our INCV method recovers the community structure through two separate

---

**Algorithm 1:** Inductive Node-split Cross-Validation (INCV)

---

**Input:** Adjacency matrix $A$; maximum number of communities $K_{\max} \geq 2$; number of folds $F \geq 2$; a community detection method; validation type $\texttt{type} \in \{\texttt{likelihood}, \texttt{MSE}\}$; probability estimation option $\texttt{prob.est} \in \{\texttt{validation-MLE}, \texttt{training-plug-in}\}$.

**Output:** Optimal community number $K_{\text{INCV}}$.

**1:** Randomly partition the node set $\mathcal{N} = \{1, 2, \ldots, n\}$ into $F$ nearly equal-sized subsets, denoted as $\mathcal{N}^{(1)}, \mathcal{N}^{(2)}, \ldots, \mathcal{N}^{(F)}$.

**2: for** $f = 1$ **to** $F$ **do**

**3:**     Partition the adjacency matrix $A$ into three matrix blocks:

$$A^{(11)} = (A_{ij})_{i,j \in \mathcal{N} \setminus \mathcal{N}^{(f)}},$$
$$A^{(12)} = (A_{ij})_{i \in \mathcal{N} \setminus \mathcal{N}^{(f)}, j \in \mathcal{N}^{(f)}},$$
$$A^{(22)} = (A_{ij})_{i,j \in \mathcal{N}^{(f)}}.$$

**4:**     **for** $K = 2$ **to** $K_{\max}$ **do**

**5:**         Run the input community detection algorithm on $A^{(11)}$ to cluster $\mathcal{N} \setminus \mathcal{N}^{(f)}$ into $K$ communities.

**6:**         Classify nodes in $\mathcal{N}^{(f)}$ into these $K$ communities using the one-vs-all rule (2).

**7:**         **if** *prob.est* = *validation-MLE* **then**

**8:**             Estimate the block probabilities $\{\hat{p}_{uv}\}$ from $A^{(22)}$ using (4).

**9:**         **else**

**10:**             Estimate the block probabilities $\{\hat{p}_{uv}\}$ from $A^{(11)}$ using (8).

**11:**         **if** *type* = *likelihood* **then**

**12:**             Compute the negative log-likelihood loss $L_K^{(f)} = \ell(\{\hat{p}_{uv}\}, \hat{\phi}_K)$ using (5) on $A^{(22)}$.

**13:**         **else**

**14:**             Compute the mean squared error $L_K^{(f)} = \text{MSE}(\{\hat{p}_{uv}\} \mid \hat{\phi}_K)$ using (7) on $A^{(22)}$.

**15: return** $K_{\text{INCV}} = \arg\min_{2 \leq K \leq K_{\max}} \frac{1}{F} \sum_{f=1}^{F} L_K^{(f)}$.

---

steps: clustering on $A^{(11)}$ and classification on $A^{(12)}$. Second, Chen and Lei's method estimates the connectivity probabilities $\{p_{uv}\}_{u,v=1}^K$ using the same matrix entries in $(A^{(11)}, A^{(12)})$ that are also used for clustering. In contrast, our INCV method estimates the connectivity probabilities via the maximum likelihood estimator or MSE based on $A^{(22)}$, whose entries can be regarded as independent conditional on the estimated community labeling function $\hat{\phi}_K$ obtained from $A^{(11)}$ and $A^{(12)}$. This conditional independence property plays a crucial role in establishing the theoretical consistency of the proposed INCV procedure.

The classification step in INCV is related to the predictive assignment framework of Bhadra et al. (2026), but the goals and the considered induction rules are different. Predictive assignment aims at scalable community detection for a given $K$; it constructs model-based prototypes or structural-link quantities from the training subgraph and assigns the remaining nodes by comparing them with these estimated targets. In contrast, INCV is designed for cross-validation-based selection of an unknown $K$, uses a direct induction rule, such as the affinity-based rule in the main text or the likelihood-based rule considered in the supplement, and reserves the induced subgraph $A^{(22)}$ for validation. Thus, the novelty of INCV is not the induction idea alone, but its use within a node-splitting cross-validation framework for selecting an unknown number of communities.

## 3  Consistency of INCV

In this section, we establish the consistency of the proposed INCV procedure under a set of mild assumptions on the network model and the community detection algorithm, in both affiliation setting and general SBM setting. Consider an undirected network $(\mathcal{N}, \mathcal{E})$ generated from a stochastic block model with $K^*$ disjoint communities of sizes $\{n_k\}_{k=1}^{K^*}$ and a true community labeling function $\phi^*$. Throughout this section, we assume that the true number of communities $K^*$ is fixed and satisfies $2 \le K^* \le K_{\max}$ for some known positive integer $K_{\max}$. Let $n = |\mathcal{N}| = \sum_{k=1}^{K^*} n_k$ denote the total number of nodes in the network.

### 3.1  Affiliation Block Models

We first consider a simple case that the true block model is an assortative affiliation block model (Frank and Harary, 1982; Allman et al., 2011), which is a special case of SBM. Specifically speaking, here we assume that the within-community connection probabilities are all equal to $p$ and the between-community connection probabilities are all equal to $q$ where $0 < q < p < 1$.

We run the Inductive Node-split Cross Validation for the network with one random partition of the node set $\mathcal{N}$ into the training set $\mathcal{N}^{(1)}$ and the test set $\mathcal{N}^{(2)}$ according to a splitting ratio $\tau = |\mathcal{N}^{(1)}|/|\mathcal{N}| \in (0,1)$. Different from the previous algorithm designed for a general SBM, the loss function is adapted to this special affiliation structure, leading to a slightly modified computation.

To proceed, given a candidate $K$, denote the estimated label obtained from the classification step as $\hat{\phi}_K$. In the validation step of INCV, under the assumption that the within-community connection probability is equal to $p$ and the between-community connection probability is equal

to $q$, we define
$$W_{\hat{\phi}_K} := \{(i,j) : i,j \in \mathcal{N}, \ i < j, \ \hat{\phi}_K(i) = \hat{\phi}_K(j)\},$$
and
$$B_{\hat{\phi}_K} := \{(i,j) : i,j \in \mathcal{N}, \ i < j, \ \hat{\phi}_K(i) \neq \hat{\phi}_K(j)\}.$$

The sets $W_{\hat{\phi}_K}$ and $B_{\hat{\phi}_K}$ partition the upper off-diagonal entries of the adjacency matrix $A$ into two disjoint groups: the within-community pairs and the between-community pairs.

Restricting to the entries in the validation block $A^{(22)}$, we further define
$$W_{\hat{\phi}_K}^{(2)} := \{(i,j) : i,j \in \mathcal{N}^{(2)}, \ i < j, \ \hat{\phi}_K(i) = \hat{\phi}_K(j)\},$$
and
$$B_{\hat{\phi}_K}^{(2)} := \{(i,j) : i,j \in \mathcal{N}^{(2)}, \ i < j, \ \hat{\phi}_K(i) \neq \hat{\phi}_K(j)\}.$$

These sets are subsets of $W_{\hat{\phi}_K}$ and $B_{\hat{\phi}_K}$ corresponding to the entries of $A^{(22)}$ only.

Under the affiliation structure, the minimized negative log-likelihood function defined in (5) can be simplified as

$$\ell(\hat{p}, \hat{q} \mid \hat{\phi}_K) = - \sum_{(i,j) \in W_{\hat{\phi}_K}^{(2)}} \left[ A_{ij}^{(22)} \log \hat{p} + (1 - A_{ij}^{(22)}) \log(1 - \hat{p}) \right]$$
$$- \sum_{(i,j) \in B_{\hat{\phi}_K}^{(2)}} \left[ A_{ij}^{(22)} \log \hat{q} + (1 - A_{ij}^{(22)}) \log(1 - \hat{q}) \right],$$

where the corresponding maximum likelihood estimators are given by

$$\hat{p} = \frac{\sum_{(i,j) \in W_{\hat{\phi}_K}^{(2)}} A_{ij}^{(22)}}{|W_{\hat{\phi}_K}^{(2)}|}, \qquad \hat{q} = \frac{\sum_{(i,j) \in B_{\hat{\phi}_K}^{(2)}} A_{ij}^{(22)}}{|B_{\hat{\phi}_K}^{(2)}|}. \tag{9}$$

The optimal community number $K$ for this single-split INCV is then selected according to (6). If the MSE is preferred, then one can plug in the estimated label $\hat{\phi}_K$ and the expressions of $\hat{p}$ and $\hat{q}$ obtained in (9) into (7), to correspondingly change the computation.

Now we introduce several definitions and assumptions required to establish the consistency result.

**Definition 1** (Balanced Community Structure). A network $(\mathcal{N}, \mathcal{E})$ with size $n$ and $K \geq 2$ disjoint communities is claimed to have a balanced community structure, if its minimum community size
$$n_{\min} = \min_{1 \leq k \leq K}(n_k) \geq \frac{n\pi_0}{K}$$
for some positive constant $\pi_0 \in (0,1)$ independent of $n$.

*Assumption* 1 (Balanced Assumption). The true community structure (under $\phi^*$) of a network and the recovered community structure (under $\hat{\phi}_K$) from INCV for $K \in \{2, \ldots, K_{\max}\}$ are balanced with a constant $\pi_0 \in (0,1)$.

*Remark* 1. The first part of Assumption 1 that $\phi^*$ is balanced is widely adopted in existing

network cross-validation literature, such as Chen and Lei (2018), Li et al. (2020) and Chakrabarty et al. (2025). For the second part that $\hat{\phi}_K$ is balanced, a similar assumption is made in Yang et al. (2025). We shall view it as a technical regularization on the estimated labels, rather than as an assumption on the intrinsic clustering accuracy under possible model misspecification. In detail, take the spectral clustering with exact $k$-means as an example. When $K \leq K^*$, then by the Nonsplitting Property (NSP) proposed in Jin et al. (2023), we know on $\mathcal{N}^{(1)}$ the assumption holds. When $K > K^*$, on $\mathcal{N}^{(1)}$, this condition can be enforced through a simple post-processing step, such as merging or trimming clusters whose sizes fall below a prespecified threshold. On $\mathcal{N}^{(2)}$, denote $k_{\min}$ and $k_{\max}$ as the minimum and maximum communities for estimated label $\hat{\phi}_K$ on $\mathcal{N}^{(2)}$, and $\mathcal{N}_{\min}$ and $\mathcal{N}_{\max}$ as corresponding node sets. If the balanced condition is not satisfied, then we shall re-classify the node $j_0$ from $\mathcal{N}_{\max}$ to $\mathcal{N}_{\min}$ that satisfies

$$
j_0 = \arg \max_{j \in \mathcal{N}_{\max}^{(2)}} \left( \sum_{\substack{i \in \mathcal{N}^{(1)} \\ \hat{\phi}_K(i) = k_{\min}}} A_{ij}^{(12)} \right).
$$

This reclassification step can be repeated until the balanced condition holds.

The role of this condition is mainly theoretical. In the affiliation SBM analysis, when $K > K^*$, the proof needs to rule out degenerate overfitted partitions in which almost all test nodes are assigned to a small number of estimated communities. The balanced-label condition provides a sensible way to guarantee that within- and between-type validation edge sets have sufficiently large cardinalities, which is essential for controlling the validation loss under overfitting.

We emphasize that this post-processing step is not part of the default empirical implementation of INCV. In all numerical experiments and real data analyses reported in this paper, we use the natural outputs of the clustering and classification steps without forced reclassification. Thus, Assumption 1 should be understood as a theoretical device for excluding degenerate candidate partitions. To assess its practical relevance, we report in the supplementary material the empirical balance ratio of the estimated labels for overfitted candidate values $K > K^*$, and the results show that severe imbalance is not a serious issue in our simulations.

*Assumption* 2. The within and between community probability $p$ and $q$ satisfy $0 < \delta_n < q < p < 1 - \delta_n < 1$ for some non-increasing sequence $\{\delta_n\}$.

Assumption 2 ensures that the between-community connections are not too sparse $0 < \delta_n < q$, where the approved sparsity level is controlled by the sequence $\{\delta_n\}$. Also, the within-community connections are not too dense either ($p < 1 - \delta_n < 1$).

*Assumption* 3 (Clustering performance). When the candidate community number $K = K^*$, the clustering method used in the Clustering step of INCV gives a consistent estimate $\phi_{K^*}$ from $A^{(11)}$ so that on $\mathcal{N}^{(1)}$

$$
\Pr\left(\hat{\phi}_{K^*} \neq \phi^*\right) \leq g(n\tau) \tag{10}
$$

up to label permutation, where $n$ is the network size, $\tau$ is the node splitting ratio for the training subset, and $g(x)$ is a positive function such that $g(x) \to 0$ as $x \to \infty$.

Assumption 3 concerns the quality of the community detection method used in the training step of INCV, which is separate from the validation requirements in the consistency theorem.

For theoretical clarity, Assumption 3 is formulated as an exact-recovery condition. In the numerical experiments, we implement spectral clustering with a practical multi-start approximate $k$-means algorithm rather than exact $k$-means, since globally solving the $k$-means problem is computationally intractable in general. Such a distinction between the idealized clustering step in theory and the practical implementation in computation is commonly adopted in spectral clustering based network analysis; see, for example, Jin et al. (2023). Concrete sufficient conditions under which Assumption 3 holds, and their compatibility with Theorem 1 and Corollary 1, are discussed after Corollary 1.

Now we can state our main theorem for affiliation SBMs.

**Theorem 1** (Consistency of INCV for affiliation SBMs). *Consider an undirected network $(N, E)$ generated from an affiliation stochastic block model with a true community number $K^*$ satisfying $2 \leq K^* \leq K_{\max}$ for some positive integer $K_{\max}$. The parameters $p$ and $q$ are the within and between community connection probabilities. The INCV has a splitting ratio $\tau = |N^{(1)}|/|N| \in (0,1)$. Under Assumptions 1–3, if the following requirements are met under likelihood objective:*

*(i) $\tau = \omega\left(1/n\right), \tau(p-q)^2 = \omega\left(\log n/n\right),$*

*(ii) $(1-\tau)\delta_n = \omega\left(1/n\right), (1-\tau)(p-q)^2 = \omega\left(\log n \left|\log(\delta_n/2)\right|/n\right),$*

*or the above second condition modified to*

*(ii') $1 - \tau = \omega\left(1/n\right), (1-\tau)(p-q)^2 = \omega\left(\log n/n\right),$*

*for the MSE objective, then the optimal community number $\hat{K}_{\mathrm{INCV}}$ obtained from the INCV procedure satisfies*

$$\lim_{n\to\infty} \Pr\big(\hat{K}_{\mathrm{INCV}} = K^*\big) = 1.$$

*Remark* 2. Condition (i) speaks for the cross validation splitting ratio $\tau$. Intuitively, $\tau$ cannot be too small, because otherwise the training set would be too small to recover the community structure. On the other hand, from condition (ii) or (ii'), the test ratio $1 - \tau$ cannot be small either, because the test subset needs to be large enough to estimate $p$ and $q$ and to evaluate the objective function. Also notice that the second part of Condition (i) and (ii) or (ii') states dependency of $\tau$, $1 - \tau$ on $p - q$ and $\delta_n$. When the discrepancy $p - q$ of the network is large enough to distinguish different communities, the splitting ratio $\tau$ could be close to either 0 or 1. However, when the discrepancy $p - q$ of the network is not large enough to distinguish different communities, $\tau$ may be required as a constant. It is obvious that the two conditions are met in a simple case where $\tau$, $p$, and $q$ are constant (and thus, $\delta_n$ is a constant sequence).

Beyond the affiliation models, the existing works on network cross-validation, such as Chen and Lei (2018), Li et al. (2020) and Chakrabarty et al. (2025), focus on general stochastic block models and establish consistency results primarily in terms of avoiding underfitting. Yang et al. (2025) adopted a penalization point of view, in order to address the overfitting problem. They do not consider the affiliation SBM as a special case, nor do they investigate whether cross-validation alone can achieve full consistency under such a simplified structure. Although here the block probability matrix admits a highly structured form, even under this simplified setting, preventing overfitting via cross-validation is not clear as reflected in the previous results with

both node- or edge-splitting CV methods. Theorem 1 shows that, under suitable conditions on the splitting ratio and signal strength, cross-validation by itself can be sufficient to guarantee full consistency.

The key reason why INCV can rule out overfitting in the affiliation SBM is that the validation step, together with the two-parameter affiliation structure, can detect the structural distortion introduced by splitting a true community. When $K > K^*$, an overfitted labeling must split at least one true community into two or more estimated communities. Under the affiliation structure, such a split changes the role of many validation edges: edges that are truly within-community may be treated as between-community edges by the estimated partition. Since all within-community probabilities are pooled into a common parameter $p$ and all between-community probabilities are pooled into a common parameter $q$, this misallocation creates a systematic difference in the validation loss. In contrast, under a fully unrestricted SBM, the newly created block pairs can each be assigned their own probability parameters, and the same split may not generate a leading-order loss gap.

Moreover, our setting extends the traditional sparse network formulation in which the block-wise connection probability matrix is assumed to take the form $B_n = \rho_n B_0$, where $\rho_n \to 0$ controls the overall sparsity level and $B_0$ is a fixed matrix. In contrast, our analysis allows the within and between community probabilities $p$ and $q$ to vary more flexibly with $n$, subject only to mild lower and upper bounds. As a result, the classical sparse stochastic block model with multiplicative scaling is naturally covered as a special case of our framework, with corresponding results presented in the following corollary.

**Corollary 1.** *Consider an undirected network $(\mathcal{N}, \mathcal{E})$ generated from an affiliation stochastic block model with $p$ and $q$ being the within and between community connection probabilities. If $p$ and $q$ satisfy $p = \rho_n p_0$ and $q = \rho_n q_0$ for $p_0 > q_0$ fixed, then under Assumptions 1 and 3, if the following requirements are met under likelihood objective:*

*(i)  $\tau = \omega\left(1/n\right), \tau\rho_n = \omega\left(\log n/n\right),$*

*(ii)  $1 - \tau = \omega\left(1/n\right), (1-\tau)\rho_n^{3/2} = \omega\left(\sqrt{\log n}/n\right),$*

*or the above second condition modified to*

*(ii′)  $1 - \tau = \omega\left(1/n\right), (1-\tau)\sqrt{\rho_n} = \omega\left(\sqrt{\log n}/n\right),$*

*for the MSE objective, then the optimal community number $\hat{K}_{\mathrm{INCV}}$ obtained from the INCV procedure satisfies*

$$\lim_{n \to \infty} \Pr\bigl(\hat{K}_{\mathrm{INCV}} = K^*\bigr) = 1.$$

*Remark* 3. Here, this specific regime allows us to use Bernstein's inequality to attain stricter tail probability bounds, in order to relax the condition on $\rho_n$. For the likelihood objective, when the splitting ratio $\tau$ is of constant order, conditions (i) and (ii) together imply that $\rho_n = \omega(n^{-2/3}\log^{1/3} n)$. This requirement is weaker than the corresponding condition imposed in Li et al. (2020) for the binomial deviance loss. A methodological explanation is that the proposed INCV framework does not rely on low-rank approximation of the adjacency matrix. As a result, it permits the use of clustering methods that achieve exact recovery under suitable

signal-to-noise conditions. By contrast, existing edge-splitting based cross-validation procedures typically require reconstructing a low-rank surrogate of the probability matrix in order to stabilize estimation. This reconstruction step effectively restricts the admissible clustering methods, with spectral-type algorithms being the primary viable choice, and precludes the direct use of exact recovery procedures. For the MSE objective, the conditions reduce to $\rho_n = \omega(\log n/n)$, which aligns with the optimal rates established in the existing literature.

*Remark* 4. We now clarify how Assumption 3 can be verified and how it combines with the requirements in Theorem 1 and Corollary 1. Let

$$m = |\mathcal{N}^{(1)}| \asymp n\tau, \qquad \Delta_{n,\tau} = \frac{m(p-q)^2}{p}.$$

For spectral clustering with exact $k$-means, let $U$ be the matrix of the leading $K^*$ eigenvectors of the population probability matrix on the training subgraph, and let $\widehat{U}$ be the corresponding eigenvector matrix of $A^{(11)}$. Using the two-to-infinity norm perturbation decomposition in Theorem 3.1 of Cape et al. (2019), one obtains, under a standard logarithmic expected-degree condition and sufficient signal strength, a row-wise eigenvector perturbation bound of the form

$$\|\widehat{U} - UO\|_{2,\infty} \le C \max\left\{ \frac{1}{\Delta_{n,\tau}}, \sqrt{\frac{\log n}{\Delta_{n,\tau}}} \right\}$$

with high probability, for some orthogonal matrix $O$. Combining this row-wise perturbation control with the NSP argument of Jin et al. (2023), we obtain that spectral clustering with exact $k$-means satisfies Assumption 3 whenever

$$n\tau p \gg \log n, \qquad \Delta_{n,\tau} \gg \log n.$$

In particular, since $p \le 1$, condition (i) in Theorem 1, $\tau(p-q)^2 = \omega\left(\log n/n\right)$ implies both $n\tau p \gg \log n$ and $\Delta_{n,\tau} = n\tau(p-q)^2/p \gg \log n$. Thus the training-stage exact-recovery requirement for this idealized spectral clustering step is compatible with the signal-strength condition in Theorem 1.

Under the sparse multiplicative scaling in Corollary 1, where $p = \rho_n p_0$ and $q = \rho_n q_0$ with fixed $p_0 > q_0 > 0$, the above spectral clustering requirements reduce to the same order:

$$n\tau p \asymp n\tau\rho_n, \qquad \Delta_{n,\tau} = \frac{n\tau(p-q)^2}{p} \asymp n\tau\rho_n.$$

Hence both $n\tau p \gg \log n$ and $\Delta_{n,\tau} \gg \log n$ are implied by $\tau\rho_n = \omega\left(\log n/n\right)$, which matches condition (i) in Corollary 1. This same condition is also consistent with likelihood-based training methods. Indeed, the likelihood modularity of Bickel and Chen (2009) is a profile-likelihood criterion, and its exact-consistency result requires the expected degree parameter $\lambda_n$ to satisfy $\lambda_n/\log n \to \infty$. Applied to the training subgraph, where $\lambda_m \asymp n\tau\rho_n$, this again gives $\tau\rho_n = \omega(\log n/n)$. Therefore, in the classical sparse scaling regime, the training-stage exact-recovery requirements for both spectral and likelihood-based methods are compatible with Corollary 1.

## 3.2 General Stochastic Block Models

Now we turn to the general stochastic block model, as introduced in Section 2.1. We note that most existing theoretical studies on network cross-validation and model selection are conducted under this general SBM framework (Chen and Lei, 2018; Li et al., 2020; Chakrabarty et al., 2025; Yang et al., 2025). For the sake of comparison and to align with the prevailing literature, we therefore investigate the consistency of INCV in this setting. Compared to the affiliation SBM, the general SBM allows for heterogeneous block-wise connection probabilities, which necessitates additional assumptions and technical refinements in the analysis.

*Assumption* 4 (Column-wise identifiable SBM). The true stochastic block model satisfies the following condition: for each true community $v \in [K^*]$, the $v$-th column of the connectivity matrix $P = (p_{uv})_{u,v=1}^{K^*}$ admits a unique maximizer

$$\zeta(v) := \arg \max_{u \in [K^*]} p_{uv},$$

and the mapping $v \mapsto \zeta(v)$ is a bijection on $[K^*]$. Moreover, assume that there exists a nonincreasing sequence $\{\delta_n\}$ such that

$$0 < \delta_n < \min_{u,v \in [K^*]} p_{uv} < \max_{u,v \in [K^*]} p_{uv} < 1 - \delta_n < 1.$$

*Remark* 5. Assumption 4 is tailored to the current one-vs-all classification step in INCV. Conditional on accurate training labels, a test node from true community $v$ is classified by comparing its empirical connectivities to the estimated training communities. The population version of this rule is therefore determined by the largest entry in the $v$-th column of the block probability matrix $P$. The uniqueness of this maximizer guarantees that the population classifier is well-defined, while the bijectivity of $v \mapsto \zeta(v)$ rules out the possibility that two distinct true communities are mapped to the same estimated community. Without this condition, the classification step may systematically mix different true communities, so that the estimated labels on $\mathcal{N}^{(2)}$ no longer align with the true labels up to a global permutation.

  This condition is stronger than the classical identifiability condition based on distinct rows of the block probability matrix. It should be viewed as a technical condition for the present one-vs-all classifier, rather than as a necessary condition for the general INCV framework. A different induction rule, for example one based on a full likelihood or another multi-class classifier, may allow weaker identifiability conditions. In the supplementary material, we provide an illustrative experiment showing that when this column-wise separation condition is violated, the one-vs-all induction step can fail even when the block matrix has distinct rows, but a likelihood-based induction rule can substantially improve performance, suggesting that the condition is tied to the simple induction rule rather than to the general INCV framework itself.

  At the same time, Assumption 4 is more general than the conventional assortative condition $\zeta(v) = v$ for all $v$, which requires within-community connections to be the strongest. By allowing $\zeta(v) \neq v$, it also covers certain disassortative or bipartite-like structures, where the strongest connections occur across different communities.

**Definition 2** (Classification gap). Under Assumption 4, in order to quantify the separation in

the general SBM, we define

$$\beta_n := \min_{v \in [K^*]} \left( p_{\zeta(v)v} - \max_{u \neq \zeta(v)} p_{uv} \right),$$

which measures the minimal column-wise gap between the most likely and the second most likely communities in the classification step. In the case of the affiliation SBM, $\beta_n = p - q$.

Here a larger value of $\beta_n$ leads to sharper tail probability bounds for the classification error on $N^{(2)}$. Now we can state our main theorem for general SBMs.

**Theorem 2** (Consistency of INCV for general SBMs). *Consider an undirected network $(N, E)$ generated from a stochastic block model with a true community number $K^*$ satisfying $2 \leq K^* \leq K_{\max}$ for some positive integer $K_{\max}$. The INCV has a splitting ratio $\tau = |N^{(1)}|/|N| \in (0, 1)$. Under Assumptions 1, 3 and 4 with the estimated-label part of Assumption 1 required only for candidate values $K \leq K^*$, if the following requirements are met under likelihood objective:*

*(i) $\tau = \omega(1/n), \tau\beta_n^2 = \omega(\log n/n)$,*

*(ii) $(1 - \tau)\delta_n = \omega(1/n), (1 - \tau)\beta_n^2 = \omega(\log n |\log(\delta_n/2)|/n)$,*

*or the above second condition modified to*

*(ii′) $1 - \tau = \omega(1/n), (1 - \tau)\beta_n^2 = \omega(\log n/n)$,*

*for the MSE objective, then the optimal community number $\hat{K}_{\mathrm{INCV}}$ obtained from the INCV procedure satisfies*

$$\lim_{n \to \infty} \Pr(\hat{K}_{\mathrm{INCV}} < K^*) = 0.$$

Since Theorem 2 establishes only half-consistency, the proof only needs to compare candidate values $K < K^*$ with the true value $K^*$. Therefore, the estimated-label balance requirement in Assumption 1 is used only for $K \leq K^*$, and no balance condition is imposed on overfitted candidate partitions $K > K^*$ for this result.

*Remark* 6. Suppose the true model is the affiliation stochastic block model with within- and between-community connection probabilities $p$ and $q$, but we apply INCV and analyze it under the general SBM framework of Theorem 2. Then as mentioned before, the column-wise separation parameter $\beta_n$ is specialized to $\beta_n = p - q$. Consequently, the regularity conditions in Theorem 2 reduce to the familiar affiliation form, for example

$$\tau = \omega(1/n), \qquad \tau(p - q)^2 = \omega\left(\frac{\log n}{n}\right),$$

and similarly for the conditions involving $(1 - \tau)$ in both objectives. The main practical consequence is that, when one does not exploit the affiliation structure in the analysis (i.e. treats the model as a general SBM), the theoretical guarantee provided by Theorem 2 is only half-consistent. In contrast, when the affiliation structure is acknowledged and leveraged (see Theorem 1), INCV can be shown to achieve full consistency without additional penalization.

Under the general stochastic block model, the half-consistency result established in Theorem 2 is in line with existing network cross-validation literature. In particular, prior works such as Chen and Lei (2018), Li et al. (2020), and Chakrabarty et al. (2025) also guarantee that $\Pr(\hat{K} < K^*) \to 0$, while stopping short of establishing full consistency. Our result shows that this limitation persists under the proposed INCV framework, suggesting that without additional structural assumptions, ruling out overfitting via cross-validation alone remains challenging.

To illustrate the underlying difficulty, we consider the MSE objective. For a candidate community number $K$, the difference between the empirical loss at $K$ and that at the true community number $K^*$ concentrates around

$$\sum_{k_1,k_2,l_1,l_2} |T_{k_1 k_2 l_1 l_2}| (\hat{p}_{k_1 k_2} - p_{l_1 l_2})^2,$$

where

$$T_{k_1 k_2 l_1 l_2} := \big\{ (i,j) : i,j \in \mathcal{N}^{(2)}, \, i < j, \, \hat{\phi}_K(i) = k_1, \, \hat{\phi}_K(j) = k_2, \, \phi^*(i) = l_1, \, \phi^*(j) = l_2 \big\}.$$

When $K > K^*$, overfitting may occur through splitting one true community into two sub-communities. In this case, the estimated block probabilities $\hat{p}_{k_1 k_2}$ may remain close to the corresponding true parameters $p_{l_1 l_2}$, so that the leading-order behavior of the above sum is not substantially altered. As a result, both the expectation of the loss difference and its stochastic deviation (around this mean) are of the same order, making it impossible to separate the overfitted model from the true model with high probability. This lack of a detectable loss gap intuitively explains the challenge for establishing full consistency by CV alone.

Despite this limitation, our analysis offers a complementary perspective by demonstrating that the half-consistency phenomenon persists across a wide range of network regimes encompassed by our framework. In particular, the results reveal that the observed limitation of cross-validation is not tied to specific sparsity scalings, but arises under mild identification assumptions and broadly applicable conditions. This observation highlights the role of the INCV framework as a general platform for examining the inherent behavior of cross-validation in network model selection. Similarly to Corollary 1, we have the following corollary for the general SBM case.

**Corollary 2.** *Consider an undirected network $(\mathcal{N}, \mathcal{E})$ generated from a general stochastic block model with connectivity matrix $P = \rho_n P_0$, where $P_0$ satisfies the first part of Assumption 4. Under Assumptions 1 and 3 with the estimated-label part of Assumption 1 required only for candidate values $K \leq K^*$, if the following requirements are met under likelihood objective:*

*(i) $\tau = \omega(1/n), \tau \rho_n = \omega(\log n / n),$*

*(ii) $1 - \tau = \omega(1/n), (1-\tau)\rho_n^{3/2} = \omega(\sqrt{\log n}/n),$*

*or the above second condition modified to*

*(ii') $1 - \tau = \omega(1/n), (1-\tau)\sqrt{\rho_n} = \omega(\sqrt{\log n}/n),$*

*for the MSE objective, then the optimal community number $\hat{K}_{\mathrm{INCV}}$ obtained from the INCV procedure satisfies*

$$\lim_{n \to \infty} \Pr(\hat{K}_{\mathrm{INCV}} < K^*) = 0.$$

Notice that here $\beta_n \asymp \rho_n$, and thus this result is analogous to that in Corollary 1.

*Remark* 7. One possible remedy to prevent overfitting is to introduce an explicit complexity penalty, as proposed in Yang et al. (2025). In detail, use likelihood objective as an example, we may modify it as

$$\check{\ell}(\{\widehat{p}_{uv}\}, \hat{\phi}_K) = \sum_{u,v=1}^{K} \sum_{(i,j)\in\mathcal{E}_{uv}} \left( \left\{ A_{ij}^{(22)} \log \widehat{p}_{uv} + (1 - A_{ij}^{(22)}) \log(1 - \widehat{p}_{uv}) \right\} + d_K \lambda_n \right),$$

where $d_K$ is the complexity term for SBM model with $K$ communities, and $\lambda_n$ represents the penalty order. For the setting in Corollary 2, $d_K = K(K+1)/2$ and $\lambda_n \asymp \frac{\rho_n^2}{\sqrt{\log n}}$ shall guarantee full consistency.

However, such a result relies on additional regularization beyond the principle cross-validation itself, while our focus is to examine under various structural conditions whether cross-validation alone is sufficient to achieve consistency.

*Remark* 8. Theorem 2 and Corollary 2 also reflect the necessity of additional structural assumptions in order to distinguish certain overfitting configurations. In particular, when the block-wise connectivity matrix allows each block to be estimated independently, then splitting a true community into two sub-communities may lead to lack of sufficient signal.

However, if additional structure is imposed then such degeneracy can be resolved. For example, when all the off-diagonal entries are equal, splitting one true community then forces a non-negligible number of truly within-community validation edges to be treated as between-community edges, which perturbs the pooled estimate of the between-community probability and creates a validation loss gap, which serves well the affiliation SBM as a special case. Characterizing the minimal structural conditions required to guarantee full consistency remains an interesting direction for future research.

# 4  Simulations

In this section, we present some simulation results to evaluate the performance of the INCV method. Since full consistency of INCV is established for the affiliation SBM and all the conventional CV methods (including both node- and edge-splitting ones) without penalization are only half-consistent (in which case the PNN-CV method performs better in terms of achieving full consistency, see Yang et al. (2025)), we focus on the affiliation SBM in the simulations.

Our first simulation is to explore how accurately the proposed method can determine the community number under different network sizes, which helps us to understand the speed of convergence. We simulate networks under the affiliation stochastic block model with a known community number $K^* = 4$. When generating the network connections, we consider three different combinations of the within-community probability $p$ and the between-community probability $q$. For the hardest case $p = 0.3$ and $q = 0.2$, we additionally consider larger network sizes $n \in \{200, 400, 600, 800, 1000\}$. We also run the INCV method under four choices of the cross validation fold number: $2, 3, 5$ and $10$. The success rate of INCV is calculated from 100 iterations and plotted against the network size $n$ in Figure 2, representing both likelihood and MSE objective. Here NLL represents for negavite log-likelihood.

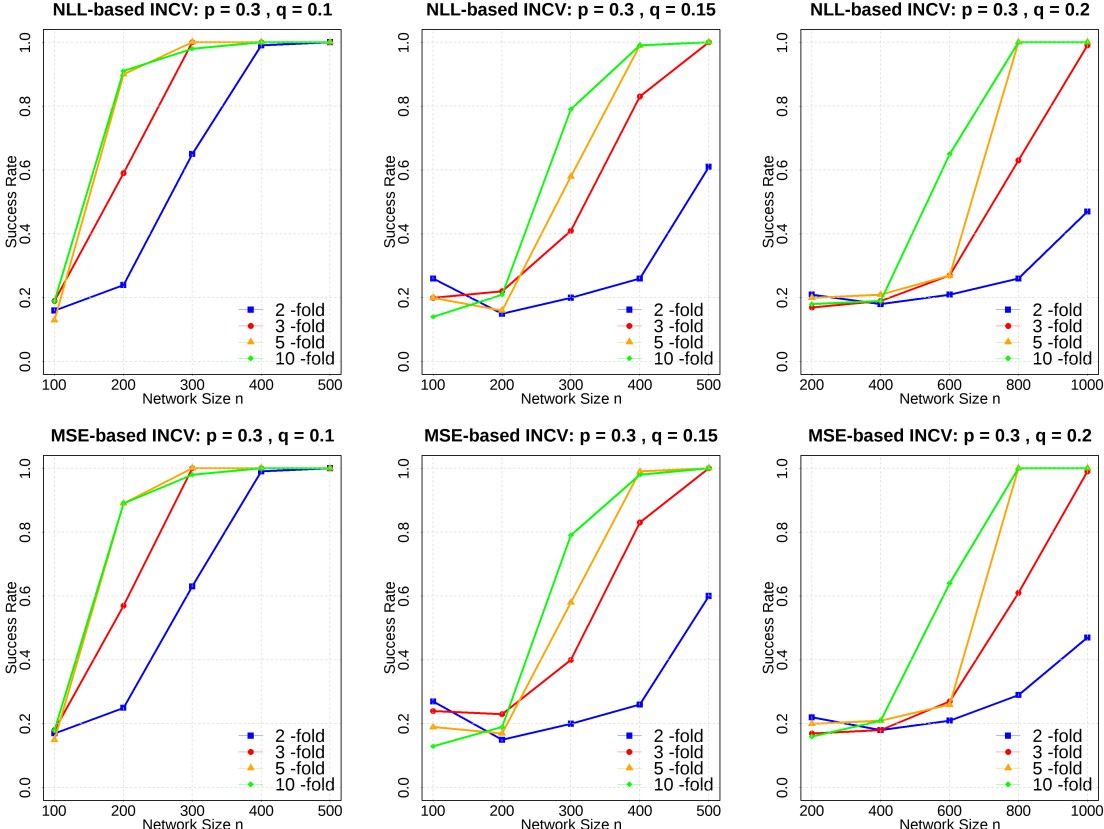

Figure 2: INCV success rates over the network sizes in three scenarios and two objectives.

Figure 2 shows that the trend for the two objectives are mainly the same. The INCV success rate is generally low at $n = 100$, and in most cases its success rate will improve quickly as the network size increases. Across all settings, larger fold numbers tend to yield faster convergence, with 10-fold cross-validation consistently outperforming 5-fold and smaller folds, achieving the highest success rates. Therefore, we describe the convergence behavior mainly based on the 10-fold results.

More precisely, in the easiest case with $p = 0.3$ and $q = 0.1$, both 5-fold and 10-fold INCV reach success rates above 0.9 and converge around $n = 300$. For the moderately difficult case $p = 0.3$ and $q = 0.15$, convergence is slightly slower, but the success rate still reaches 1.0 for 5-fold and 10-fold INCV around $n = 400$. In the hardest case $p = 0.3$ and $q = 0.2$, the convergence speed slows down substantially, with the success rate approaching 1.0 only when $n$ increases to around 800.

Next we explore how different levels of network sparsity and the imbalance of community sizes would affect the INCV performance. In this simulation, we consider a network generated from affiliation SBM that has $n = 500$ nodes with a varying true community number $K^* \in \{2, 3, 6, 8\}$. To change the network sparsity levels, we choose the between-community connection probability $q \in \{0.02, 0.05, 0.075, 0.1, 0.15, 0.2, 0.25, 0.3\}$ and set the within-community probability $p = 3q$. Furthermore, we control the community size imbalance level by changing the smallest community size in the network (denote as $n_{\min}$), while maintaining the remaining communities at nearly equal sizes. At each scenario of a true community number $K^*$, we compare the success rate

of the 10-fold INCV calculated from 100 iterations. Our simulation results are summarized in Figure 3.

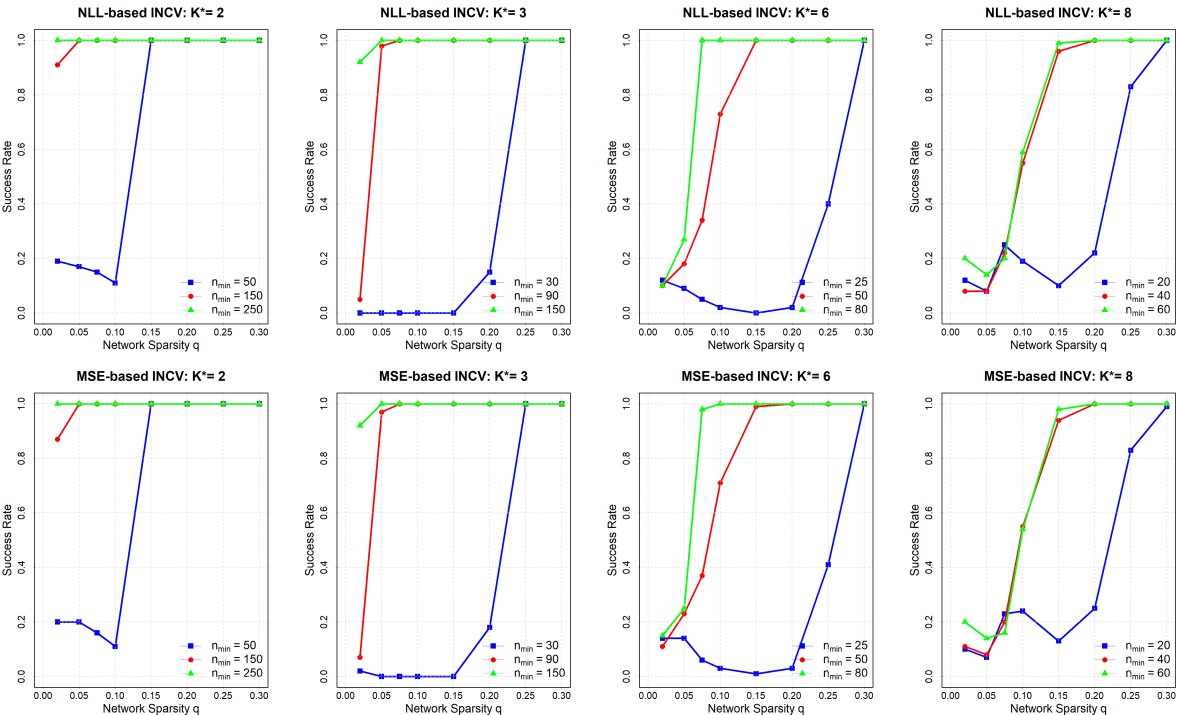

Figure 3: 10-fold INCV success rates over the network sparsity levels varying by choosing the between-community probability q. The community imbalance levels are controlled by the smallest community size $n_{\min}$.

Figure 3 shows that in general the INCV method achieves a better performance when (1) the community numbers are low (e.g. $K^* = 2, 3,$ or $4$), (2) the network is not too sparse (e.g. $q \geq 0.15$), and (3) the network has relatively balanced community sizes (i.e. $n_{\min} \approx n/K^*$). It is also shown when the community number $K^*$ is low, the INCV method has more robust performance against imbalanced community sizes and sparse networks. On the other hand, at higher $K^*$, the INCV performance becomes low when the network contains a small community. In other words, to determine the community number $K$ would be a challenging task for a sparse network with many underlying imbalanced communities.

In the last simulation, we compare INCV with several existing model selection methods for networks. The first group consists of cross-validation-based methods, including NCV (Chen and Lei, 2018), ECV (Li et al., 2020), and NETCROP (Chakrabarty et al., 2025). We also include two BIC-type likelihood criteria, LRBIC (Wang and Bickel, 2017) and CLBIC (Saldana et al., 2017), as additional non-CV benchmarks. We set the size of the network fixed at $n = 500$ and varying $K^* \in \{3, 6\}$. The success rates of different methods are displayed in Table 1 at different network sparsity and unbalanced levels. Similarly as in the above simulation, we set $p = 3q$ with $q \in \{0.05, 0.1, 0.2, 0.3\}$ and for different $K^*$ we consider diverse candidate set for $n_{\min}$.

Several observations can be made from Table 1. First, among the cross-validation-based methods, INCV is highly competitive and often improves upon NCV and ECV. As $n_{\min}$ varies, INCV exhibits remarkable robustness to community size imbalance. While the accuracy of all methods generally improves as the signal-to-noise ratio increases, INCV reaches consistency

Table 1: Comparison of INCV with cross-validation-based and BIC-type model selection methods.

| $K^*$ | $q$ | $n_{\min}$ | INCV.nll | INCV.mse | NCV | ECV | NETCROP | LRBIC | CLBIC |
|---|---|---|---|---|---|---|---|---|---|
| 3 | 0.05 | 30 | 0.00 | 0.00 | 0.00 | 0.00 | 0.00 | 0.00 | 0.02 |
| | | 90 | 1.00 | 1.00 | 0.82 | 0.94 | 0.88 | 1.00 | 0.94 |
| | | 150 | 1.00 | 1.00 | 1.00 | 1.00 | 0.46 | 1.00 | 0.98 |
| | 0.1 | 30 | 0.00 | 0.00 | 0.00 | 0.00 | 0.00 | 0.52 | 0.59 |
| | | 90 | 1.00 | 1.00 | 0.99 | 1.00 | 1.00 | 1.00 | 0.91 |
| | | 150 | 1.00 | 1.00 | 1.00 | 1.00 | 1.00 | 1.00 | 0.92 |
| | 0.2 | 30 | 0.13 | 0.16 | 0.00 | 0.01 | 0.01 | 1.00 | 0.80 |
| | | 90 | 1.00 | 1.00 | 0.99 | 1.00 | 1.00 | 1.00 | 0.97 |
| | | 150 | 1.00 | 1.00 | 1.00 | 1.00 | 1.00 | 1.00 | 0.98 |
| | 0.3 | 30 | 1.00 | 1.00 | 0.96 | 1.00 | 1.00 | 1.00 | 1.00 |
| | | 90 | 1.00 | 1.00 | 0.97 | 0.98 | 1.00 | 1.00 | 1.00 |
| | | 150 | 1.00 | 1.00 | 0.96 | 0.97 | 1.00 | 1.00 | 1.00 |
| 6 | 0.05 | 25 | 0.11 | 0.17 | 0.19 | 0.07 | 0.11 | 0.00 | 0.00 |
| | | 50 | 0.21 | 0.25 | 0.20 | 0.09 | 0.00 | 0.01 | 0.06 |
| | | 80 | 0.22 | 0.21 | 0.33 | 0.25 | 0.00 | 0.64 | 0.78 |
| | 0.1 | 25 | 0.01 | 0.01 | 0.04 | 0.01 | 0.91 | 0.00 | 0.00 |
| | | 50 | 0.80 | 0.73 | 0.68 | 0.30 | 0.97 | 0.99 | 1.00 |
| | | 80 | 1.00 | 1.00 | 0.98 | 0.97 | 0.60 | 1.00 | 1.00 |
| | 0.2 | 25 | 0.01 | 0.04 | 0.24 | 0.26 | 1.00 | 0.73 | 0.93 |
| | | 50 | 1.00 | 1.00 | 0.85 | 0.92 | 1.00 | 1.00 | 1.00 |
| | | 80 | 1.00 | 1.00 | 0.90 | 0.91 | 1.00 | 1.00 | 1.00 |
| | 0.3 | 25 | 1.00 | 1.00 | 0.42 | 0.58 | 1.00 | 1.00 | 1.00 |
| | | 50 | 1.00 | 1.00 | 0.77 | 0.82 | 1.00 | 1.00 | 1.00 |
| | | 80 | 1.00 | 1.00 | 0.82 | 0.90 | 1.00 | 1.00 | 1.00 |

(accuracy $= 1.00$) significantly faster than its competitors, particularly in the more complex $K^* = 6$ scenarios. Moreover, in dense regimes ($q = 0.3$), both NCV and ECV start to struggle, while our proposed method stables at 100% accuracy. Moreover, the performance of INCV.nll and INCV.mse is comparable, suggesting that the proposed cross-validation framework is robust to the choice of objective functions.

The BIC-type methods provide strong benchmarks and perform well in many settings, especially when the signal is sufficiently strong. In most moderate or strong signal regimes, INCV also reaches the same conclusion as the BIC-type criteria, with all methods accurately selecting the true number of communities. At the same time, no method uniformly dominates across all regimes. In sparse and difficult $K^* = 6$ settings, such as $q = 0.05$ with small $n_{\min}$, the BIC-type methods may perform poorly, while INCV and other CV-based methods still retain nontrivial success rates. Conversely, in some highly imbalanced $K^* = 3$ settings, BIC-type methods are more effective than INCV. This suggests that different model selection principles may have different advantages across regimes.

A notable phenomenon in Table 1 is the non-monotonic performance of NETCROP in specific regimes (e.g., $K^* = 3, q = 0.05, n_{\min} = 150$ and $K^* = 6, q = 0.1, n_{\min} = 80$). In these cases, NETCROP's accuracy significantly deteriorates as the network becomes balanced, while INCV maintains perfect accuracy. This counter-intuitive behavior also characterizes the robustness of proposed INCV method.

Additional experiments are reported in the supplementary material. These include a report for the computational cost for different methods, an empirical diagnostic for the estimated-label balance condition, simulations under general SBM settings using alternative plug-in validation rule, comparing two induction rules under violated separation condition, and a preliminary DCBM extension. These experiments are intended to clarify the practical behavior and possible extensions of INCV beyond the main affiliation-SBM theory.

# 5    Applications

In this section, we apply the INCV method to study two real-life networks about international trade and the U.S. Senate co-sponsoring, respectively.

## 5.1    The International Trade Network

The countries around the world export and import goods and services among themselves, creating a complicated network of international trade. It is known that some countries establish a close trading partnership from geographical proximity or by joining a trade agreement such as Trans-Pacific Partnership (TPP) and Transatlantic Trade and Investment Partnership (TTIP). From a global perspective, it is valuable to understand how many distinct trading groups exist in the international trade network and how a country trades with others inside or outside the trading group.

Our international trade data come from Westveld and Hoff (2011), which includes the import and export amount (in US dollars) among 58 countries in the world from 1981 to 2000. To simplify the data processing, our analysis only focuses on the data for year 2000. The first step is to construct the adjacency matrix for these 58 countries. Because connections are considered undirected in the stochastic block model, we add the import and export volume together to obtain the gross trade amount between every pair of the countries. Then a connection ($A_{ij} = 1$) is assigned between country $i$ and country $j$, if the gross trade amount between the two exceeds the 3rd quartile of the trade amounts for either country. Note that this definition of connection takes into account the unequal sizes of the two countries. Under this definition, although the gross trade amount between a small country (e.g. Haiti) and a large one (e.g. the United States) may only account for a small fraction of the large, the two countries can still be considered connected if the large country is a major trade partner of the small one. After obtaining the adjacency matrix for the 58 countries, we apply the 10-fold INCV to determine how many distinct communities are in the trade network. We plot the negative likelihood and MSE versus the candidate $K$ values from 2 to 10 in Figure 4(a), where $K = 4$ is shown to have the lowest value under both objectives. This concludes four distinct trading groups in the international trade network.

We plot the country names in the four detected trading groups by different colors and also the trade connections among the 58 countries in Figure 4(b). The four trading groups have a clear geographical and economic meaning. The central yellow group identifies the world's most dominant economies and industrialized powerhouses, including the G7 nations and major European traders, acting as the primary hubs of global finance; the community in the blue color are mainly for other developed countries in Europe or around the Mediterranean Sea; the red group is mainly for Asian Pacific countries including Australia and New Zealand and other Asian countries; and the green group consists of the countries in North, Central, and South America.

## 5.2    The U.S. Senate Network

In the second application, we apply the INCV method to study the political network for the U.S. Senate. Fowler (2006) had presented a dataset for all the legislative records from the 93th to

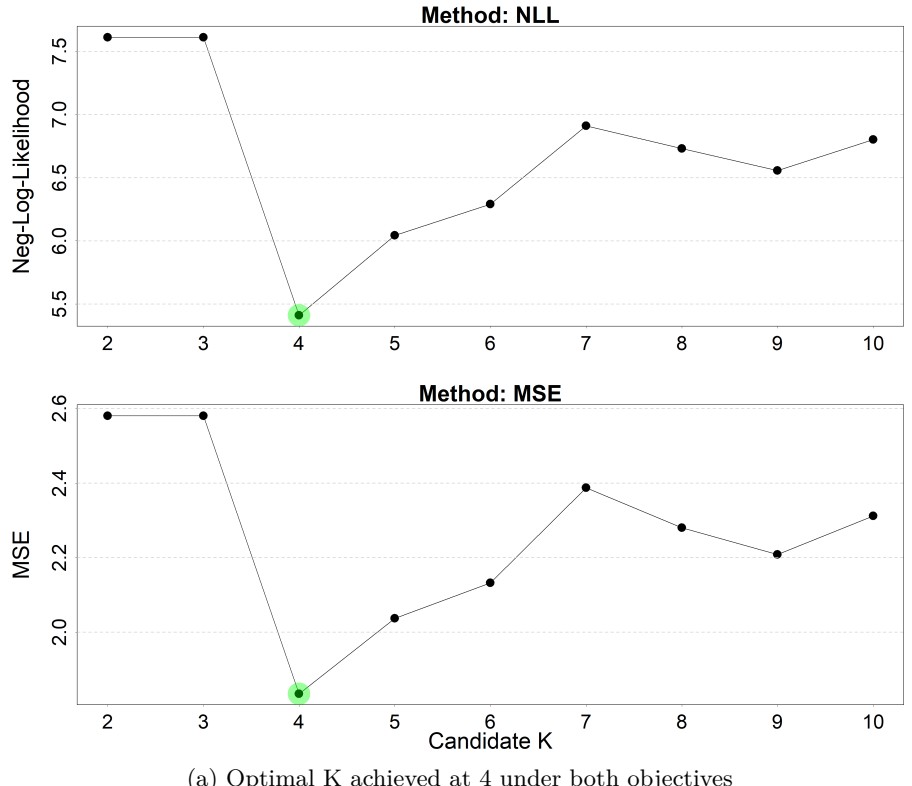

(a) Optimal K achieved at 4 under both objectives

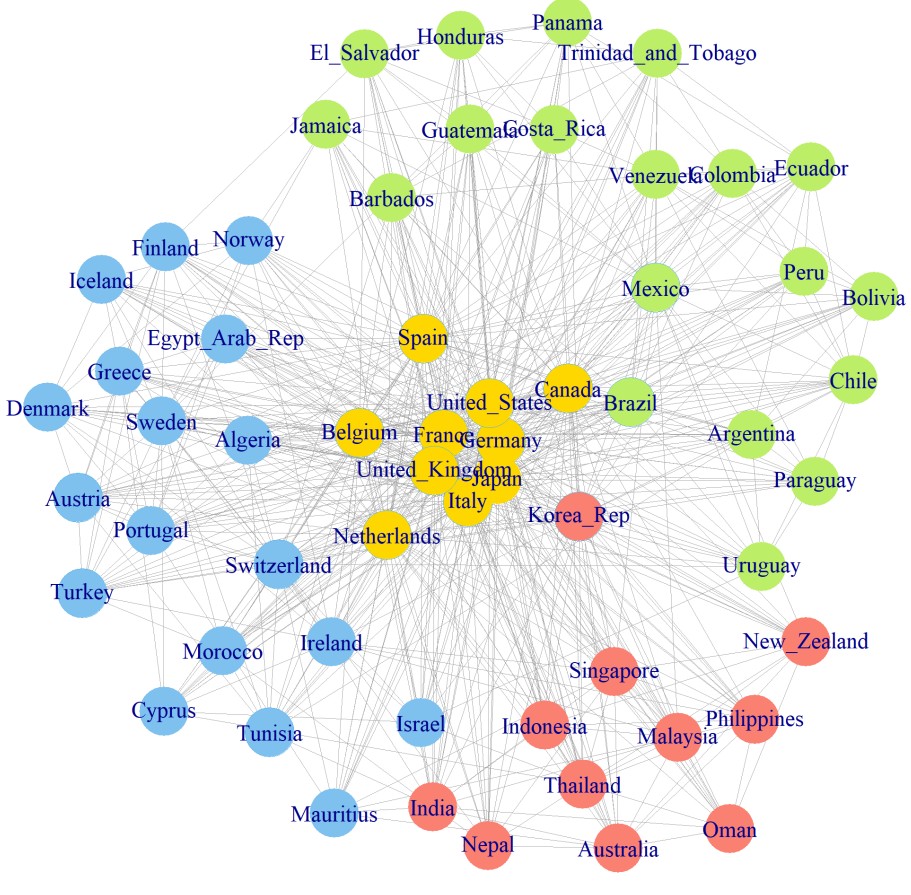

(b) Clustering of countries ($K = 4$)

Figure 4: INCV community detection for International Trade Network.

the 108th U.S. Congress between 1973 and 2004. In the current study, we re-analyze the 108th U.S. Senate co-sponsored legislation data for the 108th U.S. Senate using the INCV method to determine how many small political clusters existed inside the 108th U.S. Senate.

The 108th U.S. Senate ran from January 3, 2003 to January 3, 2005, under the third and fourth years of the George W. Bush presidency. It was composed of 51 Republicans (Majority Leader: Bill Frist, TN), 48 Democrats (Minority Leader: Tom Daschle, SD), and 1 Independent (Jim Jeffords, VT) who was aligned with Democrats according to the CRS Report for Congress. Fowler's data showed that a total of 3,035 Senate bills were introduced with 869 (or 28.6 percent) of them having no co-sponsors and the remaining 2,166 (or 71.4 percent) bills having at least one co-sponsor. The average number of co-sponsors is equal to 5.3. In our construction of the adjacency matrix, a connection ($A_{ij} = 1$) is assigned between the two Senate members, if either member had co-sponsored at least four legislative bills introduced by the other. Here we exclude those bills with many co-sponsors from this analysis, since those bills might represent a common political view shared by most of the members while not reflecting the political affiliations between the members, as well as making the adjacency matrix too saturated to be clustered.

We apply 10-fold INCV on the adjacency matrix and plot the negative log likelihood and MSE against the candidate $K$ values from 2 to 10 in Figure 5(a). While both objectives are minimized at $K = 2$, the objective values at $K = 8$ are notably comparable, suggesting a potential hierarchical structure where the two primary communities could be further partitioned into finer sub-communities. We plot the two estimated clusters and the senators' co-sponsorship connections in Figure 5(b). To facilitate further investigation, we also summarize the party composition for these two clusters. The blue community is predominantly composed of Republicans (42 out of 46), while the red group is largely dominated by Democrats (44 out of 54), exhibiting a high degree of partisan homophily. This stark contrast suggests that the underlying cosponsorship network is a robust reflection of political affiliation. Furthermore, the few cross-partisan nodes residing within these clusters may represent moderate members who facilitate inter-party communication, potentially explaining why a more granular partition (e.g., $K = 8$) also yields competitive objective values.

To further assess the structural quality of the selected community partitions, we compute modularity (Newman and Girvan, 2004; Newman, 2006) and normalized cut (Shi and Malik, 2000) for each candidate value of $K$ in the two real-data applications. For the latter metric, we report Ncut/$K$ rather than the raw multiway Ncut, because the raw multiway Ncut is a sum over the $K$ communities and therefore tends to increase mechanically with $K$. This averaging makes the metric more comparable across different candidate values of $K$. These metrics are not used in the INCV selection procedure, but provide complementary diagnostics for the resulting partitions.

Table 2 shows that the INCV-selected partitions achieve favorable values under these additional metrics. For the international trade network, the selected value $\hat{K} = 4$ attains the largest modularity among all candidate values and an AvgNcut value very close to the minimum. For the Senate network, the selected value $\hat{K} = 2$ yields the smallest AvgNcut, while its modularity is nearly identical to the maximum value attained at $K = 3$. Together with the clear interpretation of the inferred clusters for the two networks, these additional diagnostics provide further

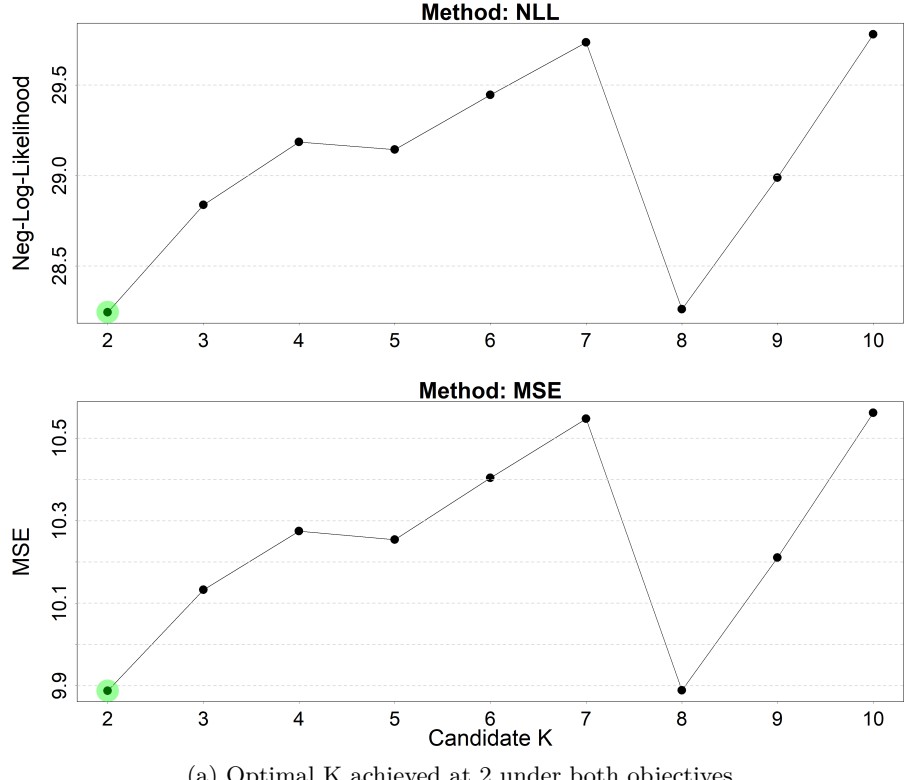

(a) Optimal K achieved at 2 under both objectives

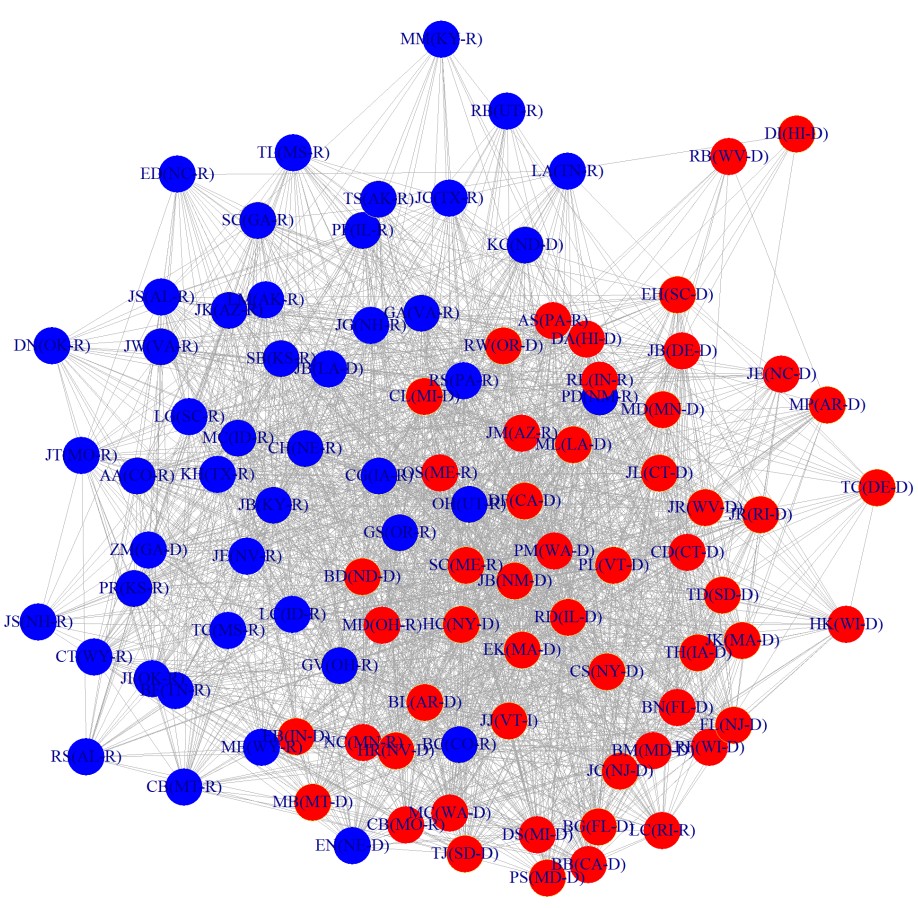

(b) Clustering of senators ($K = 2$)

Figure 5: INCV community detection for U.S. Senate Network.

Table 2: Additional clustering quality metrics for candidate community numbers in the real-data applications. Higher modularity and lower AvgNcut indicate better community separation. The INCV-selected values are highlighted in bold.

| Candidate $K$ | Trade Modularity | Trade Ncut/$K$ | Senate Modularity | Senate Ncut/$K$ |
|---|---|---|---|---|
| 2 | -0.153 | 0.661 | **0.114** | **0.382** |
| 3 | -0.002 | 0.642 | 0.116 | 0.548 |
| 4 | **0.053** | **0.649** | 0.048 | 0.711 |
| 5 | 0.052 | 0.676 | 0.015 | 0.788 |
| 6 | 0.030 | 0.762 | -0.006 | 0.836 |
| 7 | 0.023 | 0.808 | -0.019 | 0.875 |
| 8 | 0.047 | 0.824 | -0.022 | 0.903 |
| 9 | 0.014 | 0.875 | -0.005 | 0.896 |
| 10 | 0.037 | 0.869 | 0.013 | 0.889 |

support for the community structures selected by INCV.

## 6 Discussion

In this article, we propose INCV, a novel cross-validation framework based on node splitting for model selection in stochastic block models. To the best of our knowledge, this work establishes the first full consistency result for cross-validation alone under a simplified affiliation SBM setting. The main mechanism behind the full consistency result is that, under the affiliation structure, overfitting by splitting a true community necessarily misclassifies many truly within-community validation edges as between-community edges, producing a detectable validation loss gap. We further discuss the behavior of INCV under more general SBM structures, providing insight into why full consistency may be difficult to achieve in general and under what conditions it may still hold. Numerical simulations also demonstrate the robustness of the proposed method across a range of network regimes.

Looking forward, we have also identified a few enhancement opportunities for INCV. Firstly, although the main theory focuses on SBMs, the node-splitting and induction idea can be adapted to degree-corrected stochastic block models (DCSBMs, Karrer and Newman, 2011). In the supplementary material, we provide a preliminary DCSBM implementation, where the clustering, induction, and validation steps are modified to account for node-level degree heterogeneity. The numerical results suggest that such an extension is practically feasible. A rigorous theoretical analysis for DCSBMs, as well as extensions to broader network models with degree heterogeneity or covariates, remains an important direction for future work.

Secondly, our current INCV method uses the "One-vs-Other" classifier to cluster the nodes in $\mathcal{N}^{(2)}$ based on their connections to the nodes in $\mathcal{N}^{(1)}$. This rule leads to transparent analysis, but also requires a column-wise separation condition in the general SBM setting. In the supplementary material, we consider an alternative likelihood-based induction rule and show empirically that it can substantially improve performance when the column-wise separation condition is violated. More advanced multi-class classification or likelihood-based induction procedures may therefore further improve INCV, especially for sparse, unbalanced, or non-assortative networks.

Finally, as discussed in Section 3.2, whether cross-validation alone can guarantee full consistency for general SBMs, and what minimal structural assumptions are sufficient to ensure such guarantees, remain open questions. We believe that further investigation along this direction would deepen the theoretical understanding of cross-validation methods for network models.

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

## Supplemental Material

Supplemental material includes mathematical proofs and additional numerical results that support the findings of this study, with title "Supplementary Materials for 'Inductive Node-split Cross-Validation in Networks' ".

## Acknowledgement

We thank the two reviewers for their constructive comments that have substantially improved our manuscript.

### Code Availability

All analysis code is available in the GitHub repository: [https://github.com/ivylinzhang97/incv-community-detection](https://github.com/ivylinzhang97/incv-community-detection).

