# Supplementary Materials for "Inductive Node-split Cross-Validation in Networks"

Lin Zhang*

Cotality, Data Science and Platform

and

Bokai Yang

Qiuzhen College, Tsinghua University

and

Yuhong Yang†

Yau Mathematical Sciences Center, Tsinghua University

Beijing Institute of Mathematical Sciences and Applications

In this supplementary file we will produce the proofs for the two main theorems and two corollaries introduced in the main article, as well as conduct five supplementary numerical simulations to interpret several assumptions in the main text and show possible extensions and potentials for the INCV framework.

## Contents

---

*The first two authors contribute equally to this work.

†Corresponding author: yyangsc@mail.tsinghua.edu.cn

# A  Proofs

## A.1  Supporting Lemmas

We first list some supporting lemmas that will be used in our proofs.

**Lemma A.1** (Hoeffding's inequality). *Let $X_1, X_2, \ldots, X_n$ be independent random variables such that $X_i \in [a_i, b_i]$ almost surely. Let $S = X_1 + \cdots + X_n$. Then for all $t > 0$:*

$$\mathbb{P}(S - \mathbb{E}S \geq t) \leq \exp\left(-\frac{2t^2}{\sum_{i=1}^n (b_i - a_i)^2}\right).$$

**Lemma A.2** (Bernstein's inequality). *Let $X_1, X_2, \ldots, X_n$ be independent random variables such that $\mathbb{E}[X_i] = 0$ for all $i$, and assume that $|X_i| \leq M$ almost surely. Then for all $t > 0$,*

$$\mathbb{P}\left[\sum_{i=1}^n X_i \geq t\right] \leq \exp\left(-\frac{t^2/2}{\sum_{i=1}^n \mathbb{E}[X_i^2] + \frac{1}{3}Mt}\right).$$

**Lemma A.3** (Serfling (1974)). *Let $S_n \sim \text{Hypergeometric}(n; D; N)$ denote the sum of a random sample of size $n$ drawn without replacement from a population consisting of $D$ ones and $N - D$ zeros. Then for any $t > 0$,*

$$\mathbb{P}\left(S_n - \frac{nD}{N} \geq nt\right) \leq \exp\left(-\frac{2nt^2}{1 - (n-1)/N}\right), \tag{1}$$

*and*

$$\mathbb{P}\left(S_n - \frac{nD}{N} \leq -nt\right) \leq \exp\left(-\frac{2nt^2}{1 - (n-1)/N}\right). \tag{2}$$

*Proof of Lemma A.3.* The proof of the upper-tail inequality (1) follows from Serfling (1974). The lower-tail bound (2) is obtained by applying the upper-tail inequality (1) to $n - S_n \sim$ Hypergeometric$(n; N - D; N)$. $\qquad\square$

## A.2  Proof of Theorem 1

We first introduce a number of supporting lemmas for the proof under affiliation SBM case. Our next two lemmas give a lower and upper bound for the cardinalities of $W_{\phi_K}$ and $B_{\phi_K}$.

**Lemma A.4.** *For any network of size $n$ with $K \geq 2$ communities given by a community labeling function $\phi_K$, the cardinalities of $W_{\phi_K}$ and $B_{\phi_K}$ satisfy*

$$|W_{\phi_K}| \geq \frac{n^2 - nK}{2K}, \tag{3}$$

$$|B_{\phi_K}| \leq \frac{(K-1)n^2}{2K}, \tag{4}$$

*where the equality is achieved when every community have an equal size of $n/K$.*

*Proof of Lemma A.4.* Let $n_k$ denote the size of the $k$-th community in the network. Define $\alpha_k = n_k/n$ for $k = 1, 2, \ldots, K$. By Cauchy's inequality, $\sum_{k=1}^K \alpha_k^2 \geq 1/K$, where the equality

sign is achieved when $\alpha_1 = \alpha_2 = \cdots = \alpha_K = 1/K$. Hence,

$$|W_{\phi_K}| = \sum_{k=1}^{K} \binom{n_k}{2} = \frac{n^2}{2} \sum_{k=1}^{K} \alpha_k^2 - \frac{n}{2} \geq \frac{n^2 - nK}{2K}. \tag{5}$$

$$|B_{\phi_K}| = \sum_{k<\ell} n_k n_\ell = \frac{n^2 - n}{2} - |W_{\phi_K}| \leq \frac{(K-1)n^2}{2K}, \tag{6}$$

where the equality is achieved when every community has an equal size of $n/K$. □

The lemma above states the lower bound of $|W_{\phi_K}|$ has an order of $n^2$. This $n^2$ order does not apply to the lower bound of $|B_{\phi_K}|$ in a general case. An extreme example is when the first $K-1$ communities all have a size of 1 and remaining one has a size of $n - K + 1$, then $|B_{\phi_K}|$ has an order of $n$. However, if we restrict our discussion to those networks whose communities have comparable sizes, then the same $n^2$ order can be proved for the lower bound of $|B_{\phi_K}|$. Recall the balanced community structure defined in Definition 1 in the article.

**Lemma A.5.** *For a network with a balanced community structure, the cardinalities of $W_{\phi_K}$ and $B_{\phi_K}$ satisfy*

$$|W_{\phi_K}| \leq \frac{n^2}{2} \left(1 - \frac{K-1}{K} \pi_0 (2 - \pi_0)\right) - \frac{n}{2},$$

$$|B_{\phi_K}| \geq \frac{n^2 (K-1) \pi_0 (2 - \pi_0)}{2K},$$

*where the equality is achieved when there are $K-1$ communities with their sizes all equal to the minimum size $n\pi_0/K$, and the remaining one community has a size of $n - (K-1)n\pi_0/K$.*

*Proof of Lemma A.5.* Define $\alpha_k = n_k/n$ for $k = 1, 2, \ldots, K$. Then $\sum_{k=1}^{K} \alpha_k = 1$ and $\alpha_k \geq \pi_0/K$ by the balanced condition. It can be proved by contradiction that $\sum_{k=1}^{K} \alpha_k^2$ achieves its maximum when there are $K-1$ $(\alpha_k)$ equal to $\pi_0/K$, and the remaining one is equal to $1 - (K-1)\pi_0/K$, i.e.

$$\sum_{k=1}^{K} \alpha_k^2 \leq (K-1) \left(\frac{\pi_0}{K}\right)^2 + \left(1 - \frac{(K-1)\pi_0}{K}\right)^2.$$

This is because otherwise, without loss of generality, assume $\sum_{k=1}^{K} \alpha_k^2$ achieves its maximum when $\alpha_1 = \pi_0/K + \beta_1$ and $\alpha_2 = \pi_0/K + \beta_2$ for two strictly positive constants $\beta_1$ and $\beta_2$. Then

$$\sum_{k=1}^{K} \alpha_k^2 = \left(\frac{\pi_0}{K} + \beta_1\right)^2 + \left(\frac{\pi_0}{K} + \beta_2\right)^2 + \sum_{k=3}^{K} \alpha_k^2$$

$$< \left(\frac{\pi_0}{K} + \beta_1 + \beta_2\right)^2 + \left(\frac{\pi_0}{K}\right)^2 + \sum_{k=3}^{K} \alpha_k^2.$$

This contradicts to the assumption that $\sum_{k=1}^{K} \alpha_k^2$ achieves the maximum value. Thus,

$$|W_{\phi_K}| = \sum_{k=1}^{K} \binom{n_k}{2} = \frac{n^2}{2} \sum_{k=1}^{K} \alpha_k^2 - \frac{n}{2} \leq \frac{n^2}{2} \left(1 - \frac{K-1}{K} \pi_0 (2 - \pi_0)\right) - \frac{n}{2},$$

$$|B_{\phi_K}| = \sum_{k<\ell} n_k n_\ell = \frac{n^2 - n}{2} - |W_{\phi_K}| \geq \frac{n^2 (K-1) \pi_0 (2 - \pi_0)}{2K},$$

where the equality is achieved when there are $K-1$ communities with their sizes all equal to the minimum size $n\pi_0/K$, and the remaining one community has a size of $n-(K-1)n\pi_0/K$. $\square$

**Lemma A.6.** *Under Assumptions 1 and 2, the maximum likelihood estimate $\hat{p}$ and $\hat{q}$ as defined in Section 3.1 for any candidate $K \geq 2$ are bounded away from $0$ and $1$ in probability so that*

$$\Pr\left(\frac{\delta_n}{2} < \hat{p} < 1 - \frac{\delta_n}{2}\right) \geq 1 - 2\exp\left(-C_p n^2 (1-\tau)^2 \delta_n^2\right), \tag{7}$$

$$\Pr\left(\frac{\delta_n}{2} < \hat{q} < 1 - \frac{\delta_n}{2}\right) \geq 1 - 2\exp\left(-C_q n^2 (1-\tau)^2 \delta_n^2\right), \tag{8}$$

*where $\tau$ is the INCV node splitting ratio, $\delta_n$ is defined in Assumption 2, $C_p$ and $C_q$ are two positive constants only dependent on $K$ and $\pi_0$ (from Assumption 1).*

*Proof of Lemma A.6.* First consider $\hat{q}$. Recall that

$$\hat{q} = \frac{\sum_{(i,j)\in B^{(2)}_{\hat{\phi}_K}} A^{(22)}_{ij}}{|B^{(2)}_{\hat{\phi}_K}|}.$$

Because every $A^{(22)}_{ij}$ has an expected value equal to $p$ or $q$, it is obvious by Assumption 2 that

$$0 < \delta_n < q \leq \mathbb{E}(\hat{q}) \leq p < 1 - \delta_n < 1. \tag{9}$$

Note that by Assumption 1 the recovered community structure (under $\hat{\phi}_K$) is known balanced with respect to a constant $\pi_0 \in (0,1)$ on the training set $\mathcal{N}^{(1)}$. In the Classification step of INCV, we first classify the nodes in the test set $\mathcal{N}^{(2)}$ to the $K$ communities according to Section 2.2. Let $n^{(2)}_k$ denote the number of nodes in $\mathcal{N}^{(2)}$ being classified to the $k$-th community for $k = 1, 2, \ldots, K$.

By the discussion in Remark 1 after Assumption 1, we know that the recovered label $\hat{\phi}_K$ on $\mathcal{N}^{(2)}$ is still balanced with respect to constant $\pi_0$. Thus by applying Lemma A.4 and Lemma A.5 to the recovered network of $\mathcal{N}^{(2)}$, we have

$$\frac{n^2(1-\tau)^2(K-1)\pi_0(1-\pi_0)}{2K} \leq |B^{(2)}_{\hat{\phi}_K}| \leq \frac{(K-1)n^2(1-\tau)^2}{2K}.$$

Next using the Law of Total Probability and Hoeffding's Inequality, we have

$$\Pr\left(|\hat{q} - \mathbb{E}(\hat{q})| \geq \frac{\delta_n}{2}\right) = \sum_{h=\frac{n^2(1-\tau)^2(K-1)\pi_0(1-\pi_0)}{2K}}^{\frac{(K-1)n^2(1-\tau)^2}{2K}} \Pr\left(|\hat{q} - \mathbb{E}(\hat{q})| \geq \frac{\delta_n}{2} \mid |B^{(2)}_{\hat{\phi}_K}| = h\right) \Pr\left(|B^{(2)}_{\hat{\phi}_K}| = h\right)$$

$$\leq \sum_{h=\frac{n^2(1-\tau)^2(K-1)\pi_0(1-\pi_0)}{2K}}^{\frac{(K-1)n^2(1-\tau)^2}{2K}} 2\exp\left(-\frac{\delta_n^2}{2}h\right) \Pr\left(|B^{(2)}_{\hat{\phi}_K}| = h\right)$$

$$\leq 2\exp\left(-\frac{n^2(1-\tau)^2(K-1)\pi_0(1-\pi_0)\delta_n^2}{4K}\right)$$

$$\leq 2\exp\left(-C_q n^2(1-\tau)^2\delta_n^2\right). \tag{10}$$

Equations (9) and (10) together imply

$$\Pr\left(\frac{\delta_n}{2} < \hat{q} < 1 - \frac{\delta_n}{2}\right) \geq 1 - 2\exp\left(-C_q n^2 (1-\tau)^2 \delta_n^2\right).$$

The probability inequality (7) for $\hat{p}$ can be proved in a similar manner. $\qquad\square$

**Lemma A.7.** *Suppose an undirected network $(\mathcal{N}, \mathcal{E})$ generated from a stochastic block model satisfied Assumptions 1–3. When $K = K^*$, the derived community labeling function $\hat{\phi}_{K^*}$ on $\mathcal{N}^{(2)}$ in the Classification step of INCV satisfies*

$$\Pr(\hat{\phi}_{K^*} \neq \phi^*) \leq 2K^* n(1-\tau)\exp\left(-c_1 n\tau(p-q)^2\right) + K^*\exp(-c_2 n(1-\tau)) + g(n\tau),$$

*where $\tau$ is the node splitting ratio, $p$ and $q$ are the within and between communities connection probabilities, $g(x)$ is the function defined in Assumption 3, and $c_1$ and $c_2$ are positive constants dependent on $\pi_0$ (from Assumption 1) and $K^*$ only.*

*Proof of Lemma A.7.* By Assumption 1, there exists a positive constant $\pi_0 \in (0,1)$ such that

$$n_{\min} = \min\{n_1, n_2, \ldots, n_{K^*}\} \geq \frac{n\pi_0}{K^*}.$$

Let the notation $n_k^{(1)}$ denote the cardinality of a subset of community $k$ that belongs to the training set $\mathcal{N}^{(1)}$. Thus, each $n_k^{(1)}$ follows a hypergeometric distribution with parameter $(\tau n, n_k, n)$.

By setting $t = n_k/(2n)$ in (2) of Lemma A.3, we have for any $n_k^{(1)}$,

$$\Pr\left(n_k^{(1)} \leq \frac{n\pi_0\tau}{2K^*}\right) \leq \Pr\left(n_k^{(1)} \leq \frac{n_k\tau}{2}\right)$$

$$\leq \Pr\left(n_k^{(1)} - \tau n\frac{n_k}{n} \leq -\tau n\frac{n_k}{2n}\right)$$

$$\leq \exp\left(-\frac{2\tau n(n_k^2)/(4n^2)}{1 - (\tau n - 1)/n}\right)$$

$$\leq \exp\left(-\frac{\tau\pi_0^2 n}{2K^{*2}(1 - (\tau n - 1)/n)}\right)$$

$$\leq \exp(-c_0 n\tau), \tag{11}$$

for a positive constant $c_0$ dependent on $K^*$ and $\pi_0$ only.

Next we consider the case when $\hat{\phi}_{K^*} = \phi^*$ on $\mathcal{N}^{(1)}$. In this case, for any node $j \in \mathcal{N}^{(2)}$, let $l_j = \phi^*(j)$ denote its true community. Then according to the classification criterion, we have

$$\Pr\left(\hat{\phi}_{K^*}(j) \neq \phi^*(j)\right) \leq \sum_{\substack{l=1 \\ l \neq l_j}}^{K^*} \Pr\left(\frac{\sum_{i \in \mathcal{N}^{(1)}:\phi^*(i)=l_j} A_{ij}^{(12)}}{n_{l_j}^{(1)}} < \frac{\sum_{i \in \mathcal{N}^{(1)}:\phi^*(i)=l} A_{ij}^{(12)}}{n_l^{(1)}}\right)$$

$$\leq \sum_{\substack{l=1 \\ l \neq l_j}}^{K^*} \left[\Pr\left(\frac{\sum_{i \in \mathcal{N}^{(1)}:\phi^*(i)=l_j} A_{ij}^{(12)}}{n_{l_j}^{(1)}} - p < -\frac{p-q}{2}\right)\right.$$

$$\left. + \Pr\left(\frac{\sum_{i \in \mathcal{N}^{(1)}:\phi^*(i)=l} A_{ij}^{(12)}}{n_l^{(1)}} - q > \frac{p-q}{2}\right)\right]. \tag{12}$$

Applying the Law of Total Probability and Hoeffding Inequality, we obtain

$$\Pr\left(\frac{\sum_{i \in \mathcal{N}^{(1)}:\phi^*(i)=l_j} A_{ij}^{(12)}}{n_{l_j}^{(1)}} - p < -\frac{p-q}{2}\right)$$

$$\leq \sum_{h=0}^{n\tau} \Pr\left(\frac{\sum_{i \in \mathcal{N}^{(1)}:\phi^*(i)=l_j} A_{ij}^{(12)}}{n_{l_j}^{(1)}} - p < -\frac{p-q}{2} \;\middle|\; n_{l_j}^{(1)} = h\right) \Pr(n_{l_j}^{(1)} = h)$$

$$\leq \sum_{h=\frac{n\pi_0\tau}{2K^*}}^{n\tau} \exp\left(-\frac{h(p-q)^2}{2}\right) \Pr(n_{l_j}^{(1)} = h) + \Pr\left(n_{l_j}^{(1)} \leq \frac{n\pi_0\tau}{2K^*}\right)$$

$$\leq \exp\left(-\frac{n\tau\pi_0(p-q)^2}{4K^*}\right) + \exp(-c_0 n\tau)$$

$$\leq \exp\left(-c_1 n\tau(p-q)^2\right).$$

Similarly, we can prove

$$\Pr\left(\frac{\sum_{i \in \mathcal{N}^{(1)}:\phi^*(i)=l} A_{ij}^{(12)}}{n_l^{(1)}} - q > \frac{p-q}{2}\right) \leq \exp\left(-c_1 n\tau(p-q)^2\right).$$

Therefore, (12) can be rewritten as

$$\Pr\left(\hat{\phi}_{K^*}(j) \neq \phi^*(j)\right) \leq \sum_{\substack{l=1 \\ l \neq l_j}}^{K^*} 2\exp\left(-c_1 n\tau(p-q)^2\right)$$

$$= 2K^* \exp\left(-c_1 n\tau(p-q)^2\right).$$

It is also known by Assumption 3 that the probability of $\hat{\phi}_{K^*} \neq \phi^*$ on $\mathcal{N}^{(1)}$ is bounded by $g(n\tau)$. Thus on $\mathcal{N}^{(2)}$, the $\hat{\phi}_{K^*}$ obtained according to the classification step shall satisfy

$$\Pr(\hat{\phi}_{K^*} \neq \phi^*) \leq \sum_{j \in \mathcal{N}^{(2)}} \Pr\left(\hat{\phi}_{K^*}(j) \neq \phi^*(j)\right) + g(n\tau)$$

$$\leq n(1-\tau)2K^* \exp\left(-c_1 n\tau(p-q)^2\right) + g(n\tau)$$

$$\leq 2K^* n(1-\tau) \exp\left(-c_1 n\tau(p-q)^2\right) + g(n\tau). \tag{13}$$

Finally, by using a similar argument as (11), we can show there exists a positive constant $c_2$ such that

$$\Pr\left(n_k^{(2)} \leq \frac{n\pi_0(1-\tau)}{K^*}\right) \leq \exp(-c_2 n(1-\tau)), \quad k = 1, 2, \ldots, K^*.$$

Hence,

$$\Pr\left(\min_{1 \leq k \leq K^*} n_k^{(2)} \leq \frac{n\pi_0(1-\tau)}{K^*}\right) \leq K^* \exp(-c_2 n(1-\tau)). \tag{14}$$

Recall that the re-classification process as described in Remark 1 is only needed when the recovered network structure for $\mathcal{N}^{(2)}$ is not balanced with respect to $\pi_0$. Thus according to (13) and (14), when $K = K^*$, the derived labeling function $\hat{\phi}_{K^*}$ on $\mathcal{N}^{(2)}$ even with the re-classification

shall satisfy

$$\Pr(\hat{\phi}_{K^*} \neq \phi^*) \leq 2K^*n(1-\tau)\exp\left(-c_1 n\tau(p-q)^2\right) + K^*\exp(-c_2 n(1-\tau)) + g(n\tau).$$

$\square$

Lemma A.7 applies to the case where $K = K^*$. When a candidate $K \neq K^*$ in INCV, define

$$
\begin{aligned}
T_{11} &:= \{(i,j) : i,j \in \mathcal{N}, i < j, \hat{\phi}_K(i) = \hat{\phi}_K(j), \phi^*(i) = \phi^*(j)\}, \\
T_{12} &:= \{(i,j) : i,j \in \mathcal{N}, i < j, \hat{\phi}_K(i) \neq \hat{\phi}_K(j), \phi^*(i) = \phi^*(j)\}, \\
T_{21} &:= \{(i,j) : i,j \in \mathcal{N}, i < j, \hat{\phi}_K(i) = \hat{\phi}_K(j), \phi^*(i) \neq \phi^*(j)\}, \\
T_{22} &:= \{(i,j) : i,j \in \mathcal{N}, i < j, \hat{\phi}_K(i) \neq \hat{\phi}_K(j), \phi^*(i) \neq \phi^*(j)\}.
\end{aligned}
\tag{15}
$$

The next two lemmas show that when $K > K^*$ the cardinalities of $T_{12}$ and $T_{22}$ have a lower bound, and when $K < K^*$ the cardinalities of $T_{11}$ and $T_{21}$ have a lower bound.

**Lemma A.8.** *Under Assumption 1, for any $K^+ > K^*$, there exist positive constants $\gamma_1, \gamma_2 \in (0,1)$ depending on $K^+$, such that*

$$|T_{12}| \geq \gamma_1 \frac{\pi_0^2}{K^{*2}} n^2, \tag{16}$$

$$|T_{22}| \geq \gamma_2 \frac{\pi_0^2}{K^{*2}} n^2. \tag{17}$$

*Proof of Lemma A.8.* First, consider the true community structure in the network as $\mathcal{N} = [n] = \bigcup_{k=1}^{K^*} C_k^*$ with $|C_k^*| = n_k$. For a candidate $K^+ > K^*$, we have the estimated community structure as $\mathcal{N} = \bigcup_{k=1}^{K^+} \hat{C}_k$ with $|\hat{C}_i| = \hat{n}_i$. Because the recovered network is assumed to have a balanced community structure under Assumption 1, there exists a positive constant $\pi_0 \in (0,1)$ such that $\hat{n}_{\min} = \min_{1 \leq i \leq K^+}(\hat{n}_i) \geq \pi_0 n / K^+$.

Since $K^+ > K^*$, there must be at least one community under the true labeling function $\phi^*$ that has a certain proportion of its nodes being assigned to different communities, and the proportion is at least of order $\delta_1 n_{\min}/n$ for some positive $\delta_1 \in (0,1)$ depending on $K^+$. Otherwise, we assume for each of the original communities, given any small $\epsilon > 0$, the number of nodes that are split out satisfies

$$S_k < n_k \epsilon, \quad k = 1, \ldots, K^*.$$

Then the nodes that are split out have cardinality at most

$$\sum_{k=1}^{K^*} S_k < n\epsilon.$$

Because $\epsilon$ is arbitrarily small, there must be at least one community whose size is smaller than $n\epsilon$. This contradicts the balanced community structure assumption that $\min_{1 \leq k \leq K^*} n_k \geq n\pi_0/K^*$. Without loss of generality, we may assume the 1st community based on the true labeling function $\phi^*$ has at least $\delta_1 n_{\min}$ nodes assigned to different communities under the estimated labeling

function $\hat{\phi}$ in INCV. Thus,

$$|T_{12}| \geq (n_1 - \delta_1 n_{\min})(\delta_1 n_{\min}) \geq \delta_1(1 - \delta_1)\frac{\pi_0^2}{K^{*2}}n^2.$$

This concludes (16).

To prove (17), without loss of generality assume that the 1st community has the largest size $n_1$ among all communities under $\phi^*$. When the network is re-clustered to $K^+$ communities under $\hat{\phi}$, at least one of the $K^+$ communities (say Community A) will contain a minimum of $n'_1 \in [n_1/K^+, n_1]$ nodes from the 1st community. Consider the following two scenarios:

(1) $\limsup(n'_1/n_1) = 1$;

(2) $\limsup(n'_1/n_1) < 1$.

In the first scenario, there exists at least another community under $\phi^*$ that will have a minimum of $\rho \min_{1 \leq k \leq K^*} n_k$ nodes assigned to a different Community B under $\hat{\phi}$ for some $\rho \in (0,1)$. This is because otherwise, every other community under $\phi^*$ must have at least $(1-\epsilon_k)n_k$ nodes assigned to Community A under $\hat{\phi}$ for arbitrarily small $\epsilon_k = o(1)$, $k = 2, \ldots, K^*$. Consequently, the number of nodes not assigned to Community A will be at most $(n_1 - n'_1) + \sum_{k=2}^{K^*} \epsilon_k n_k = o(n)$. This contradicts the balanced community structure condition. Therefore, in Scenario (1),

$$|T_{22}| \geq n'_1 \rho \min_{1 \leq k \leq K^*} n_k \geq \gamma_2 \frac{\pi_0^2}{K^{*2}}n^2,$$

for some positive constant $\gamma_2 \in (0,1)$.

In the second scenario, if the 2nd community under $\phi^*$ has $\rho n_2$ nodes not assigned to Community A under $\hat{\phi}$ for some $\rho \in (0,1)$, then the same inequality holds for $|T_{22}|$. On the other hand, if the 2nd community under $\phi^*$ has $(1 - \epsilon)n_2$ nodes assigned to Community A under $\hat{\phi}$ for some $\epsilon = o(1)$, then

$$|T_{22}| \geq (n_1 - n'_1)(1 - \epsilon)n_2 \geq \gamma_2 \frac{\pi_0^2}{K^{*2}}n^2,$$

for some positive constant $\gamma_2 \in (0,1)$. $\qquad\square$

**Lemma A.9.** *Under Assumption 1, for any $K^- < K^*$, there exist positive constants $\nu_1, \nu_2 \in (0,1)$ depending on $K^-$, such that*

$$|T_{21}| \geq \nu_1 \frac{\pi_0^2}{K^{*2}}n^2, \tag{18}$$

$$|T_{11}| \geq \nu_2 \frac{\pi_0^2}{K^{*2}}n^2. \tag{19}$$

*Proof of Lemma A.9.* The inequality (18) can be proved in a similar way as the proof of (16) in Lemma A.8. To prove (19), assume the 1st community under $\phi^*$ has $n_1$ nodes. When the network is re-clustered into $K^-$ communities, at least one of the $K^-$ communities must contain at least $\lfloor n_1/K^- \rfloor$ nodes from the 1st community. Therefore,

$$|T_{11}| \geq \binom{\lfloor n_1/K^- \rfloor}{2} \geq \nu_2 \frac{\pi_0^2}{K^{*2}}n^2,$$

for some constant $\nu_2 \in (0,1)$. $\qquad\square$

Note that $T_{11}, T_{12}, T_{21}, T_{22}$ are defined for all pairs of nodes in $\mathcal{N}$. Since the Evaluation step in INCV only consider the pairs of nodes in the test set $\mathcal{N}^{(2)}$, it is convenient to introduce the following notations:

$$
\begin{aligned}
T_{11}^{(2)} &:= \{(i,j) : i,j \in \mathcal{N}^{(2)}, i < j, \hat{\phi}_K(i) = \hat{\phi}_K(j), \phi^*(i) = \phi^*(j)\}, \\
T_{12}^{(2)} &:= \{(i,j) : i,j \in \mathcal{N}^{(2)}, i < j, \hat{\phi}_K(i) \neq \hat{\phi}_K(j), \phi^*(i) = \phi^*(j)\}, \\
T_{21}^{(2)} &:= \{(i,j) : i,j \in \mathcal{N}^{(2)}, i < j, \hat{\phi}_K(i) = \hat{\phi}_K(j), \phi^*(i) \neq \phi^*(j)\}, \\
T_{22}^{(2)} &:= \{(i,j) : i,j \in \mathcal{N}^{(2)}, i < j, \hat{\phi}_K(i) \neq \hat{\phi}_K(j), \phi^*(i) \neq \phi^*(j)\}.
\end{aligned}
\tag{20}
$$

**Lemma A.10.** *Under Assumption 1, for any $K^+ > K^*$, there exist positive constants $\gamma_1, \gamma_2 \in (0,1)$ depending on $K^+$, such that*

$$
|T_{12}^{(2)}| \geq \gamma_1 \frac{\pi_0^2}{K^{*2}} n^2 (1-\tau)^2,
\tag{21}
$$

$$
|T_{22}^{(2)}| \geq \gamma_2 \frac{\pi_0^2}{K^{*2}} n^2 (1-\tau)^2.
\tag{22}
$$

*For any $K^- < K^*$, there exist positive constants $\nu_1, \nu_2 \in (0,1)$ depending on $K^-$, such that*

$$
|T_{21}^{(2)}| \geq \nu_1 \frac{\pi_0^2}{K^{*2}} n^2 (1-\tau)^2,
\tag{23}
$$

$$
|T_{11}^{(2)}| \geq \nu_2 \frac{\pi_0^2}{K^{*2}} n^2 (1-\tau)^2.
\tag{24}
$$

*Proof of Lemma A.10.* As shown in the discussion in Remark 1, by applying the re-classification procedure, the recovered network structure for $\mathcal{N}^{(2)}$ is balanced with respect to $\pi_0$. Since $|\mathcal{N}^{(2)}| = n(1-\tau)$, the bounds in (21)–(24) follow directly from Lemmas A.8 and A.9. $\qquad\square$

**Lemma A.11.** *Under Assumption 1, when the candidate community number $K^+ > K^*$ in INCV, the estimated between-community probability $\hat{q}$ diverges from the true parameter $q$ in the sense that there exists a positive constant $d^+ = \gamma_1 \pi_0^2 (p-q)/(2\gamma_1 \pi_0^2 + 2K^{*2})$ such that*

$$
\Pr(\hat{q} - q > d^+) \geq 1 - \exp\left(-C_{K^+} n^2 (1-\tau)^2 (p-q)^2\right),
\tag{25}
$$

*where $C_{K^+}$ is a positive constant depending on $\pi_0, K^*$, and $K^+$.*

*On the other hand, with a candidate community number $K^- < K^*$ in INCV, the estimated within-community probability $\hat{p}$ diverges from the true parameter $p$ in the sense that there exists a positive constant $d^- = \nu_1 \pi_0^2 (p-q)/(2\nu_1 \pi_0^2 + 2K^{*2})$ such that*

$$
\Pr(\hat{p} - p < -d^-) \geq 1 - \exp\left(-C_{K^-} n^2 (1-\tau)^2 (p-q)^2\right),
\tag{26}
$$

*where $C_{K^-}$ is a positive constant depending on $\pi_0, K^*$, and $K^-$.*

*Proof of Lemma A.11.* When $K^+ > K^*$, the expected value of $\hat{q}$ conditioned on the $\mathcal{E} \backslash \mathcal{E}^{(2)}$ (that

is, removing the validation block) can be written as

$$\mathbb{E}[\hat{q}] = \frac{p|T_{12}^{(2)}| + q|T_{22}^{(2)}|}{|T_{12}^{(2)}| + |T_{22}^{(2)}|} = \eta p + (1 - \eta)q,$$

where $\eta = |T_{12}^{(2)}|/(|T_{12}^{(2)}| + |T_{22}^{(2)}|)$. Next, define

$$\omega := \frac{1}{1 + K^{*2}/\gamma_1 \pi_0^2} > 0.$$

By (21) and the fact that $|T_{22}^{(2)}| \leq n^2(1-\tau)^2$, we have $\eta \geq \omega$, so

$$\begin{aligned}
\Pr\left(\hat{q} - q > \frac{\omega(p-q)}{2}\right) &= \Pr\left(\hat{q} - \mathbb{E}[\hat{q}] + \mathbb{E}[\hat{q}] - q > \frac{\omega(p-q)}{2}\right) \\
&= \Pr\left(\hat{q} - \mathbb{E}[\hat{q}] + \eta(p-q) > \frac{\omega(p-q)}{2}\right) \\
&\geq 1 - \Pr\left(\hat{q} - \mathbb{E}[\hat{q}] \leq -\frac{\omega(p-q)}{2}\right).
\end{aligned} \tag{27}$$

From the Hoeffding inequality argument similar to (10), we have

$$\Pr\left(\hat{q} - \mathbb{E}[\hat{q}] \leq -\frac{\omega(p-q)}{2}\right) \leq \exp\left(-C_{K^+} n^2(1-\tau)^2(p-q)^2\right). \tag{28}$$

where $C_{K^+}$ is a positive constant dependent on $\pi_0$ and $K^+$. Finally combining (27) and (28), we conclude (25). The argument for $\hat{p}$ in (26) when $K^- < K^*$ follows similarly. □

Our last lemma gives a lower and upper bound for the Kullback-Leibler divergence from one Bernoulli random variable to another. This lemma will be used for the proof of Theorem 1 under minimal negative log-likelihood loss function.

**Lemma A.12.** *$P_1$ and $P_2$ are two Bernoulli distributions with the parameter $p_1 \in (0,1)$ and $p_2 \in (0,1)$ respectively. The Kullback-Leibler divergence of $P_2$ from $P_1$ satisfies*

$$\frac{(p_1 - p_2)^2}{(p_1 + p_2)(2 - p_1 - p_2)} \leq D_{KL}(P_1 \| P_2) \leq \frac{(p_1 - p_2)^2}{p_2(1 - p_2)}. \tag{29}$$

*Proof of Lemma A.12.* Recall that the Kullback-Leibler divergence of $P_2$ from $P_1$ has a form

$$D_{KL}(P_1 \| P_2) = p_1 \log \frac{p_1}{p_2} + (1 - p_1) \log \frac{1 - p_1}{1 - p_2}.$$

First consider the case of $p_1 \geq p_2$, where we may write

$$D_{KL}(P_1 \| P_2) = \int_{p_2}^{p_1} \left(\frac{p_1}{x} - \frac{1 - p_1}{1 - x}\right) dx.$$

Since $f(x) = p_1/x - (1 - p_1)/(1 - x)$ is a non-negative continuous decreasing function on $[p_2, p_1]$, then by the Mean Value Theorem, there exists a constant $c \in (p_2, (p_1 + p_2)/2)$ such that

$$D_{KL}(P_1 \| P_2) \geq \int_{p_2}^{\frac{p_1 + p_2}{2}} \left(\frac{p_1}{x} - \frac{1 - p_1}{1 - x}\right) dx$$

$$= \left( \frac{p_1}{c} - \frac{1-p_1}{1-c} \right) \left( \frac{p_1+p_2}{2} - p_2 \right)$$

$$\geq \left( \frac{p_1}{(p_1+p_2)/2} - \frac{1-p_1}{1-(p_1+p_2)/2} \right) \left( \frac{p_1-p_2}{2} \right)$$

$$= \frac{(p_1-p_2)^2}{(p_1+p_2)(2-p_1-p_2)}.$$

Also, since $0 \leq f(x) \leq f(p_2) = (p_1 - p_2)/(p_2(1-p_2))$, we have

$$D_{KL}(P_1\|P_2) = \int_{p_2}^{p_1} \left( \frac{p_1}{x} - \frac{1-p_1}{1-x} \right) dx \leq \frac{(p_1-p_2)^2}{p_2(1-p_2)}.$$

When $p_1 < p_2$, write $q_1 = 1 - p_1$ and $q_2 = 1 - p_2$ so that $q_1 > q_2$. Notice that both the Kullback-Leibler divergence and the lower and upper bound in (29) are invariant between $(p_1, p_2)$ and $(q_1, q_2)$. So (29) holds true for $q_1$ and $q_2$, as well as for $p_1$ and $p_2$.  □

*Proof of Theorem 1 (likelihood objective).* For a candidate $K \in \{2, \ldots, K_{\max}\}$, denote the community labeling function derived from INCV as $\hat{\phi}_K$ and denote the minimized negative log-likelihood function as

$$\hat{\ell}_K = \ell(\hat{p}, \hat{q} \mid \hat{\phi}_K),$$

whose form is given in Section 3.1. In order to prove the consistency, it is equivalent to show that when $K \neq K^*$,

$$\Pr\left( \hat{\ell}_{K^*} - \hat{\ell}_K < 0 \right) \to 1.$$

Because $\hat{p}$ and $\hat{q}$ are the maximum likelihood estimators, it is natural to have

$$\hat{\ell}_{K^*} = \ell(\hat{p}, \hat{q} \mid \hat{\phi}_{K^*}) \leq \ell(p, q \mid \hat{\phi}_{K^*}),$$

where $p$ and $q$ are the true within and between community connection probabilities for the network. For simplicity, denote $\ell(p, q \mid \hat{\phi}_{K^*})$ as $\tilde{\ell}_{K^*}$. Thus it is sufficient to show that when $K \neq K^*$,

$$\Pr\left( \tilde{\ell}_{K^*} - \hat{\ell}_K < 0 \right) \to 1. \tag{30}$$

Recall that under Assumptions 1 and 3, Lemma A.7 states that on the test set $\mathcal{N}^{(2)}$,

$$\Pr(\hat{\phi}_{K^*} \neq \phi^*) \leq 2K^* n(1-\tau) \exp\left( -c_1 n\tau(p-q)^2 \right) + K^* \exp(-c_2 n(1-\tau)) + g(n\tau),$$

where $c_1$ and $c_2$ are positive constants dependent on $K^*$ and $\pi_0$ only. The conditions (i) and (ii) in the theorem statement imply that $\Pr(\hat{\phi}_{K^*} = \phi^*) \to 1$ as $n \to \infty$. Write

$$\Pr(\tilde{\ell}_{K^*} - \hat{\ell}_K < 0) \geq \Pr(\tilde{\ell}_{K^*} - \hat{\ell}_K < 0, \hat{\phi}_{K^*} = \phi^*)$$

$$= \Pr(\tilde{\ell}_{K^*} - \hat{\ell}_K < 0 \mid \hat{\phi}_{K^*} = \phi^*) \Pr(\hat{\phi}_{K^*} = \phi^*).$$

Thus, in order to show (30), it is sufficient to prove that when $K \neq K^*$,

$$\Pr(\hat{\ell}_K - \tilde{\ell}_{K^*} > 0 \mid \hat{\phi}_{K^*} = \phi^*) \to 1.$$

Recall the definitions of $T_{11}^{(2)}$, $T_{12}^{(2)}$, $T_{21}^{(2)}$, and $T_{22}^{(2)}$ in (20). The minimized negative log-likelihood $\hat{\ell}_K = \ell(\hat{p}, \hat{q} \mid \hat{\phi}_K)$ can be rewritten as

$$
\begin{aligned}
\hat{\ell}_K = -&\sum_{(i,j)\in T_{11}^{(2)}} \left[ A_{ij}^{(22)} \log \hat{p} + (1 - A_{ij}^{(22)}) \log(1 - \hat{p}) \right] \\
-&\sum_{(i,j)\in T_{12}^{(2)}} \left[ A_{ij}^{(22)} \log \hat{q} + (1 - A_{ij}^{(22)}) \log(1 - \hat{q}) \right] \\
-&\sum_{(i,j)\in T_{21}^{(2)}} \left[ A_{ij}^{(22)} \log \hat{p} + (1 - A_{ij}^{(22)}) \log(1 - \hat{p}) \right] \\
-&\sum_{(i,j)\in T_{22}^{(2)}} \left[ A_{ij}^{(22)} \log \hat{q} + (1 - A_{ij}^{(22)}) \log(1 - \hat{q}) \right].
\end{aligned}
$$

Similarly, when $\hat{\phi}_{K^*} = \phi^*$, $\tilde{\ell}_{K^*} = \ell(p, q \mid \phi^*)$ can be rewritten as

$$
\begin{aligned}
\tilde{\ell}_{K^*} = -&\sum_{(i,j)\in T_{11}^{(2)}} \left[ A_{ij}^{(22)} \log p + (1 - A_{ij}^{(22)}) \log(1 - p) \right] \\
-&\sum_{(i,j)\in T_{12}^{(2)}} \left[ A_{ij}^{(22)} \log p + (1 - A_{ij}^{(22)}) \log(1 - p) \right] \\
-&\sum_{(i,j)\in T_{21}^{(2)}} \left[ A_{ij}^{(22)} \log q + (1 - A_{ij}^{(22)}) \log(1 - q) \right] \\
-&\sum_{(i,j)\in T_{22}^{(2)}} \left[ A_{ij}^{(22)} \log q + (1 - A_{ij}^{(22)}) \log(1 - q) \right].
\end{aligned}
$$

Furthermore, define

$$
\begin{aligned}
\tilde{\xi}_{K^*} = &\sum_{(i,j)\in T_{11}^{(2)}} \left[ p \log p + (1 - p) \log(1 - p) \right] + \sum_{(i,j)\in T_{12}^{(2)}} \left[ p \log p + (1 - p) \log(1 - p) \right] \\
+ &\sum_{(i,j)\in T_{21}^{(2)}} \left[ q \log q + (1 - q) \log(1 - q) \right] + \sum_{(i,j)\in T_{22}^{(2)}} \left[ q \log q + (1 - q) \log(1 - q) \right],
\end{aligned}
$$

and for $K \neq K^*$, define

$$
\begin{aligned}
\hat{\xi}_K = &\sum_{(i,j)\in T_{11}^{(2)}} \left[ p \log \hat{p} + (1 - p) \log(1 - \hat{p}) \right] + \sum_{(i,j)\in T_{12}^{(2)}} \left[ p \log \hat{q} + (1 - p) \log(1 - \hat{q}) \right] \\
+ &\sum_{(i,j)\in T_{21}^{(2)}} \left[ q \log \hat{p} + (1 - q) \log(1 - \hat{p}) \right] + \sum_{(i,j)\in T_{22}^{(2)}} \left[ q \log \hat{q} + (1 - q) \log(1 - \hat{q}) \right].
\end{aligned}
$$

Then we can write

$$
\hat{\ell}_K - \tilde{\ell}_{K^*} = \underbrace{(\tilde{\ell}_{K^*} - \tilde{\xi}_{K^*}) + (\hat{\xi}_K - \hat{\ell}_K)}_{I} + \underbrace{\tilde{\xi}_{K^*} - \hat{\xi}_K}_{II}. \tag{31}
$$

For $I$, we have

$$
I = \tilde{\ell}_{K^*} - \tilde{\xi}_{K^*} + \hat{\xi}_K - \hat{\ell}_K
$$

$$= \sum_{(i,j) \in T_{11}^{(2)}} \left( A_{ij}^{(22)} - p \right) \log \frac{p(1-\hat{p})}{(1-p)\hat{p}} + \sum_{(i,j) \in T_{12}^{(2)}} \left( A_{ij}^{(22)} - p \right) \log \frac{p(1-\hat{q})}{(1-p)\hat{q}}$$

$$+ \sum_{(i,j) \in T_{21}^{(2)}} \left( A_{ij}^{(22)} - q \right) \log \frac{q(1-\hat{p})}{(1-q)\hat{p}} + \sum_{(i,j) \in T_{22}^{(2)}} \left( A_{ij}^{(22)} - q \right) \log \frac{q(1-\hat{q})}{(1-q)\hat{q}}. \tag{32}$$

By Lemma A.6, there exists a constant

$$M_{11,n} = 2 \left| \log \frac{\delta_n}{2} \right| > \log \frac{(1-\delta_n)(1-\delta_n/2)}{\delta_n^2/2} > 0$$

depending on $\delta_n$ in Assumption 2 such that

$$\Pr \left( \left| \log \frac{p(1-\hat{p})}{(1-p)\hat{p}} \right| \geq M_{11,n} \right) \leq 2 \exp \left( -C_p n^2 (1-\tau)^2 \delta_n^2 \right).$$

Therefore, using Hoeffding's inequality, we obtain

$$\Pr \left( \left| \log \frac{p(1-\hat{p})}{(1-p)\hat{p}} \sum_{(i,j) \in T_{11}^{(2)}} \left( A_{ij}^{(22)} - p \right) \right| > 2n(1-\tau)\log(n(1-\tau)) \left| \log \frac{\delta_n}{2} \right| \right)$$

$$\leq \Pr \left( \left| \sum_{(i,j) \in T_{11}^{(2)}} \left( A_{ij}^{(22)} - p \right) \right| > n(1-\tau)\log(n(1-\tau)) \right) + \Pr \left( \left| \log \frac{p(1-\hat{p})}{(1-p)\hat{p}} \right| \geq M_{11,n} \right)$$

$$\leq \sum_{h=0}^{n^2(1-\tau)^2} \Pr \left( \left| \sum_{(i,j) \in T_{11}^{(2)}} \left( A_{ij}^{(22)} - p \right) \right| > n(1-\tau)\log(n(1-\tau)) \left| |T_{11}^{(2)}| = h \right. \right) \Pr \left( |T_{11}^{(2)}| = h \right)$$

$$+ 2 \exp \left( -C_p n^2 (1-\tau)^2 \delta_n^2 \right)$$

$$\leq 2 \sum_{h=0}^{n^2(1-\tau)^2} \exp \left( -\frac{2n^2(1-\tau)^2 \log^2 (n(1-\tau))}{h} \right) \Pr \left( |T_{11}^{(2)}| = h \right) + 2 \exp \left( -C_p n^2 (1-\tau)^2 \delta_n^2 \right)$$

$$\leq 2 \exp \left( -\frac{2n^2(1-\tau)^2 \log^2 (n(1-\tau))}{n^2(1-\tau)^2} \right) \sum_{h=0}^{n^2(1-\tau)^2} \Pr \left( |T_{11}^{(2)}| = h \right) + 2 \exp \left( -C_p n^2 (1-\tau)^2 \delta_n^2 \right)$$

$$\leq 2 \exp \left( -2 \log^2 (n(1-\tau)) \right) + 2 \exp \left( -C_p n^2 (1-\tau)^2 \delta_n^2 \right). \tag{33}$$

Similar inequalities hold for $I_{12}, I_{21}, I_{22}$. Therefore, under condition (ii),

$$|I| = \left| \tilde{\ell}_{K^*} - \tilde{\xi}_{K^*} + \hat{\xi}_K - \hat{\ell}_K \right| = O_p \left( n(1-\tau)\log(n(1-\tau)) \left| \log \frac{\delta_n}{2} \right| \right). \tag{34}$$

Next, we expand the term $II$ in (31) as

$$II = \tilde{\xi}_{K^*} - \hat{\xi}_K$$

$$= \left| T_{11}^{(2)} \right| p \log \frac{p}{\hat{p}} + \left| T_{11}^{(2)} \right| (1-p) \log \frac{(1-p)}{(1-\hat{p})} + \left| T_{12}^{(2)} \right| p \log \frac{p}{\hat{q}} + \left| T_{12}^{(2)} \right| (1-p) \log \frac{(1-p)}{(1-\hat{q})}$$

$$+ \left| T_{21}^{(2)} \right| q \log \frac{q}{\hat{p}} + \left| T_{21}^{(2)} \right| (1-q) \log \frac{(1-q)}{(1-\hat{p})} + \left| T_{22}^{(2)} \right| q \log \frac{q}{\hat{q}} + \left| T_{22}^{(2)} \right| (1-q) \log \frac{(1-q)}{(1-\hat{q})}$$

$$= \left| T_{11}^{(2)} \right| D_{KL}(p \parallel \hat{p}) + \left| T_{12}^{(2)} \right| D_{KL}(p \parallel \hat{q}) + \left| T_{21}^{(2)} \right| D_{KL}(q \parallel \hat{p}) + \left| T_{22}^{(2)} \right| D_{KL}(q \parallel \hat{q}). \tag{35}$$

By definition, all four KL divergences are positive. When $K > K^*$, consider only $\left|T_{22}^{(2)}\right| \cdot D_{KL}(q \parallel \hat{q})$. By Lemma A.12,

$$D_{KL}(q \parallel \hat{q}) \geq \frac{(q - \hat{q})^2}{(q + \hat{q})(2 - q - \hat{q})} \geq (q - \hat{q})^2.$$

From Lemma A.10, when $K > K^*$,

$$\left|T_{22}^{(2)}\right| \geq \frac{\gamma_2 \pi_0^2}{K^{*2}} n^2 (1 - \tau)^2$$

for some positive constant $\gamma_2 \in (0, 1)$. In addition when $K > K^*$, by Lemma A.11,

$$\Pr\left(\hat{q} - q > d_+\right) \geq 1 - \exp\left(-C_K n^2 (1 - \tau)^2 (p - q)^2\right),$$

where $d^+ = \gamma_1 \pi_0^2 (p - q)/(2\gamma_1 \pi_0^2 + 2K^{*2})$ and $C_{K+}$ is a positive constant depending on $\pi_0, K^*$ and $K^+$ only. Thus,

$$\Pr\left(\left|T_{22}^{(2)}\right| D_{KL}(q \parallel \hat{q}) \geq n^2 (1 - \tau)^2 \frac{\gamma_2 \pi_0^2}{K^{*2}} d^{+2}\right)$$
$$\geq \Pr\left((q - \hat{q})^2 \geq d^{+2}\right) \geq \Pr\left(\hat{q} - q \geq d^+\right)$$
$$\geq 1 - \exp\left(-C_{K+} n^2 (1 - \tau)^2 (p - q)^2\right). \tag{36}$$

Similarly, when $K < K^*$, by (24) and (26), we obtain

$$\Pr\left(\left|T_{11}^{(2)}\right| D_{KL}(p \parallel \hat{p}) \geq n^2 (1 - \tau)^2 \frac{\nu_2 \pi_0^2}{K^{*2}} d^{-2}\right)$$
$$\geq \Pr\left((p - \hat{p})^2 \geq d^{-2}\right) \geq \Pr\left(\hat{p} - p \leq -d^-\right)$$
$$\geq 1 - \exp\left(-C_{K-} n^2 (1 - \tau)^2 (p - q)^2\right). \tag{37}$$

Combining (35), (36) and (37), we have shown that for any $K \neq K^*$, under condition (ii) in the theorem statement, there exists a constant $L > 0$ dependent on $\pi_0, K^*$, and $K$ such that

$$\Pr\left(II = \tilde{\xi}_{K^*} - \hat{\xi}_K \geq L n^2 (1 - \tau)^2 (p - q)^2\right) \geq 1 - \exp\left(-C' n^2 (1 - \tau)^2 (p - q)^2\right). \tag{38}$$

Finally, putting (34) and (38) together, under condition (ii) which also indicates $(1 - \tau)(p - q) = \omega(1/n)$, we conclude when $K \neq K^*$,

$$\Pr\left(\hat{\ell}_K - \tilde{\ell}_{K^*} > 0 \mid \phi_{K^*} = \phi^*\right) \to 1 \quad \text{as } n \to \infty.$$

This gives the desired result. $\qquad\square$

*Proof of Theorem 1 (MSE objective).* For a candidate $K \in \{2, \ldots, K_{\max}\}$, recall the community labeling function derived from INCV is denoted as $\hat{\phi}_K$, and by slightly reusing the notation before without causing confusion, we denote the minimized MSE loss as

$$\hat{\ell}_K = \mathrm{MSE}(\hat{p}, \hat{q} \mid \hat{\phi}_K),$$

whose form is given in Section 2.2 and 3.1. Similarly, in order to prove the consistency, it is equivalent to show that when $K \neq K^*$,

$$\Pr\left(\hat{\ell}_{K^*} - \hat{\ell}_K < 0\right) \to 1.$$

Notice that if we denote generally

$$\mathrm{MSE}(a, b \mid \hat{\phi}_K) = \sum_{(i,j) \in W^{(2)}_{\hat{\phi}_K}} (A^{(22)}_{ij} - a)^2 + \sum_{(i,j) \in B^{(2)}_{\hat{\phi}_K}} (A^{(22)}_{ij} - b)^2,$$

then by the definition of $\hat{p}$ and $\hat{q}$,

$$\hat{\ell}_{K^*} = \mathrm{MSE}(\hat{p}, \hat{q} \mid \hat{\phi}_{K^*}) \leq \mathrm{MSE}(p, q \mid \hat{\phi}_{K^*}),$$

where $p$ and $q$ are the true within and between community connection probabilities for the network. For simplicity, denote $\mathrm{MSE}(p, q \mid \hat{\phi}_{K^*})$ as $\tilde{\ell}_{K^*}$. Thus it is sufficient to show that when $K \neq K^*$,

$$\Pr\left(\tilde{\ell}_{K^*} - \hat{\ell}_K < 0\right) \to 1.$$

Still, by Lemma A.7 and conditions (i), (ii$'$), we have $\Pr\left(\hat{\phi}_{K^*} = \phi^*\right) \to 1$ as $n \to \infty$. Similarly as before, it suffices to prove that when $K \neq K^*$,

$$\Pr(\tilde{\ell}_{K^*} - \hat{\ell}_K < 0 \mid \hat{\phi}_{K^*} = \phi^*) \to 1.$$

Here, the MSE losses can be written as

$$
\begin{aligned}
\hat{\ell}_K &= \sum_{(i,j) \in T^{(2)}_{11}} (A^{(22)}_{ij} - \hat{p})^2 + \sum_{(i,j) \in T^{(2)}_{12}} (A^{(22)}_{ij} - \hat{q})^2 + \sum_{(i,j) \in T^{(2)}_{21}} (A^{(22)}_{ij} - \hat{p})^2 + \sum_{(i,j) \in T^{(2)}_{22}} (A^{(22)}_{ij} - \hat{q})^2 \\
&= \sum_{(i,j) \in \mathcal{E}^{(2)}} \left(A^{(22)}_{ij}\right)^2 - 2\left(\sum_{(i,j) \in T^{(2)}_{11}} \hat{p}A^{(22)}_{ij} + \sum_{(i,j) \in T^{(2)}_{12}} \hat{q}A^{(22)}_{ij} + \sum_{(i,j) \in T^{(2)}_{21}} \hat{p}A^{(22)}_{ij} + \sum_{(i,j) \in T^{(2)}_{22}} \hat{q}A^{(22)}_{ij}\right) \\
&\quad + \left(\sum_{(i,j) \in T^{(2)}_{11}} \hat{p}^2 + \sum_{(i,j) \in T^{(2)}_{12}} \hat{q}^2 + \sum_{(i,j) \in T^{(2)}_{21}} \hat{p}^2 + \sum_{(i,j) \in T^{(2)}_{22}} \hat{q}^2\right) \\
&:= \sum_{(i,j) \in \mathcal{E}^{(2)}} \left(A^{(22)}_{ij}\right)^2 + \hat{\varphi}_K,
\end{aligned}
$$

and

$$
\begin{aligned}
\tilde{\ell}_{K^*} &= \sum_{(i,j) \in T^{(2)}_{11}} (A^{(22)}_{ij} - p)^2 + \sum_{(i,j) \in T^{(2)}_{12}} (A^{(22)}_{ij} - p)^2 + \sum_{(i,j) \in T^{(2)}_{21}} (A^{(22)}_{ij} - q)^2 + \sum_{(i,j) \in T^{(2)}_{22}} (A^{(22)}_{ij} - q)^2 \\
&= \sum_{(i,j) \in \mathcal{E}^{(2)}} \left(A^{(22)}_{ij}\right)^2 - 2\left(\sum_{(i,j) \in T^{(2)}_{11}} pA^{(22)}_{ij} + \sum_{(i,j) \in T^{(2)}_{12}} pA^{(22)}_{ij} + \sum_{(i,j) \in T^{(2)}_{21}} qA^{(22)}_{ij} + \sum_{(i,j) \in T^{(2)}_{22}} qA^{(22)}_{ij}\right)
\end{aligned}
$$

$$+ \left( \sum_{(i,j) \in T_{11}^{(2)}} p^2 + \sum_{(i,j) \in T_{12}^{(2)}} p^2 + \sum_{(i,j) \in T_{21}^{(2)}} q^2 + \sum_{(i,j) \in T_{22}^{(2)}} q^2 \right)$$

$$:= \sum_{(i,j) \in \mathcal{E}^{(2)}} \left( A_{ij}^{(22)} \right)^2 + \tilde{\varphi}_{K^*}.$$

Thus, we just need to prove that when $K \neq K^*$,

$$\Pr(\tilde{\varphi}_{K^*} - \hat{\varphi}_K < 0 \mid \hat{\phi}_{K^*} = \phi^*) \to 1.$$

Furthermore define

$$\tilde{\xi}_{K^*} = - \left( \sum_{(i,j) \in T_{11}^{(2)}} p^2 + \sum_{(i,j) \in T_{12}^{(2)}} p^2 + \sum_{(i,j) \in T_{21}^{(2)}} q^2 + \sum_{(i,j) \in T_{22}^{(2)}} q^2 \right),$$

and for $K \neq K^*$ define

$$\hat{\xi}_K = - 2 \left( \sum_{(i,j) \in T_{11}^{(2)}} \hat{p}p + \sum_{(i,j) \in T_{12}^{(2)}} \hat{q}p + \sum_{(i,j) \in T_{21}^{(2)}} \hat{p}q + \sum_{(i,j) \in T_{22}^{(2)}} \hat{q}q \right)$$

$$+ \left( \sum_{(i,j) \in T_{11}^{(2)}} \hat{p}^2 + \sum_{(i,j) \in T_{12}^{(2)}} \hat{q}^2 + \sum_{(i,j) \in T_{21}^{(2)}} \hat{p}^2 + \sum_{(i,j) \in T_{22}^{(2)}} \hat{q}^2 \right).$$

Then we can write

$$\hat{\varphi}_K - \tilde{\varphi}_{K^*} = \underbrace{(\hat{\varphi}_K - \hat{\xi}_K) + (\tilde{\xi}_{K^*} - \tilde{\varphi}_{K^*})}_{I} + \underbrace{(\hat{\xi}_K - \tilde{\xi}_{K^*})}_{II}. \tag{39}$$

For $I$, we have

$$I = 2 \left( \sum_{(i,j) \in T_{11}^{(2)}} (p - \hat{p}) \left( A_{ij}^{(22)} - p \right) + \sum_{(i,j) \in T_{12}^{(2)}} (p - \hat{q}) \left( A_{ij}^{(22)} - p \right) \right.$$

$$\left. + \sum_{(i,j) \in T_{21}^{(2)}} (q - \hat{p}) \left( A_{ij}^{(22)} - q \right) + \sum_{(i,j) \in T_{22}^{(2)}} (q - \hat{q}) \left( A_{ij}^{(22)} - q \right) \right)$$

$$:= 2 \left( I_{11} + I_{12} + I_{21} + I_{22} \right).$$

Notice that $|p - \hat{p}| \leq 2$ always holds. Thus, for $I_{11}$, by Hoeffding's inequality and a similar analysis as (33), we have

$$\Pr \left( \left| (p - \hat{p}) \sum_{(i,j) \in T_{11}^{(2)}} \left( A_{ij}^{(22)} - p \right) \right| > 2n(1 - \tau) \log(n(1 - \tau)) \right)$$

$$\leq \Pr\left(\left|\sum_{(i,j)\in T_{11}^{(2)}}\left(A_{ij}^{(22)}-p\right)\right| > n(1-\tau)\log(n(1-\tau))\right)$$

$$\leq \sum_{h=0}^{n^2(1-\tau)^2} \Pr\left(\left|\sum_{(i,j)\in T_{11}^{(2)}}\left(A_{ij}^{(22)}-p\right)\right| > n(1-\tau)\log(n(1-\tau))\,\middle|\,|T_{11}^{(2)}|=h\right)\Pr\left(|T_{11}^{(2)}|=h\right)$$

$$\leq 2\sum_{h=0}^{n^2(1-\tau)^2} \exp\left(-\frac{2n^2(1-\tau)^2\log^2\left(n(1-\tau)\right)}{h}\right)\Pr\left(|T_{11}^{(2)}|=h\right)$$

$$\leq 2\exp\left(-2\log^2\left(n(1-\tau)\right)\right). \tag{40}$$

Similar inequalities hold for $I_{12}, I_{21}, I_{22}$. Therefore, under condition (ii′),

$$|I| = \left|\hat\varphi_K - \hat\xi_K + \tilde\xi_{K^*} - \tilde\varphi_{K^*}\right| = O_p\left(n(1-\tau)\log(n(1-\tau))\right). \tag{41}$$

Next we write the term $II$ in (39) as

$$II = \sum_{(i,j)\in T_{11}^{(2)}}(\hat p - p)^2 + \sum_{(i,j)\in T_{12}^{(2)}}(\hat q - p)^2 + \sum_{(i,j)\in T_{21}^{(2)}}(\hat p - q)^2 + \sum_{(i,j)\in T_{22}^{(2)}}(\hat q - q)^2. \tag{42}$$

Each of the four terms here are positive obviously, and has exactly the form of the lower bound for KL divergence we used in the likelihood objective case. Thus, using an exact argument as in (36), (37) and (38), we have

$$\Pr\left(II = \hat\xi_K - \tilde\xi_{K^*} \geq Ln^2(1-\tau)^2(p-q)^2\right) \geq 1 - \exp\left(-C'n^2(1-\tau)^2(p-q)^2\right). \tag{43}$$

Putting (41) and (43) together, using condition (ii′) which also indicates $(1-\tau)(p-q) = \omega(1/n)$, we reach the conclusion. $\qquad\square$

## A.3 Proof of Corollary 1

We first modify several lemmas used before to give a more precise tail probability bound in this specific regime.

**Lemma A.13** (Modification of Lemma A.6). *Under Assumption 1, the maximum likelihood estimate $\hat p$ and $\hat q$ as defined in Section 3.1 for any candidate $K \geq 2$ are of order $\rho_n$ in the sense that*

$$\Pr\left(\frac{\rho_n q_0}{2} < \hat p < 2\rho_n p_0\right) \geq 1 - \exp\left(-C_p n^2(1-\tau)^2\rho_n\right), \tag{44}$$

$$\Pr\left(\frac{\rho_n q_0}{2} < \hat q < 2\rho_n q_0\right) \geq 1 - \exp\left(-C_p n^2(1-\tau)^2\rho_n\right), \tag{45}$$

*where $\tau$ is the INCV node splitting ratio, $\rho_n$ is the sparsity coefficient, $C_p$ and $C_q$ are two positive constants only dependent on $K$ and $\pi_0$ (from Assumption 1).*

*Proof of Lemma A.13.* First consider $\hat{q}$. Recall that

$$\hat{q} = \frac{\sum_{(i,j)\in B^{(2)}_{\hat{\phi}_K}} A^{(22)}_{ij}}{|B^{(2)}_{\hat{\phi}_K}|}.$$

Because every $A^{(22)}_{ij}$ has an expected value equal to $p$ or $q$, it is obvious by Assumption 2 that

$$\rho_n q_0 = q \leq \mathbb{E}(\hat{q}) \leq p = \rho_n p_0. \tag{46}$$

Still, let $n^{(2)}_k$ denote the number of nodes in $\mathcal{N}^{(2)}$ being classified to the $k$-th community for $k = 1, 2, \ldots, K$. We have exactly the same as in Lemma A.6 that

$$\frac{n^2(1-\tau)^2(K-1)\pi_0(1-\pi_0)}{2K} \leq |B^{(2)}_{\hat{\phi}_K}| \leq \frac{(K-1)n^2(1-\tau)^2}{2K}.$$

Next using the Law of Total Probability and Bernstein's Inequality, we have

$$\Pr\left(|\hat{q} - \mathbb{E}(\hat{q})| \geq \frac{\rho_n q_0}{2}\right) = \sum_{h=\frac{n^2(1-\tau)^2(K-1)\pi_0(1-\pi_0)}{2K}}^{\frac{(K-1)n^2(1-\tau)^2}{2K}} \Pr\left(|\hat{q} - \mathbb{E}(\hat{q})| \geq \frac{\rho_n q_0}{2} \mid |B^{(2)}_{\hat{\phi}_K}| = h\right) \Pr\left(|B^{(2)}_{\hat{\phi}_K}| = h\right)$$

$$\leq \sum_{h=\frac{n^2(1-\tau)^2(K-1)\pi_0(1-\pi_0)}{2K}}^{\frac{(K-1)n^2(1-\tau)^2}{2K}} 2\exp\left(-\frac{\rho_n^2 q_0^2 h^2}{2\left(h\rho_n p_0 + \rho_n q_0 h/6\right)}\right) \Pr\left(|B^{(2)}_{\hat{\phi}_K}| = h\right)$$

$$\leq 2\exp\left(-\frac{n^2(1-\tau)^2(K-1)\pi_0(1-\pi_0)\rho_n q_0^2}{4K(p_0 + q_0/6)}\right)$$

$$\leq 2\exp\left(-C_q n^2(1-\tau)^2\rho_n\right). \tag{47}$$

Equations (46) and (47) together imply

$$\Pr\left(\frac{\rho_n q_0}{2} < \hat{p} < 2\rho_n p_0\right) \geq 1 - \exp\left(-C_p n^2(1-\tau)^2\rho_n\right).$$

The probability inequality (44) for $\hat{p}$ can be proved in a similar manner. $\qquad\square$

**Lemma A.14** (Modification of Lemma A.7). *Suppose an undirected network $(\mathcal{N}, \mathcal{E})$ generated from a stochastic block model satisfied Assumptions 1,3, and mechanism for $p$, $q$ in Corollary 1. When $K = K^*$, the derived community labeling function $\hat{\phi}_{K^*}$ on $\mathcal{N}^{(2)}$ in the Classification step of INCV satisfies*

$$\Pr(\hat{\phi}_{K^*} \neq \phi^*) \leq 2K^* n(1-\tau)\exp(-c_1 n\tau\rho_n) + K^*\exp(-c_2 n(1-\tau)) + g(n\tau),$$

*where $\tau$ is the node splitting ratio, $g(x)$ is the function defined in Assumption 3, and $c_1$ and $c_2$ are positive constants dependent on $\pi_0$ (from Assumption 1), $K^*$, $p_0$ and $q_0$ only.*

*Proof of Lemma A.14.* By Assumption 1, there exists a positive constant $\pi_0 \in (0,1)$ such that

$$n_{\min} = \min\{n_1, n_2, \ldots, n_{K^*}\} \geq \frac{n\pi_0}{K^*}.$$

Let the notation $n_k^{(1)}$ denote the cardinality of a subset of community $k$ that belongs to the training set $\mathcal{N}^{(1)}$. Thus, each $n_k^{(1)}$ follows a hypergeometric distribution with parameter $(\tau n, n_k, n)$, and (11) still holds, which says

$$\Pr\left(n_k^{(1)} \leq \frac{n\pi_0\tau}{2K^*}\right) \leq \exp(-c_0 n\tau)$$

for a positive constant $c_0$ dependent on $K^*$ and $\pi_0$ only.

Next we consider the case when $\hat{\phi}_{K^*} = \phi^*$ on $\mathcal{N}^{(1)}$. In this case, for any node $j \in \mathcal{N}^{(2)}$, let $l_j = \phi^*(j)$ denote its true community. Then according to the classification criterion, we have exactly the same

$$\Pr\left(\hat{\phi}_{K^*}(j) \neq \phi^*(j)\right) \leq \sum_{\substack{l=1 \\ l \neq l_j}}^{K^*} \Pr\left(\frac{\sum_{i \in \mathcal{N}^{(1)}:\phi^*(i)=l_j} A_{ij}^{(12)}}{n_{l_j}^{(1)}} < \frac{\sum_{i \in \mathcal{N}^{(1)}:\phi^*(i)=l} A_{ij}^{(12)}}{n_l^{(1)}}\right)$$

$$\leq \sum_{\substack{l=1 \\ l \neq l_j}}^{K^*} \left[ \Pr\left(\frac{\sum_{i \in \mathcal{N}^{(1)}:\phi^*(i)=l_j} A_{ij}^{(12)}}{n_{l_j}^{(1)}} - p < -\frac{p-q}{2}\right)\right.$$

$$\left. + \Pr\left(\frac{\sum_{i \in \mathcal{N}^{(1)}:\phi^*(i)=l} A_{ij}^{(12)}}{n_l^{(1)}} - q > \frac{p-q}{2}\right) \right]. \tag{48}$$

Applying the Law of Total Probability and Bernstein's Inequality, we obtain

$$\Pr\left(\frac{\sum_{i \in \mathcal{N}^{(1)}:\phi^*(i)=l_j} A_{ij}^{(12)}}{n_{l_j}^{(1)}} - p < -\frac{p-q}{2}\right)$$

$$\leq \sum_{h=0}^{n\tau} \Pr\left(\frac{\sum_{i \in \mathcal{N}^{(1)}:\phi^*(i)=l_j} A_{ij}^{(12)}}{n_{l_j}^{(1)}} - \rho_n p_0 < -\frac{\rho_n(p_0-q_0)}{2} \,\bigg|\, n_{l_j}^{(1)} = h\right) \Pr(n_{l_j}^{(1)} = h)$$

$$\leq \sum_{h=\frac{n\pi_0\tau}{2K^*}}^{n\tau} \exp\left(-\frac{\rho_n^2(p_0-q_0)^2 h^2}{2\left(h\rho_n p_0 + \rho_n(p_0-q_0)h/6\right)}\right) \Pr(n_{l_j}^{(1)} = h) + \Pr\left(n_{l_j}^{(1)} \leq \frac{n\pi_0\tau}{2K^*}\right)$$

$$\leq \exp\left(-\frac{n\tau\pi_0(p_0-q_0)^2\rho_n}{4K^*(7p_0/6 - q_0/6)}\right) + \exp(-c_0 n\tau)$$

$$\leq \exp(-c_1 n\tau\rho_n).$$

Similarly, we can prove

$$\Pr\left(\frac{\sum_{i \in \mathcal{N}^{(1)}:\phi^*(i)=l} A_{ij}^{(12)}}{n_l^{(1)}} - q > \frac{p-q}{2}\right) \leq \exp(-c_1 n\tau\rho_n).$$

Therefore, (48) can be rewritten as

$$\Pr\left(\hat{\phi}_{K^*}(j) \neq \phi^*(j)\right) \leq \sum_{\substack{l=1 \\ l \neq l_j}}^{K^*} 2\exp(-c_1 n\tau\rho_n)$$

$$= 2K^*\exp(-c_1 n\tau\rho_n).$$

It is also known by Assumption 3 that the probability of $\hat{\phi}_{K^*} \neq \phi^*$ on $\mathcal{N}^{(1)}$ is bounded by $g(n\tau)$. Thus on $\mathcal{N}^{(2)}$, the $\hat{\phi}_{K^*}$ obtained according to the classification step shall satisfy

$$\begin{aligned}
\Pr(\hat{\phi}_{K^*} \neq \phi^*) &\leq \sum_{j \in \mathcal{N}^{(2)}} \Pr\left(\hat{\phi}_{K^*}(j) \neq \phi^*(j)\right) + g(n\tau) \\
&\leq n(1-\tau)2K^* \exp(-c_1 n\tau\rho_n) + g(n\tau) \\
&\leq 2K^* n(1-\tau) \exp(-c_1 n\tau\rho_n) + g(n\tau).
\end{aligned} \tag{49}$$

Finally, combining (49) and (14), when $K = K^*$, the derived labeling function $\hat{\phi}_{K^*}$ on $\mathcal{N}^{(2)}$ even with the re-classification shall satisfy

$$\Pr(\hat{\phi}_{K^*} \neq \phi^*) \leq 2K^* n(1-\tau) \exp(-c_1 n\tau\rho_n) + K^* \exp(-c_2 n(1-\tau)) + g(n\tau).$$

$\square$

**Lemma A.15** (Modification of Lemma A.11). *Under Assumption 1, when the candidate community number $K^+ > K^*$ in INCV, the estimated between-community probability $\hat{q}$ diverges from the true parameter $q$ in the sense that there exists a positive constant $d^+ = \gamma_1 \pi_0^2 (p-q)/(2\gamma_1 \pi_0^2 + 2K^{*2})$ such that*

$$\Pr(\hat{q} - q > d^+) \geq 1 - \exp\left(-C_{K^+} n^2 (1-\tau)^2 \rho_n\right), \tag{50}$$

*where $C_{K^+}$ is a positive constant depending on $\pi_0, K^*, K^+, p_0$ and $q_0$.*

*On the other hand, with a candidate community number $K^- < K^*$ in INCV, the estimated within-community probability $\hat{p}$ diverges from the true parameter $p$ in the sense that there exists a positive constant $d^- = \nu_1 \pi_0^2 (p-q)/(2\nu_1 \pi_0^2 + 2K^{*2})$ such that*

$$\Pr(\hat{p} - p < -d^-) \geq 1 - \exp\left(-C_{K^-} n^2 (1-\tau)^2 \rho_n\right), \tag{51}$$

*where $C_{K^-}$ is a positive constant depending on $\pi_0, K^*, K^-, p_0$ and $q_0$.*

*Proof of Lemma A.15.* When $K^+ > K^*$, we can still write

$$\mathbb{E}[\hat{q}] = \frac{p|T_{12}^{(2)}| + q|T_{22}^{(2)}|}{|T_{12}^{(2)}| + |T_{22}^{(2)}|} = \eta p + (1-\eta)q,$$

where $\eta = |T_{12}^{(2)}|/(|T_{12}^{(2)}| + |T_{22}^{(2)}|)$. We also define

$$\omega := \frac{1}{1 + K^{*2}/\gamma_1 \pi_0^2} > 0.$$

By (21) and the fact that $|T_{22}^{(2)}| \leq n^2(1-\tau)^2$, we have $\eta \geq \omega$, so (27) still holds, which says

$$\Pr\left(\hat{q} - q > \frac{\omega(p-q)}{2}\right) \geq 1 - \Pr\left(\hat{q} - \mathbb{E}[\hat{q}] \leq -\frac{\omega(p-q)}{2}\right).$$

From the Bernstein's inequality argument similar to (47), we have

$$\Pr\left(\hat{q} - \mathbb{E}[\hat{q}] \leq -\frac{\omega(p-q)}{2}\right) \leq \exp\left(-C_{K^+}n^2(1-\tau)^2\rho_n\right). \tag{52}$$

where $C_{K^+}$ is a positive constant dependent on $\pi_0$ and $K^+$. Finally combining (27) and (52), we conclude (50). The argument for $\hat{p}$ in (51) when $K^- < K^*$ follows similarly. $\square$

*Proof of Corollary 1 (likelihood objective).* The proof of this corollary follows exactly the same procedure as in the proof of Theorem 1, except a few tail probability bounds changed. Thus, we just briefly present this proof, while emphasizing the replacement on those modified tail probability bounds.

We use the same notation as in the proof of Theorem 1. Lemma A.14 states that on $\mathcal{N}^{(2)}$,

$$\Pr(\hat{\phi}_{K^*} \neq \phi^*) \leq 2K^*n(1-\tau)\exp(-c_1 n\tau\rho_n) + K^*\exp(-c_2 n(1-\tau)) + g(n\tau).$$

Thus condition (i) and (ii) imply $\Pr(\hat{\phi}_{K^*} = \phi^*) \to 1$ as $n \to \infty$. Still we just need to prove

$$\Pr\left(\hat{\ell}_K - \tilde{\ell}_{K^*} > 0 \mid \phi_{K^*} = \phi^*\right) \to 1 \quad \text{as } n \to \infty.$$

Use the same expansion for $\hat{\ell}_K$, $\tilde{\ell}_{K^*}$ and adopt the same definition for $\hat{\xi}_K$, $\tilde{\xi}_{K^*}$, as well as the terms $I, II, I_{11}, I_{12}, I_{21}$ and $I_{22}$.

By Lemma A.13, there exists a constant

$$M_{11} = 2\log\frac{2p_0}{q_0} > \log\frac{(\rho_n p_0)(1-\rho_n q_0/2)}{(\rho_n q_0/2)(1-\rho_n p_0)} > 0$$

such that when $n$ sufficiently large,

$$\Pr\left(\left|\log\frac{p(1-\hat{p})}{(1-p)\hat{p}}\right| \geq M_{11}\right) \leq 2\exp\left(-C_p n^2(1-\tau)^2\rho_n\right).$$

Therefore, using Bernstein's inequality, we obtain for any constant $C > 0$,

$$\Pr\left(\left|\log\frac{p(1-\hat{p})}{(1-p)\hat{p}}\sum_{(i,j)\in T_{11}^{(2)}}\left(A_{ij}^{(22)}-p\right)\right| > M_{11}n(1-\tau)\sqrt{C\rho_n p_0\log(n(1-\tau))}\right)$$

$$\leq \Pr\left(\left|\sum_{(i,j)\in T_{11}^{(2)}}\left(A_{ij}^{(22)}-p\right)\right| > n(1-\tau)\sqrt{C\rho_n p_0\log(n(1-\tau))}\right) + \Pr\left(\left|\log\frac{p(1-\hat{p})}{(1-p)\hat{p}}\right| \geq M_{11}\right)$$

$$\leq \sum_{h=0}^{n^2(1-\tau)^2}\Pr\left(\left|\sum_{(i,j)\in T_{11}^{(2)}}\left(A_{ij}^{(22)}-p\right)\right| > n(1-\tau)\sqrt{C\rho_n p_0\log(n(1-\tau))}\,\middle|\, |T_{11}^{(2)}| = h\right)\Pr\left(|T_{11}^{(2)}| = h\right)$$

$$+ 2\exp\left(-C_p n^2(1-\tau)^2\rho_n\right)$$

$$\leq 2\sum_{h=0}^{n^2(1-\tau)^2}\exp\left(-\frac{Cn^2(1-\tau)^2\rho_n p_0\log(n(1-\tau))}{2(h\rho_n p_0 + n(1-\tau)\sqrt{C\rho_n p_0\log(n(1-\tau))}/3)}\right)\Pr\left(|T_{11}^{(2)}| = h\right)$$

$$+ 2\exp\left(-C_p n^2(1-\tau)^2\rho_n\right)$$

$$\leq 2\exp\left(-\frac{2n^2(1-\tau)^2\rho_n p_0 \log\left(n\left(1-\tau\right)\right)}{2(n^2(1-\tau)^2\rho_n p_0 + n(1-\tau)\sqrt{C\rho_n p_0 \log(n(1-\tau))}/3)}\right) + 2\exp\left(-C_p n^2(1-\tau)^2\rho_n\right)$$

$$\leq 2\exp\left(-\frac{C}{4}\log\left(n(1-\tau)\right)\right) + 2\exp\left(-C_p n^2(1-\tau)^2\rho_n\right), \tag{53}$$

where the last inequality holds by condition (ii), which implies

$$n^2(1-\tau)^2\rho_n p_0 \gg n(1-\tau)\sqrt{C\rho_n p_0 \log(n(1-\tau))}/3, \qquad \text{as } n \to \infty.$$

Similar inequalities hold for $I_{12}, I_{21}, I_{22}$. Therefore, under condition (ii), let $C > 4$, we have

$$|I| = \left|\tilde{\ell}_{K^*} - \tilde{\xi}_{K^*} + \hat{\xi}_K - \hat{\ell}_K\right| = O_p\left(n(1-\tau)\sqrt{\rho_n \log(n(1-\tau))}\right). \tag{54}$$

For the term $II$, a similar analysis as in the proof of Theorem 1 still holds, except for the tail probability changed by Lemma A.15. We could obtain that when $K > K^*$,

$$\Pr\left(\left|T_{22}^{(2)}\right| D_{KL}(q \parallel \hat{q}) \geq n^2(1-\tau)^2\frac{\gamma_2\pi_0^2}{K^{*2}}d^{+2}\right) \geq 1 - \exp\left(-C_{K^+}n^2(1-\tau)^2\rho_n\right),$$

and when $K < K^*$,

$$\Pr\left(\left|T_{11}^{(2)}\right| D_{KL}(p \parallel \hat{p}) \geq n^2(1-\tau)^2\frac{\nu_2\pi_0^2}{K^{*2}}d^{-2}\right) \geq \quad 1 - \exp\left(-C_{K^-}n^2(1-\tau)^2\rho_n\right).$$

Thus, there exists $L > 0$ dependent on $\pi_0$, $K^*$, $K$, $p_0$ and $q_0$ such that

$$\Pr\left(II = \tilde{\xi}_{K^*} - \hat{\xi}_K \geq Ln^2(1-\tau)^2\rho_n^2\right) \geq 1 - \exp\left(-C'n^2(1-\tau)^2\rho_n\right). \tag{55}$$

Finally, putting (54) and (55) together, we conclude when $K \neq K^*$,

$$\Pr\left(\tilde{\ell}_{K^*} - \hat{\ell}_K > 0 \mid \phi_{K^*} = \phi^*\right) \to 1 \quad \text{as } n \to \infty.$$

This gives the desired result. $\qquad\square$

*Proof of Corollary 1 (MSE objective).* Similarly as in the proof of likelihood objective, we know by Lemma A.14, under conditions (i) and (ii′), we have $\Pr\left(\hat{\phi}_{K^*} = \phi^*\right) \to 1$ on $\mathcal{N}^{(2)}$.

We use the same expansion for $\hat{\ell}_K$, $\tilde{\ell}_{K^*}$ and use the same definition for $\hat{\varphi}_K$, $\tilde{\varphi}_{K^*}$, $\hat{\xi}_K$, $\tilde{\xi}_{K^*}$, $I$, $I_{11}$, $I_{12}$, $I_{21}$, $I_{22}$ and $II$.

Notice that by Lemma A.13, we have $\hat{p}$ concentrates around order $\rho_n$. More precisely, we have

$$\Pr\left(|p - \hat{p}| \geq \rho_n p_0\right) \leq 2\exp\left(-C_p n^2(1-\tau)^2\rho_n\right).$$

Thus, for $I_{11}$, by Bernstein's inequality and a similar analysis as (53), we have

$$\Pr\left(\left|(p - \hat{p})\sum_{(i,j)\in T_{11}^{(2)}}\left(A_{ij}^{(22)} - p\right)\right| > \rho_n p_0 n(1-\tau)\sqrt{C\rho_n p_0 \log(n(1-\tau))}\right)$$

$$\leq \Pr\left(\left|\sum_{(i,j)\in T_{11}^{(2)}} \left(A_{ij}^{(22)} - p\right)\right| > n(1-\tau)\sqrt{C\rho_n p_0 \log(n(1-\tau))}\right) + \Pr\left(|p - \hat{p}| \geq \rho_n p_0\right)$$

$$\leq \sum_{h=0}^{n^2(1-\tau)^2} \Pr\left(\left|\sum_{(i,j)\in T_{11}^{(2)}} \left(A_{ij}^{(22)} - p\right)\right| > n(1-\tau)\sqrt{C\rho_n p_0 \log(n(1-\tau))}\,\Bigg|\, |T_{11}^{(2)}| = h\right) \Pr\left(|T_{11}^{(2)}| = h\right)$$

$$+ 2\exp\left(-C_p n^2(1-\tau)^2 \rho_n\right)$$

$$\leq 2\exp\left(-\frac{C}{4}\log\left(n(1-\tau)\right)\right) + 2\exp\left(-C_p n^2(1-\tau)^2 \rho_n\right).$$

Similar inequalities hold for $I_{12}, I_{21}, I_{22}$. Therefore, under condition (ii′),

$$|I| = \left|\hat{\varphi}_K - \hat{\xi}_K + \tilde{\xi}_{K^*} - \tilde{\varphi}_{K^*}\right| = O_p\left(\rho_n^{3/2} n(1-\tau)\sqrt{\log(n(1-\tau))}\right). \tag{56}$$

Next we write the term *II* as

$$II = \sum_{(i,j)\in T_{11}^{(2)}} (\hat{p} - p)^2 + \sum_{(i,j)\in T_{12}^{(2)}} (\hat{q} - p)^2 + \sum_{(i,j)\in T_{21}^{(2)}} (\hat{p} - q)^2 + \sum_{(i,j)\in T_{22}^{(2)}} (\hat{q} - q)^2.$$

Each of the four terms here are positive obviously, and has exactly the form of the lower bound for KL divergence we used in the likelihood objective case. Thus, using an exact argument as in (55), we have

$$\Pr\left(II = \hat{\xi}_K - \tilde{\xi}_{K^*} \geq Ln^2(1-\tau)^2 \rho_n^2\right) \geq 1 - \exp\left(-C' n^2(1-\tau)^2 \rho_n\right). \tag{57}$$

Putting (56) and (57) together, using condition (ii′) we reach the conclusion. $\qquad\square$

## A.4   Proof of Theorem 2

**Lemma A.16.** *Under Assumptions 1 and 4, the maximum likelihood estimators $\{\hat{p}_{uv}\}$ as defined in Section 2.2 for any candidate $K \geq 2$ are bounded away from $0$ and $1$ in probability so that*

$$\Pr\left(\frac{\delta_n}{2} < \hat{p}_{uv} < 1 - \frac{\delta_n}{2}\right) \geq 1 - 2\exp\left(-C_p n^2(1-\tau)^2 \delta_n^2\right), \tag{58}$$

*where $\tau$ is the INCV node splitting ratio, $\delta_n$ is defined in Assumption 4, $C_p$ is a positive constant only dependent on $K$ and $\pi_0$ (from Assumption 1).*

*Proof of Lemma A.16.* Recall that

$$\hat{p}_{uv} = \frac{\sum_{(i,j)\in \mathcal{E}_{uv}} A_{ij}^{(22)}}{|\mathcal{E}_{uv}|}.$$

Because every $A_{ij}^{(22)}$ has an expected value equal to some $p_{rs}$, it is obvious by Assumption 4 that

$$0 < \delta_n < \min_{r,s\in[K^*]} p_{rs} \leq \mathbb{E}(\hat{p}_{uv}) \leq \max_{r,s\in[K^*]} p_{uv} < 1 - \delta_n < 1. \tag{59}$$

By the discussion after Remark 4 for Assumption 4, we know that the recovered label $\hat{\phi}_K$

on $\mathcal{N}^{(2)}$ is still balanced with respect to constant $\pi_0$. Thus we know that for any $u, v$,

$$\frac{n^2(1-\tau)^2\pi_0^2}{K^2} \leq |\mathcal{E}_{uv}| \leq n^2(1-\tau)^2\left(1 - \frac{k-1}{K}\pi_0\right)^2.$$

Next using the Law of Total Probability and Hoeffding's Inequality, we have

$$\Pr\left(|\hat{p}_{uv} - \mathbb{E}(\hat{p}_{uv})| \geq \frac{\delta_n}{2}\right) = \sum_{h=\frac{n^2(1-\tau)^2\pi_0^2}{K^2}}^{n^2(1-\tau)^2\left(1-\frac{k-1}{K}\pi_0\right)^2} \Pr\left(|\hat{p}_{uv} - \mathbb{E}(\hat{p}_{uv})| \geq \frac{\delta_n}{2} \mid |\mathcal{E}_{uv}| = h\right)\Pr\left(|\mathcal{E}_{uv}| = h\right)$$

$$\leq \sum_{h=\frac{n^2(1-\tau)^2\pi_0^2}{K^2}}^{n^2(1-\tau)^2\left(1-\frac{k-1}{K}\pi_0\right)^2} 2\exp\left(-\frac{\delta_n^2}{2}h\right)\Pr\left(|\mathcal{E}_{uv}| = h\right)$$

$$\leq 2\exp\left(-\frac{n^2(1-\tau)^2\pi_0^2\delta_n^2}{K^2}\right)$$

$$\leq 2\exp\left(-C_p n^2(1-\tau)^2\delta_n^2\right). \tag{60}$$

Equations (59) and (60) together imply

$$\Pr\left(\frac{\delta_n}{2} < \hat{p}_{uv} < 1 - \frac{\delta_n}{2}\right) \geq 1 - 2\exp\left(-C_p n^2(1-\tau)^2\delta_n^2\right).$$

$\square$

**Lemma A.17.** *Suppose an undirected network $(\mathcal{N}, \mathcal{E})$ generated from a stochastic block model satisfied Assumptions 1,3,4. When $K = K^*$, the derived community labeling function $\hat{\phi}_{K^*}$ on $\mathcal{N}^{(2)}$ in the Classification step of INCV satisfies*

$$\Pr(\hat{\phi}_{K^*} \neq \zeta(\phi^*)) \leq 2K^*n(1-\tau)\exp\left(-c_1 n\tau\beta_n^2\right) + K^*\exp(-c_2 n(1-\tau)) + g(n\tau),$$

*where $\tau$ is the node splitting ratio, $\zeta$ is the bijection introduced in Assumption 4, $\beta_n$ is the identification gap mentioned in Definition 2, $g(x)$ is the function defined in Assumption 3, and $c_1$ and $c_2$ are positive constants dependent on $\pi_0$ (from Assumption 1) and $K^*$ only.*

*Proof of Lemma A.17.* Similarly as in the proof of Lemma A.7, let the notation $n_k^{(1)}$ denote the cardinality of a subset of community $k$ that belongs to the training set $\mathcal{N}^{(1)}$. Then, we have exactly as (11) which says

$$\Pr\left(n_k^{(1)} \leq \frac{n\pi_0\tau}{2K^*}\right) \leq \exp(-c_0 n\tau),$$

for a positive constant $c_0$ dependent on $K^*$ and $\pi_0$ only.

Next we consider the case when $\hat{\phi}_{K^*} = \phi^*$ on $\mathcal{N}^{(1)}$. In this case, for any node $j \in \mathcal{N}^{(2)}$, let $l_j = \phi^*(j)$ denote its true community. Then according to the classification criterion, we have

$$\Pr\left(\hat{\phi}_{K^*}(j) \neq \zeta(\phi^*(j))\right) \leq \sum_{\substack{l=1 \\ l \neq \zeta(l_j)}}^{K^*} \Pr\left(\frac{\sum_{i \in \mathcal{N}^{(1)}:\phi^*(i)=\zeta(l_j)} A_{ij}^{(12)}}{n_{\zeta(l_j)}^{(1)}} < \frac{\sum_{i \in \mathcal{N}^{(1)}:\phi^*(i)=l} A_{ij}^{(12)}}{n_l^{(1)}}\right)$$

$$\leq \sum_{\substack{l=1 \\ l \neq \zeta(l_j)}}^{K^*} \left[ \Pr\left( \frac{\sum_{i \in \mathcal{N}^{(1)}:\phi^*(i)=\zeta(l_j)} A_{ij}^{(12)}}{n_{\zeta(l_j)}^{(1)}} - p_{\zeta(l_j)l_j} < -\frac{\beta_n}{2} \right) \right.$$

$$\left. + \Pr\left( \frac{\sum_{i \in \mathcal{N}^{(1)}:\phi^*(i)=l} A_{ij}^{(12)}}{n_l^{(1)}} - p_{ll_j} > \frac{\beta_n}{2} \right) \right], \tag{61}$$

where $p_{\zeta(l_j)l_j} - p_{ll_j} \geq \beta_n$ by definition of $\zeta$ and $\beta_n$.

Applying the Law of Total Probability and Hoeffding Inequality, we obtain

$$\Pr\left( \frac{\sum_{i \in \mathcal{N}^{(1)}:\phi^*(i)=\zeta(l_j)} A_{ij}^{(12)}}{n_{\zeta(l_j)}^{(1)}} - p_{\zeta(l_j)l_j} < -\frac{\beta_n}{2} \right)$$

$$\leq \sum_{h=0}^{n\tau} \Pr\left( \frac{\sum_{i \in \mathcal{N}^{(1)}:\phi^*(i)=\zeta(l_j)} A_{ij}^{(12)}}{n_{\zeta(l_j)}^{(1)}} - p_{\zeta(l_j)l_j} < -\frac{\beta_n}{2} \,\Big|\, n_{l_j}^{(1)} = h \right) \Pr(n_{l_j}^{(1)} = h)$$

$$\leq \sum_{h=\frac{n\pi_0\tau}{2K^*}}^{n\tau} \exp\left( -\frac{h\beta_n^2}{2} \right) \Pr(n_{l_j}^{(1)} = h) + \Pr\left( n_{l_j}^{(1)} \leq \frac{n\pi_0\tau}{2K^*} \right)$$

$$\leq \exp\left( -\frac{n\tau\pi_0\beta_n^2}{4K^*} \right) + \exp(-c_0 n\tau)$$

$$\leq \exp\left( -c_1 n\tau\beta_n^2 \right).$$

Similarly, we can prove

$$\Pr\left( \frac{\sum_{i \in \mathcal{N}^{(1)}:\phi^*(i)=l} A_{ij}^{(12)}}{n_l^{(1)}} - p_{ll_j} > \frac{\beta_n}{2} \right) \leq \exp\left( -c_1 n\tau\beta_n^2 \right).$$

Therefore, (61) can be rewritten as

$$\Pr\left( \hat{\phi}_{K^*}(j) \neq \zeta(\phi^*(j)) \right) \leq \sum_{\substack{l=1 \\ l \neq l_j}}^{K^*} 2\exp\left( -c_1 n\tau\beta_n^2 \right)$$

$$= 2K^* \exp\left( -c_1 n\tau\beta_n^2 \right).$$

It is also known by Assumption 3 that the probability of $\hat{\phi}_{K^*} \neq \phi^*$ on $\mathcal{N}^{(1)}$ is bounded by $g(n\tau)$. Thus on $\mathcal{N}^{(2)}$, the $\hat{\phi}_{K^*}$ obtained according to the classification step shall satisfy

$$\Pr(\hat{\phi}_{K^*} \neq \zeta(\phi^*)) \leq \sum_{j \in \mathcal{N}^{(2)}} \Pr\left( \hat{\phi}_{K^*}(j) \neq \zeta(\phi^*(j)) \right) + g(n\tau)$$

$$\leq n(1-\tau)2K^* \exp\left( -c_1 n\tau\beta_n^2 \right) + g(n\tau)$$

$$\leq 2K^* n(1-\tau) \exp\left( -c_1 n\tau\beta_n^2 \right) + g(n\tau). \tag{62}$$

Finally, by using a similar argument as (11), we can show there exists a positive constant $c_2$ such that

$$\Pr\left( n_k^{(2)} \leq \frac{n\pi_0(1-\tau)}{K^*} \right) \leq \exp(-c_2 n(1-\tau)), \quad k = 1, 2, \ldots, K^*.$$

Hence,

$$\Pr\left(\min_{1\leq k\leq K^*} n_k^{(2)} \leq \frac{n\pi_0(1-\tau)}{K^*}\right) \leq K^* \exp(-c_2 n(1-\tau)). \tag{63}$$

Recall that the re-classification process as described in Remark 1 is only needed when the recovered network structure for $\mathcal{N}^{(2)}$ is not balanced with respect to $\pi_0$. Thus according to (62) and (63), when $K = K^*$, the derived labeling function $\hat{\phi}_{K^*}$ on $\mathcal{N}^{(2)}$ even with the re-classification shall satisfy

$$\Pr\big(\hat{\phi}_{K^*} \neq \zeta(\phi^*)\big) \leq 2K^* n(1-\tau)\exp\Big(-c_1 n\tau\beta_n^2\Big) + K^* \exp(-c_2 n(1-\tau)) + g(n\tau).$$

□

Now we shall define the following sets for convenience. For any candidate $K$, let

$$T_{k_1,k_2,l_1,l_2} := \{(i,j) : i,j \in \mathcal{N}^{(2)}, i < j, \hat{\phi}_K(i) = k_1, \hat{\phi}_K(j) = k_2, \phi^*(i) = l_1, \phi^*(j) = l_2\}. \tag{64}$$

We have the following lemma to deal with underfitting.

**Lemma A.18.** *Assume the network is drawn from the stochastic block model with $K^*$ communities satisfying Assumptions 1,3,4. Suppose we cluster the nodes into $K^- < K^*$ communities. Then, under condition (ii) in Theorem 2, with probability tending to 1, there must exist $l_1, l_2, l_3 \in [K^*]$ and $k_1, k_2 \in [K^-]$ such that*

1. *$|T_{k_1,k_2,l_1,l_2}| \geq cn^2(1-\tau)^2$*

2. *$|T_{k_1,k_2,l_1,l_3}| \geq cn^2(1-\tau)^2$*

3. *$|p_{l_1 l_2} - p_{l_1 l_3}| \geq \beta_n$.*

*for some constant c only depends on $\pi_0$, $K^*$ and $K^-$.*

*Proof of Lemma A.18.* Denote the true community structure in $\mathcal{N}^{(2)}$ as $\mathcal{N}^{(2)} = \cup_{l=1}^{K^*} C_l^{*(2)}$ and denote the estimated community structure for $K^- < K^*$ as $\mathcal{N}^{(2)} = \cup_{k=1}^{K^-} \hat{C}_k^{(2)}$ For any $l \in [K^*]$, by pigeonhole principle, there exists $\hat{l} \in [K^-]$ such that

$$|\hat{C}_{\hat{l}}^{(2)} \cap C_l^{*(2)}| \geq \frac{|C_l^{*(2)}|}{K^-} = \frac{n_l^{(2)}}{K^-}.$$

Therefore there exists $l_2, l_3$ such that $\hat{l}_2 = \hat{l}_3 =: k_2$. Choose $l_1 = \zeta(l_2)$ and $k_1 = \hat{l}_1$, then by the definition of $\zeta$ and $\beta_n$, we have

$$p_{l_1 l_2} - p_{l_1 l_3} \geq \beta_n.$$

Moreover, we have

$$|T_{k_1,k_2,l_1,l_2}| = |\hat{C}_{\hat{l}_1}^{(2)} \cap C_{l_1}^{*(2)}| \times |\hat{C}_{\hat{l}_2}^{(2)} \cap C_{l_2}^{*(2)}| \geq \frac{n_{l_1}^{(2)} n_{l_2}^{(2)}}{(K^-)^2}.$$

We already know that there exists a positive constant $c_2$ such that for any $l$,

$$\Pr\left(n_l^{(2)} \leq \frac{n\pi_0(1-\tau)}{K^*}\right) \leq \exp(-c_2 n(1-\tau)), \quad k = 1, 2, \ldots, K^*.$$

Thus by a union bound, we have

$$|T_{k_1,k_2,l_1,l_2}| \geq \frac{n_{l_1}^{(2)} n_{l_2}^{(2)}}{(K^-)^2} \geq \frac{\pi_0^2 n^2 (1-\tau)^2}{(K^-)^2 K^{*2}} =: cn^2(1-\tau)^2$$

with probability larger than $1 - K^* \exp(-c_2 n(1-\tau))$. A similar inequality holds for $|T_{k_1,k_2,l_1,l_3}|$.
□

*Proof of Theorem 2 (likelihood objective).* For a candidate $K \in \{2, \ldots, K_{\max}\}$, denote the community labeling function derived from INCV as $\hat{\phi}_K$ and denote the minimized negative log-likelihood function as

$$\hat{\ell}_K = \ell(\hat{p}_{uv} \mid \hat{\phi}_K),$$

whose form is given in Section 2.2. Similarly as Theorem 1, since $\hat{p}_{uv}$ are the maximum likelihood estimators, it is natural to have

$$\hat{\ell}_{K^*} = \ell(\hat{p}_{uv} \mid \hat{\phi}_{K^*}) \leq \ell(p_{uv} \mid \hat{\phi}_{K^*}),$$

where $p_{uv}$ are the true block probabilities. For simplicity, denote $\ell(p_{uv} \mid \hat{\phi}_{K^*})$ as $\tilde{\ell}_{K^*}$. Moreover, Lemma A.17 states that on the test set $\mathcal{N}^{(2)}$,

$$\Pr(\hat{\phi}_{K^*} \neq \zeta(\phi^*)) \leq 2K^* n(1-\tau) \exp\left(-c_1 n\tau \beta_n^2\right) + K^* \exp(-c_2 n(1-\tau)) + g(n\tau),$$

where $c_1$ and $c_2$ are positive constants dependent on $K^*$ and $\pi_0$ only. The conditions (i) and (ii) in the theorem statement imply that $\Pr(\hat{\phi}_{K^*} = \zeta(\phi^*)) \to 1$ as $n \to \infty$. Thus, in order to show consistency, it is sufficient to prove that when $K < K^*$,

$$\Pr(\hat{\ell}_K - \tilde{\ell}_{K^*} > 0 \mid \hat{\phi}_{K^*} = \zeta(\phi^*)) \to 1.$$

In fact, since permuting the estimated label $\hat{\phi}_{K^*}$ does not change the value of the minimzed negative log-likelihood objective, from now on without loss of generality, we may assume that $\zeta \equiv \mathrm{id}_{K^*}$.

Recall the definitions of $T_{k_1,k_2,l_1,l_2}$ in (64). The minimized negative log-likelihood $\hat{\ell}_K = \ell(\hat{p}_{uv} \mid \hat{\phi}_K)$ can be rewritten as

$$\hat{\ell}_K = \sum_{k_1,k_2,l_1,l_2} \sum_{(i,j) \in T_{k_1 k_2 l_1 l_2}} \left[ -A_{ij}^{(22)} \log \hat{p}_{k_1 k_2} - (1 - A_{ij}^{(22)}) \log(1 - \hat{p}_{k_1 k_2}) \right].$$

Similarly, when $\hat{\phi}_{K^*} = \phi^*$, $\tilde{\ell}_{K^*} = \ell(p_{uv} \mid \phi^*)$ can be rewritten as

$$\tilde{\ell}_{K^*} = \sum_{k_1,k_2,l_1,l_2} \sum_{(i,j) \in T_{k_1 k_2 l_1 l_2}} \left[ -A_{ij}^{(22)} \log p_{l_1 l_2} - (1 - A_{ij}^{(22)}) \log(1 - p_{l_1 l_2}) \right].$$

Furthermore, define

$$\tilde{\xi}_{K^*} = \sum_{k_1,k_2,l_1,l_2} \sum_{(i,j) \in T_{k_1 k_2 l_1 l_2}} \left[ p_{l_1 l_2} \log p_{l_1 l_2} + (1 - p_{l_1 l_2}) \log(1 - p_{l_1 l_2}) \right],$$

and for $K \neq K^*$, define

$$\hat{\xi}_K = \sum_{k_1,k_2,l_1,l_2} \sum_{(i,j)\in T_{k_1 k_2 l_1 l_2}} \left[ p_{l_1 l_2} \log \hat{p}_{k_1 k_2} + (1 - p_{l_1 l_2}) \log(1 - \hat{p}_{k_1 k_2}) \right].$$

Then we can write

$$\hat{\ell}_K - \tilde{\ell}_{K^*} = \underbrace{(\tilde{\ell}_{K^*} - \tilde{\xi}_{K^*}) + (\hat{\xi}_K - \hat{\ell}_K)}_{I} + \underbrace{\tilde{\xi}_{K^*} - \hat{\xi}_K}_{II}. \tag{65}$$

For $I$, we have

$$I = \sum_{k_1,k_2,l_1,l_2} \sum_{(i,j)\in T_{k_1 k_2 l_1 l_2}} (A_{ij}^{(22)} - p_{l_1 l_2}) \log \frac{p_{l_1 l_2}(1 - \hat{p}_{k_1 k_2})}{\hat{p}_{k_1 k_2}(1 - p_{l_1 l_2})} =: \sum_{k_1,k_2,l_1,l_2} I_{k_1,k_2,l_1,l_2}. \tag{66}$$

By Lemma A.16, there exists a constant

$$M_{11,n} = 2 \left| \log \frac{\delta_n}{2} \right| > \log \frac{(1 - \delta_n)(1 - \delta_n/2)}{\delta_n^2/2} > 0$$

depending on $\delta_n$ in Assumption 2 such that for each $k_1, k_2, l_1, l_2$,

$$\Pr\left( \left| \log \frac{p_{l_1 l_2}(1 - \hat{p}_{k_1 k_2})}{\hat{p}_{k_1 k_2}(1 - p_{l_1 l_2})} \right| \geq M_{11,n} \right) \leq 2 \exp\left( -C_p n^2 (1 - \tau)^2 \delta_n^2 \right).$$

Therefore, using Hoeffding's inequality, similarly as (33) we obtain for each

$$\Pr\left( \left| \log \frac{p_{l_1 l_2}(1 - \hat{p}_{k_1 k_2})}{\hat{p}_{k_1 k_2}(1 - p_{l_1 l_2})} \sum_{(i,j)\in T_{k_1,k_2,l_1,l_2}} \left( A_{ij}^{(22)} - p_{l_1 l_2} \right) \right| > 2n(1-\tau)\log(n(1-\tau)) \left| \log \frac{\delta_n}{2} \right| \right)$$

$$\leq \Pr\left( \left| \sum_{(i,j)\in T_{k_1,k_2,l_1,l_2}} \left( A_{ij}^{(22)} - p_{l_1 l_2} \right) \right| > n(1-\tau)\log(n(1-\tau)) \right) + \Pr\left( \left| \log \frac{p_{l_1 l_2}(1 - \hat{p}_{k_1 k_2})}{\hat{p}_{k_1 k_2}(1 - p_{l_1 l_2})} \right| \geq M_{11,n} \right)$$

$$\leq \sum_{h=0}^{n^2(1-\tau)^2} \Pr\left( \left| \sum_{(i,j)\in T_{k_1,k_2,l_1,l_2}} \left( A_{ij}^{(22)} - p_{l_1 l_2} \right) \right| > n(1-\tau)\log(n(1-\tau)) \middle| |T_{k_1,k_2,l_1,l_2}| = h \right)$$

$$\cdot \Pr(|T_{k_1,k_2,l_1,l_2}| = h) + 2 \exp\left( -C_p n^2 (1-\tau)^2 \delta_n^2 \right)$$

$$\leq 2 \sum_{h=0}^{n^2(1-\tau)^2} \exp\left( -\frac{2n^2(1-\tau)^2 \log^2(n(1-\tau))}{h} \right) \Pr(|T_{k_1,k_2,l_1,l_2}| = h) + 2 \exp\left( -C_p n^2 (1-\tau)^2 \delta_n^2 \right)$$

$$\leq 2 \exp\left( -2 \log^2(n(1-\tau)) \right) + 2 \exp\left( -C_p n^2 (1-\tau)^2 \delta_n^2 \right). \tag{67}$$

Therefore, under condition (ii),

$$|I| = \left| \tilde{\ell}_{K^*} - \tilde{\xi}_{K^*} + \hat{\xi}_K - \hat{\ell}_K \right| = O_p\left( n(1-\tau)\log(n(1-\tau)) \left| \log \frac{\delta_n}{2} \right| \right). \tag{68}$$

Next, we expand the term $II$ in (65) as

$$II = \sum_{k_1,k_2,l_1,l_2} |T_{k_1 k_2 l_1 l_2}| \left( p_{l_1 l_2} \log \frac{p_{l_1 l_2}}{\hat{p}_{k_1 k_2}} + (1 - p_{l_1 l_2}) \log \frac{1 - p_{l_1 l_2}}{1 - \hat{p}_{l_1 l_2}} \right) = \sum_{k_1,k_2,l_1,l_2} |T_{k_1 k_2 l_1 l_2}| D(p_{l_1 l_2} \| \hat{p}_{k_1 k_2}).$$

(69)

By definition, all the KL divergences are positive. When $K < K^*$, recall $k_1, k_2, l_1, l_2, l_3$ introduced in Lemma A.18. Without loss of generality, we may assume that $(k_1, k_2, l_1, l_2, l_3) = (1, 2, 1, 2, 3)$. Then consider only $|T_{1212}| \cdot D_{KL}(p_{12} \| \hat{p}_{12}) + |T_{1213}| \cdot D_{KL}(p_{13} \| \hat{p}_{12})$. By Lemma A.12,

$$D_{KL}(q \| \hat{q}) \geq \frac{(q - \hat{q})^2}{(q + \hat{q})(2 - q - \hat{q})} \geq (q - \hat{q})^2.$$

Thus we have

$$II \geq |T_{1212}| (p_{12} - \hat{p}_{12})^2 + |T_{1213}| (p_{13} - \hat{p}_{12})^2.$$

Denote

$$g(x) = |T_{1212}| (p_{12} - x)^2 + |T_{1213}| (p_{13} - x)^2$$
$$= (|T_{1212}| + |T_{1213}|) x^2 - 2 (|T_{1212}| p_{12} + |T_{1213}| p_{13}) x + \left( |T_{1212}| p_{12}^2 + |T_{1213}| p_{13}^2 \right)$$

which is minimized at

$$x_0 = \frac{|T_{1212}| p_{12} + |T_{1213}| p_{13}}{|T_{1212}| + |T_{1213}|}.$$

Denote $t = \frac{|T_{1212}|}{|T_{1212}| + |T_{1213}|}$. Then by Lemma A.18, both $|T_{1212}|$ and $|T_{1213}|$ are of order $n^2 (1 - \tau)^2$, thus $t$ is of constant order. Thus

$$II \geq g(x_0) = \left( (1 - t)^2 |T_{1212}| + t^2 |T_{1213}| \right) (p_{12} - p_{13})^2 = \Omega_p \left( n^2 (1 - \tau)^2 \beta_n^2 \right). \qquad (70)$$

Combining (68) and (70), with condition (ii), one can get the desired result. $\qquad \square$

*Proof of Theorem 2 (MSE objective).* The proof follows the same manner as in the proof of MSE objective of Theorem 1, with modifications similar to the likelihood objective above. Thus, we will just state some significant difference in the proof, while the remaining proof is omitted.

Let $\hat{\ell}_K = \text{MSE}(\hat{p}_{uv} \mid \hat{\phi}_K)$ and $\tilde{\ell}_{K^*} = \text{MSE}(p_{uv} \mid \hat{\phi}_{K^*})$. Then

$$\hat{\ell}_K = \sum_{k_1,k_2,l_1,l_2} \sum_{(i,j) \in T_{k_1 k_2 l_1 l_2}} \left( A_{ij}^{(22)} - \hat{p}_{k_1 k_2} \right)^2$$
$$= \sum_{(i,j) \in \mathcal{E}^{(2)}} \left( A_{ij}^{(22)} \right)^2 - 2 \sum_{k_1,k_2,l_1,l_2} \sum_{(i,j) \in T_{k_1 k_2 l_1 l_2}} \hat{p}_{k_1 k_2} A_{ij}^{(22)} + \sum_{k_1,k_2,l_1,l_2} \sum_{(i,j) \in T_{k_1 k_2 l_1 l_2}} \hat{p}_{k_1 k_2}^2$$
$$:= \sum_{(i,j) \in \mathcal{E}^{(2)}} \left( A_{ij}^{(22)} \right)^2 + \hat{\varphi}_K$$

and

$$\tilde{\ell}_{K^*} = \sum_{k_1,k_2,l_1,l_2} \sum_{(i,j) \in T_{k_1 k_2 l_1 l_2}} \left( A_{ij}^{(22)} - p_{l_1 l_2} \right)^2$$

$$
= \sum_{(i,j)\in\mathcal{E}^{(2)}} \left(A_{ij}^{(22)}\right)^2 - 2 \sum_{k_1,k_2,l_1,l_2} \sum_{(i,j)\in T_{k_1k_2l_1l_2}} p_{l_1l_2} A_{ij}^{(22)} + \sum_{k_1,k_2,l_1,l_2} \sum_{(i,j)\in T_{k_1k_2l_1l_2}} p_{l_1l_2}^2
$$

$$
:= \sum_{(i,j)\in\mathcal{E}^{(2)}} \left(A_{ij}^{(22)}\right)^2 + \tilde{\varphi}_{K^*},
$$

and similarly define

$$
\tilde{\xi}_{K^*} = - \sum_{k_1,k_2,l_1,l_2} \sum_{(i,j)\in T_{k_1k_2l_1l_2}} p_{l_1l_2}^2,
$$

$$
\hat{\xi}_K = -2 \sum_{k_1,k_2,l_1,l_2} \sum_{(i,j)\in T_{k_1k_2l_1l_2}} \hat{p}_{k_1k_2} p_{l_1l_2} + \sum_{k_1,k_2,l_1,l_2} \sum_{(i,j)\in T_{k_1k_2l_1l_2}} \hat{p}_{k_1k_2}^2.
$$

Write

$$
\hat{\varphi}_K - \tilde{\varphi}_{K^*} = \underbrace{(\hat{\varphi}_K - \hat{\xi}_K) + (\tilde{\xi}_{K^*} - \tilde{\varphi}_{K^*})}_{I} + \underbrace{(\hat{\xi}_K - \tilde{\xi}_{K^*})}_{II}.
$$

For $I$, we have

$$
I = 2 \sum_{k_1,k_2,l_1,l_2} \sum_{(i,j)\in T_{k_1k_2l_1l_2}} (p_{l_1l_2} - \hat{p}_{k_1k_2}) \left(A_{ij}^{(22)} - p_{l_1l_2}\right) := \sum_{k_1,k_2,l_1,l_2} I_{k_1k_2l_1l_2}.
$$

Note that $|p_{l_1l_2} - \hat{p}_{k_1k_2}| \le 2$. Thus, a similar argument as in (40) and (67) yields that

$$
|I| = O_p \left(n(1-\tau)\log(n(1-\tau))\right). \tag{71}
$$

Next we can write $II$ as

$$
II = \sum_{k_1,k_2,l_1,l_2} |T_{k_1k_2l_1l_2}| (\hat{p}_{k_1k_2} - p_{l_1l_2})^2.
$$

Thus a similar analysis as before yields

$$
II = \Omega_p \left(n^2(1-\tau)^2 \beta_n^2\right).
$$

Combining this with (71) and condition (ii′), one can get the desired result. □

## A.5 Proof of Corollary 2

This proof follows exactly the same mentality as in the proof of Corollary 1 and Theorem 2, so we just list the lemmas that need to be changed here.

**Lemma A.19.** *Under Assumptions 1 and 2, the maximum likelihood estimate $\{\hat{p}_{uv}\}$ as defined in Section 2.2 for any candidate $K \ge 2$ are of order $\rho_n$ in probability so that there exists constants $\underline{C} < \overline{C}$ depending on $P_0$ such that*

$$
\Pr\left(\underline{C}\rho_n < \hat{p}_{uv} < \overline{C}\rho_n\right) \ge 1 - 2\exp\left(-C_p n^2(1-\tau)^2\rho_n\right), \tag{72}
$$

*where $\tau$ is the INCV node splitting ratio, $C_p$ is a positive constant only dependent on $K$ and $\pi_0$ (from Assumption 1).*

**Lemma A.20.** *Suppose an undirected network $(\mathcal{N}, \mathcal{E})$ generated from a stochastic block model satisfying Assumptions 1,3 and the structure assumption in Corollary 2. When $K = K^*$, the derived community labeling function $\hat{\phi}_{K^*}$ on $\mathcal{N}^{(2)}$ in the Classification step of INCV satisfies*

$$\Pr(\hat{\phi}_{K^*} \neq \zeta(\phi^*)) \leq 2K^* n(1-\tau) \exp(-c_1 n\tau\rho_n) + K^* \exp(-c_2 n(1-\tau)) + g(n\tau),$$

*where $\tau$ is the node splitting ratio, $\zeta$ is the bijection introduced in Assumption 4, $g(x)$ is the function defined in Assumption 3, and $c_1$ and $c_2$ are positive constants dependent on $\pi_0$ (from Assumption 1) and $K^*$ only.*

**Lemma A.21.** *Assume the network is drawn from the stochastic block model with $K^*$ communities satisfying Assumptions 1,3 and the structure assumption in Corollary 2. Suppose we cluster the nodes into $K^- < K^*$ communities. Then, under condition (ii) in Corollary 2, with probability tending to 1, there must exist $l_1, l_2, l_3 \in [K^*]$ and $k_1, k_2 \in [K^-]$ such that*

1. $|T_{k_1,k_2,l_1,l_2}| \geq cn^2(1-\tau)^2$
2. $|T_{k_1,k_2,l_1,l_3}| \geq cn^2(1-\tau)^2$
3. $|p_{l_1 l_2} - p_{l_1 l_3}| \geq c\rho_n.$

*for some constant $c$ only dependes on $\pi_0$, $K^*$ and $K^-$.*

Then the proof of Corollary 2 follows from changing the Hoeffding's inequality used in the expansion of terms $I$ into Bernstein's inequality, as done in the proof of Corollary 1.

# B    Additional Simulations

## B.1    Computational Cost for Different Methods

As a continuation of the simulation comparison in Table 1 in the main article, we further report the computational cost of different methods. Since the purpose of this experiment is to provide a representative runtime comparison rather than to repeat all settings in Table 1, we use a subset of the simulation settings considered there. Specifically, we set $n = 500$, $K^* \in \{3, 6\}$, candidate values $K \in \{2, \ldots, 9\}$, and use 10-fold node splitting. For the community-size regimes, we consider $n_{\min} \in \{90, 150\}$ when $K^* = 3$ and $n_{\min} \in \{50, 80\}$ when $K^* = 6$. The sparsity levels are $q \in \{0.1, 0.3\}$, with $p = 3q$.

For each replication and each simulation setting, we record the elapsed running time in seconds for each method. The reported runtime for INCV corresponds to one complete INCV run, which produces both the likelihood-based and MSE-based selections. Therefore, the runtime of INCV should be interpreted as the cost of computing two validation criteria within the same node-splitting framework. The results are summarized in Table B.1.

Table B.1: Comparison of INCV with cross-validation-based and BIC-type model selection methods.

| Method | Mean | SD | Median |
|--------|------|------|--------|
| NCV | 1.647 | 0.227 | 1.665 |
| LRBIC | 1.938 | 0.149 | 1.896 |
| ECV | 3.681 | 0.410 | 3.606 |
| INCV | 5.106 | 0.712 | 5.135 |
| NETCROP | 5.253 | 0.321 | 5.277 |
| CLBIC | 249.778 | 18.031 | 251.423 |

Table B.1 shows that INCV has a moderate computational cost. It is slightly slower than NCV, ECV, and LRBIC in this implementation, partly because a single INCV run evaluates both the likelihood-based and MSE-based losses across folds and candidate values. Nevertheless, its runtime is comparable to NETCROP and remains within a practical range. In contrast, CLBIC is substantially more computationally intensive. Since Table 1 in the main article shows that CLBIC has performance similar to LRBIC in most settings while requiring much larger computational cost, we only include LRBIC as the BIC-type benchmark in the additional supplementary experiments below.

## B.2    Empirical Diagnostic for the Estimated-Label Balance Condition

Recall that Assumption 1 requires not only the true community labels but also the estimated labels produced by INCV to be balanced. As discussed in the main text, the latter requirement is mainly a technical regularity condition used in the theoretical analysis. In particular, in the affiliation SBM, this condition helps rule out degenerate overfitted partitions when $K > K^*$, ensuring that the relevant validation edge sets have sufficiently large cardinalities. This is needed to separate overfitted partitions from the true partition in the validation loss.

We emphasize that no forced balancing or reclassification is used in our numerical experiments. All reported numerical results are based on the natural outputs of the clustering and classification steps. To assess the practical relevance of the estimated-label balance condition, we conduct an empirical diagnostic under the affiliation SBM.

For each overfitted candidate value $K > K^*$, we compute the empirical balance ratio

$$\widehat{r}_K = \frac{K \min_{1 \leq k \leq K} |\hat{\mathcal{N}}_k|}{|\mathcal{N}|},$$

where $\hat{\mathcal{N}}_k$ denotes the $k$-th estimated community under the candidate value $K$. A value close to one indicates nearly balanced estimated communities, whereas a value close to zero indicates severe imbalance. We compute this quantity separately on the evaluation set $\mathcal{N}^{(2)}$ and on the full estimated label vector after combining the training and evaluation labels.

For each replication, we summarize $\widehat{r}_K$ over all folds and all overfitted candidate values $K > K^*$ by its mean and its 5% quantile. We then report the averages of these two summaries over 100 replications. We do not report the raw minimum, since it can be dominated by a single fold and a single large candidate value of $K$, especially when the evaluation fold is small.

The data are generated from the affiliation SBM with $q = p/3$, where $p \in \{0.1, 0.2, 0.3\}$. We consider $K^* \in \{3, 5\}$ and two community-size regimes for each $K^*$. For $n = 600$, the minimum community sizes are $n_{\min} \in \{130, 200\}$ when $K^* = 3$ and $n_{\min} \in \{80, 120\}$ when $K^* = 5$. For $n = 1200$, the same proportions are used, namely $n_{\min} \in \{260, 400\}$ for $K^* = 3$ and $n_{\min} \in \{160, 240\}$ for $K^* = 5$. The candidate values are $K \in \{2, \ldots, 8\}$ and 10-fold node splitting is used.

Table B.2: Empirical balance ratios for overfitted candidate values $K > K^*$ under the affiliation SBM. "Test Mean" and "Test Q05" are the averages over 100 replications of the within-replication mean and 5% quantile of $\widehat{r}_K$ on the evaluation set. "All Mean" and "All Q05" are the corresponding quantities for the full estimated labels.

| $n$ | $K^*$ | $p$ | $n_{\min}$ | Test Mean | Test Q05 | All Mean | All Q05 |
|-----|-------|-----|------------|-----------|----------|----------|---------|
| 600 | 3 | 0.1 | 130 | 0.542 | 0.259 | 0.734 | 0.590 |
| | | | 200 | 0.509 | 0.241 | 0.706 | 0.561 |
| | | 0.2 | 130 | 0.518 | 0.210 | 0.733 | 0.587 |
| | | | 200 | 0.501 | 0.221 | 0.711 | 0.556 |
| | | 0.3 | 130 | 0.514 | 0.198 | 0.732 | 0.590 |
| | | | 200 | 0.511 | 0.243 | 0.717 | 0.568 |
| | 5 | 0.1 | 80 | 0.472 | 0.216 | 0.726 | 0.588 |
| | | | 120 | 0.490 | 0.225 | 0.729 | 0.591 |
| | | 0.2 | 80 | 0.438 | 0.202 | 0.652 | 0.526 |
| | | | 120 | 0.403 | 0.168 | 0.602 | 0.486 |
| | | 0.3 | 80 | 0.435 | 0.207 | 0.645 | 0.517 |
| | | | 120 | 0.399 | 0.162 | 0.594 | 0.479 |
| 1200 | 3 | 0.1 | 260 | 0.614 | 0.360 | 0.758 | 0.628 |
| | | | 400 | 0.604 | 0.395 | 0.739 | 0.594 |
| | | 0.2 | 260 | 0.612 | 0.361 | 0.763 | 0.637 |
| | | | 400 | 0.608 | 0.407 | 0.750 | 0.605 |
| | | 0.3 | 260 | 0.615 | 0.361 | 0.766 | 0.643 |
| | | | 400 | 0.610 | 0.407 | 0.750 | 0.608 |
| | 5 | 0.1 | 160 | 0.546 | 0.366 | 0.678 | 0.564 |
| | | | 240 | 0.497 | 0.303 | 0.628 | 0.520 |
| | | 0.2 | 160 | 0.530 | 0.354 | 0.678 | 0.564 |
| | | | 240 | 0.479 | 0.308 | 0.623 | 0.514 |
| | | 0.3 | 160 | 0.531 | 0.355 | 0.679 | 0.562 |
| | | | 240 | 0.484 | 0.310 | 0.624 | 0.521 |

Table B.2 shows that the empirical balance ratios remain well above zero across the reported settings. The full-label balance ratios are particularly stable: even for $n = 600$, the average 5% quantile is no smaller than approximately 0.48, and for $n = 1200$ it is no smaller than approximately 0.51. This suggests that the natural INCV output does not produce degenerate estimated communities at the full-label level. The test-side ratios are smaller and more variable,

especially when $n = 600$. This is expected because, under 10-fold cross-validation, the evaluation set contains only about $n/10$ nodes. For $n = 600$ and the largest candidate value $K = 8$, the average number of evaluation nodes per estimated community is only about 7.5, so the test-side balance ratio is more sensitive to fold-level fluctuations. When the network size increases to $n = 1200$, the test-side ratios become substantially more stable, with the average 5% quantile ranging from about 0.30 to 0.41. Overall, this diagnostic supports our interpretation that the estimated-label balance condition is primarily a theoretical regularity condition used to exclude degenerate overfitted partitions. In the reported numerical experiments, severe estimated-label imbalance is not a systematic issue, and no forced balancing post-processing is used.

## B.3   Additional Experiments under General SBMs

The main theoretical contribution of this paper focuses on the full consistency of INCV under the affiliation SBM. We further conduct supplementary simulations under structured general SBMs to examine the finite-sample behavior of INCV when the exact affiliation structure is violated. These experiments are intended as auxiliary evidence rather than as the main theoretical claim.

We generate networks from a general SBM whose block probability matrix is

$$p_{kl} = \rho \cdot 0.3^{\min\{|k-l|,3\}}, \qquad 1 \le k, l \le K^*.$$

This model is not an affiliation SBM, but it retains an assortative ordered-block structure, with larger connection probabilities for communities that are closer in index. We consider $K^* \in \{3, 5\}$ and $\rho \in \{0.1, 0.3, 0.6\}$, which represents sparse, moderate and dense regimes. For $n = 600$, the minimum community sizes are $n_{\min} \in \{50, 120, 200\}$ when $K^* = 3$ and $n_{\min} \in \{40, 70, 100\}$ when $K^* = 5$. For $n = 1200$, the same proportions are used, namely $n_{\min} \in \{100, 240, 400\}$ when $K^* = 3$ and $n_{\min} \in \{80, 140, 200\}$ when $K^* = 5$. We use candidate values $K \in \{2, \ldots, 8\}$ and repeat each setting 100 times.

As discussed in the main text, under unrestricted general SBMs the validation-block MLE implementation may severely over-select the number of communities. This occurs because, when each estimated block pair is assigned its own probability parameter, re-estimating these probabilities on $A^{(22)}$ can fit random fluctuations in the validation block. This behavior is consistent with our half-consistency result and the accompanying discussion for general SBMs. Therefore, in this supplementary comparison, we focus on the predictive plug-in implementation of INCV, where block probabilities are estimated from $A^{(11)}$ and then evaluated on $A^{(22)}$. We compare INCV-NLL-plug-in and INCV-MSE-plug-in (which will be written as INCV.nll and INCV.mse, consistent with the notation in Table 1) with NCV, ECV, NETCROP, and LRBIC. CLBIC is not included here because of its substantially higher computational cost as discussed before.

Several observations can be made from Table B.3. First, LRBIC is a strong benchmark in this structured general SBM experiment and achieves perfect or nearly perfect selection accuracy in most settings. This is consistent with the fact that BIC-type criteria explicitly account for model complexity. The purpose of this supplementary experiment is not to claim that INCV uniformly dominates BIC-type selection, but to examine whether the proposed node-splitting

Table B.3: Comparison of INCV with cross-validation-based and BIC-type model selection methods under general SBMs.

| $n$ | $K^*$ | $\rho$ | $n_{\min}$ | INCV.nll | INCV.mse | NCV | ECV | NETCROP | LRBIC |
|---|---|---|---|---|---|---|---|---|---|
| 600 | 3 | 0.1 | 50 | 0.00 | 0.00 | 0.00 | 0.00 | 0.00 | 0.86 |
| | | | 120 | 0.93 | 0.99 | 0.97 | 0.99 | 0.02 | 1.00 |
| | | | 200 | 0.99 | 0.99 | 0.98 | 1.00 | 0.64 | 1.00 |
| | | 0.3 | 50 | 0.02 | 0.01 | 0.00 | 0.00 | 0.00 | 1.00 |
| | | | 120 | 1.00 | 1.00 | 1.00 | 1.00 | 1.00 | 1.00 |
| | | | 200 | 0.98 | 0.97 | 0.99 | 1.00 | 1.00 | 1.00 |
| | | 0.6 | 50 | 0.99 | 0.99 | 0.99 | 1.00 | 0.99 | 1.00 |
| | | | 120 | 0.99 | 0.99 | 1.00 | 1.00 | 1.00 | 1.00 |
| | | | 200 | 0.97 | 0.97 | 0.99 | 1.00 | 1.00 | 1.00 |
| | 5 | 0.1 | 40 | 0.00 | 0.00 | 0.03 | 0.01 | 0.48 | 0.35 |
| | | | 70 | 0.00 | 0.00 | 0.05 | 0.00 | 0.51 | 1.00 |
| | | | 100 | 0.83 | 0.84 | 0.74 | 0.88 | 0.17 | 1.00 |
| | | 0.3 | 40 | 0.10 | 0.08 | 0.00 | 0.00 | 0.81 | 1.00 |
| | | | 70 | 0.94 | 0.90 | 0.99 | 1.00 | 0.98 | 1.00 |
| | | | 100 | 0.87 | 0.85 | 0.98 | 1.00 | 1.00 | 1.00 |
| | | 0.6 | 40 | 0.86 | 0.86 | 0.97 | 0.99 | 1.00 | 1.00 |
| | | | 70 | 0.93 | 0.91 | 0.99 | 0.99 | 1.00 | 1.00 |
| | | | 100 | 0.88 | 0.93 | 1.00 | 1.00 | 1.00 | 1.00 |
| 1200 | 3 | 0.1 | 100 | 0.00 | 0.00 | 0.00 | 0.00 | 0.00 | 1.00 |
| | | | 240 | 1.00 | 1.00 | 1.00 | 1.00 | 1.00 | 1.00 |
| | | | 400 | 1.00 | 0.99 | 1.00 | 1.00 | 0.99 | 1.00 |
| | | 0.3 | 100 | 1.00 | 1.00 | 0.97 | 1.00 | 0.19 | 1.00 |
| | | | 240 | 1.00 | 1.00 | 1.00 | 1.00 | 1.00 | 1.00 |
| | | | 400 | 1.00 | 1.00 | 0.98 | 1.00 | 1.00 | 1.00 |
| | | 0.6 | 100 | 1.00 | 1.00 | 1.00 | 1.00 | 1.00 | 1.00 |
| | | | 240 | 1.00 | 1.00 | 1.00 | 1.00 | 1.00 | 1.00 |
| | | | 400 | 0.97 | 0.97 | 1.00 | 1.00 | 1.00 | 1.00 |
| | 5 | 0.1 | 80 | 0.01 | 0.00 | 0.00 | 0.00 | 0.22 | 1.00 |
| | | | 140 | 0.94 | 0.97 | 0.88 | 0.98 | 0.09 | 1.00 |
| | | | 200 | 0.99 | 0.99 | 0.99 | 1.00 | 0.59 | 1.00 |
| | | 0.3 | 80 | 0.92 | 0.92 | 0.84 | 1.00 | 0.99 | 1.00 |
| | | | 140 | 0.98 | 0.98 | 0.98 | 1.00 | 1.00 | 1.00 |
| | | | 200 | 0.97 | 0.98 | 0.99 | 1.00 | 1.00 | 1.00 |
| | | 0.6 | 80 | 0.96 | 0.98 | 1.00 | 1.00 | 1.00 | 1.00 |
| | | | 140 | 0.96 | 0.96 | 1.00 | 0.99 | 1.00 | 1.00 |
| | | | 200 | 0.99 | 0.99 | 0.99 | 0.99 | 1.00 | 1.00 |

and induction framework remain practically competitive beyond the affiliation SBM.

Second, among cross-validation-based methods, the predictive plug-in versions of INCV are broadly comparable to NCV, ECV, and NETCROP. When the signal is moderate or strong and the smallest community is not too small, INCV-NLL and INCV-MSE often achieve high or perfect selection accuracy. The comparison between $n = 600$ and $n = 1200$ also shows a clear improvement as the network size increases, suggesting a reasonable finite-sample-to-large-sample trend. In the weakest and most imbalanced regimes, especially when $K^* = 5$ and $n_{\min}$ is small, all CV-based methods may become unstable, which is expected because the smallest communities contain limited information and the block separation is weak.

Overall, these experiments support the interpretation in the main text: under general SBMs, unrestricted validation-MLE can be prone to over-selection, while the predictive plug-in version provides a more stable practical implementation. Although our full consistency theory focuses on the affiliation SBM, these results suggest that the node-splitting and induction idea remains useful under broader structured SBM regimes.

## B.4 Column-wise Separation and Induction

Assumption 4 is imposed to guarantee the validity of the one-vs-all induction step analyzed in the main text. It is discussed after this assumption that if the column-wise separation condition is violated, the induction step may systematically mix different true communities, even when the block probability matrix has distinct rows and the communities are identifiable in the usual SBM sense. To illustrate this point, we conduct the following diagnostic experiment for this separation condition.

We consider a three-community SBM with

$$B = \rho B_0, \qquad B_0 = \begin{pmatrix} 1 & 1/2 & 1 \\ 1/2 & 1 & 1/3 \\ 1 & 1/3 & 2/3 \end{pmatrix}.$$

The rows of $B_0$ are distinct, but the column-wise separation condition required by Assumption 4 is violated. Thus, this setting is designed to isolate the role of the induction step rather than to provide a standard favorable SBM scenario. We set $n = 1000$, $\rho \in \{0.1, 0.3, 0.6\}$, and the minimum community sizes are chosen from $n_{\min} \in \{100, 200, 300\}$.

We compare two induction rules within the same INCV framework. The first is the affinity-based one-vs-all rule used in the theoretical analysis. The second is a likelihood-based induction rule, which says for a node $j \in \mathcal{N}^{(2)}$,

$$\hat{\phi}_K(j) = \arg \max_{v \in [K]} \left\{ \sum_{u \in [K]} \sum_{\substack{i \in \mathcal{N}^{(1)} \\ \hat{\phi}_K(i) = u}} \left( A_{ij} \log \hat{p}_{uv} + (1 - A_{ij}) \log(1 - \hat{p}_{uv}) \right) \right\},$$

where $\hat{p}_{uv}$ is the estimator of the block connectivity matrix derived using $A^{(11)}$. The purpose of this experiment is to show that the failure under violated column-wise separation is mainly tied to the simple affinity-based induction rule, and that the broader INCV framework can potentially

accommodate weaker identifiability conditions when equipped with a more informative induction step. The result is summarized in Table B.4.

Table B.4: Selection accuracy when the column-wise separation condition is violated. "INCV.aff.nll" and "INCV.aff.mse" refer to affinity induction rule, and "INCV.loss.nll" and "INCV.loss.mse" refer to likelihood-based induction rule.

| $\rho$ | $n_{\min}$ | INCV.aff.nll | INCV.aff.mse | INCV.loss.nll | INCV.loss.mse |
|---|---|---|---|---|---|
| 0.1 | 100 | 0.00 | 0.00 | 0.00 | 0.01 |
| | 200 | 0.00 | 0.00 | 0.00 | 0.00 |
| | 300 | 0.00 | 0.00 | 0.00 | 0.00 |
| 0.3 | 100 | 0.00 | 0.00 | 0.39 | 0.37 |
| | 200 | 0.00 | 0.00 | 0.96 | 0.96 |
| | 300 | 0.00 | 0.00 | 0.98 | 0.98 |
| 0.6 | 100 | 0.00 | 0.00 | 0.85 | 0.83 |
| | 200 | 0.00 | 0.00 | 1.00 | 1.00 |
| | 300 | 0.00 | 0.00 | 0.97 | 0.97 |

The affinity-based version almost always fails in this setting, which confirms that the column-wise separation condition is important for the current one-vs-all induction rule. In contrast, the likelihood-based induction rule substantially improves the selection accuracy when the signal is moderate or strong. For example, when $n_{\min} = 200$ or $300$ and $\rho \geq 0.3$, the likelihood-based version achieves accuracy above 0.90. This suggests that the strict separation condition is mainly tied to the simple affinity-based classifier used in the theoretical analysis, and that the broader INCV framework may accommodate weaker identifiability conditions with a more refined induction step. A rigorous theory for such alternative induction rules is left for future work.

## B.5 A Preliminary Extension to Degree-Corrected Stochastic Block Models

Although the theoretical analysis in this paper focuses on SBMs, the node-splitting and induction idea can be naturally adapted to degree-corrected stochastic block models (DCSBMs). We include a preliminary numerical experiment to examine this possibility. This experiment is intended as an auxiliary illustration rather than a complete theoretical treatment of DCSBM.

Under the DCSBM, the edge probability takes the form

$$\mathbb{P}(A_{ij} = 1) = \theta_i \theta_j p_{\phi(i)\phi(j)},$$

where $\theta_i$ captures node-level degree heterogeneity, and $P = (p_{uv})_{u,v=1}^{K^*}$, $\phi$ follows the same notation as in Section 2.1 in the main text. Here, one need to impose additional assumptions on $\theta$, such as $\sum_{\phi(i)=k} \theta_i^2 = n_k$ for each $k$, to ensure identifiability. Therefore, the one-vs-all classification step used for SBMs needs to be modified to account for degree effects. The following describes the implementation used in this preliminary DCSBM experiment. It follows the same three-stage structure as INCV: clustering on $A^{(11)}$, degree-adjusted induction using $A^{(12)}$, and predictive validation on $A^{(22)}$.

**Clustering.** In our implementation, we first apply a degree-normalized spectral clustering method on $A^{(11)}$ to estimate the training labels and the degree parameters of training nodes. Specifically, for each candidate $K$, we first compute the matrix of leading $K$ eigenvectors $\hat{U}^{(1,K)}$

for $A^{(11)}$. Then, we rescale $\theta$ and $P$ into $\theta'$ and $P'$ in the sense that

$$\theta'_i = \frac{\theta_i}{\sqrt{n^{(1)}_{\phi(i)}}} \qquad \text{and} \qquad p'_{uv} = p_{uv}\sqrt{n^{(1)}_u n^{(1)}_v},$$

where $n^{(1)}_u = |\mathcal{N}^{(1)}_u|$. We then estimate $\hat{\theta}'_{i,K} := \|\hat{U}^{(1,K)}_{i,\cdot}\|_2$ for $i \in \mathcal{N}^{(1)}$, and compute the normalized eigenvector matrix $\tilde{U}^{1,K}$ by $\tilde{U}^{(1,K)}_{i,\cdot} = \hat{U}^{(1,K)}_{i,\cdot}/\hat{\theta}'_{i,K}$. Now we compute $\hat{\phi}_K$ on $\mathcal{N}^{(1)}$ by implementing $k$-means algorithm on rows of $\tilde{U}^{(1,K)}$ with no more than $K$ clusters. We end the Clustering step by computing the block matrix using the information in $A^{(11)}$ by

$$\hat{p}'_{uv,K} = \frac{\sum_{(i,j)\in\mathcal{E}^{(1)}_{uv}} A_{ij}}{\sum_{(i,j)\in\mathcal{E}^{(1)}_{uv}} \hat{\theta}'_{i,K}\hat{\theta}'_{j,K}},$$

where $\mathcal{E}^{(1)}_{uv}$ is consistent with the notation in the main text.

**Classification.** For each node $j \in \mathcal{N}^{(2)}$, using the information from the clustering step, we classify it as

$$\hat{\phi}_K(j) = \arg\max_{1\leq k\leq K}\left\{\frac{\sum_{i\in\mathcal{N}^{(1)},\hat{\phi}_K(i)=k} A^{(12)}_{ij}}{\sum_{i\in\mathcal{N}^{(1)},\hat{\phi}_K(i)=k} \hat{\theta}'_{i,K}}\right\}.$$

We can also use the information in $A^{(12)}$ to compute an estimate of $\hat{\theta}'$ on $\mathcal{N}^{(2)}$, by

$$\hat{\theta}'_{j,k} = \frac{\sum_{i\in\mathcal{N}^{(1)}} A^{(12)}_{ij}}{\sum_{k\in[K]}\sum_{i\in\hat{\mathcal{N}}^{(1)}_k} \hat{\theta}'_{i,K}\hat{p}'_{k\hat{\phi}(j),K}}.$$

**Validation.** We can use both the negative log-likelihood or the MSE as the validation loss function. For the negative log-likelihood objective, it has the form

$$\ell(\{\hat{p}'_{uv,K}\},\hat{\phi}_K,\{\hat{\theta}'_{i,K}\}) = -\sum_{u,v=1}^{K}\sum_{(i,j)\in\mathcal{E}_{uv}}\left\{A^{(22)}_{ij}\log(\hat{\theta}'_{i,K}\hat{p}'_{uv,K}\hat{\theta}'_{j,K})+(1-A^{(22)}_{ij})\log(1-\hat{\theta}'_{i,K}\hat{p}'_{uv,K}\hat{\theta}'_{j,K})\right\}.$$

If the MSE objective is used, the one shall compute

$$\text{MSE}(\{\hat{p}'_{uv,K}\},\{\hat{\theta}'_{i,K}\}\mid\hat{\phi}_K) = \frac{1}{|\mathcal{E}^{(2)}|}\sum_{u,v=1}^{K}\sum_{(i,j)\in\mathcal{E}_{uv}}\left(A^{(22)}_{ij}-\hat{\theta}'_{i,K}\hat{p}'_{uv,K}\hat{\theta}'_{j,K}\right)^2.$$

The final estimator of $K^*$ is then obtained by minimizing the loss over all candidates. Here in implementation, the predicted probabilities are truncated to lie in $[\varepsilon, 1-\varepsilon]$ for numerical stability.

We generate networks with $n = 600$ and consider $K^* \in \{3,5\}$. We use the same generation mechanism for the block connectivity matrix as in Section B.3, while we generate $\theta \in \text{Unif}(1,10)$ and normalize it with constraint $\sum_{\phi(i)=k}\theta_i^2 = n_k$. We consider the same configuration settings as in Section B.3, but only consider the smaller sample size $n = 600$. We compare the DCSBM extension of INCV with NCV, ECV, NETCROP, and LRBIC.

Table B.5 shows that LRBIC is still a strong benchmark in this DCSBM experiment, espe-

Table B.5: Comparison of INCV with cross-validation-based and BIC-type model selection methods under DCSBMs.

| $K^*$ | $\rho$ | $n_{\min}$ | INCV.nll | INCV.mse | NCV | ECV | NETCROP | LRBIC |
|---|---|---|---|---|---|---|---|---|
| 3 | 0.1 | 50 | 0.00 | 0.00 | 0.00 | 0.00 | 0.00 | 0.00 |
| | | 120 | 0.85 | 0.89 | 0.11 | 0.82 | 0.00 | 1.00 |
| | | 200 | 0.96 | 0.98 | 0.90 | 1.00 | 0.98 | 1.00 |
| | 0.3 | 50 | 0.00 | 0.00 | 0.00 | 0.00 | 0.00 | 0.99 |
| | | 120 | 1.00 | 1.00 | 1.00 | 1.00 | 1.00 | 1.00 |
| | | 200 | 1.00 | 1.00 | 1.00 | 1.00 | 1.00 | 1.00 |
| | 0.6 | 50 | 0.29 | 0.56 | 0.04 | 0.98 | 0.79 | 1.00 |
| | | 120 | 1.00 | 1.00 | 1.00 | 1.00 | 1.00 | 1.00 |
| | | 200 | 1.00 | 1.00 | 1.00 | 1.00 | 1.00 | 1.00 |
| 5 | 0.1 | 40 | 0.01 | 0.02 | 0.02 | 0.00 | 0.03 | 0.00 |
| | | 70 | 0.10 | 0.08 | 0.01 | 0.00 | 0.01 | 0.79 |
| | | 100 | 0.55 | 0.52 | 0.01 | 0.71 | 0.14 | 1.00 |
| | 0.3 | 40 | 0.02 | 0.02 | 0.03 | 0.00 | 0.00 | 0.97 |
| | | 70 | 0.97 | 0.98 | 0.80 | 1.00 | 0.86 | 1.00 |
| | | 100 | 0.94 | 0.97 | 0.89 | 1.00 | 1.00 | 1.00 |
| | 0.6 | 40 | 0.98 | 0.99 | 0.51 | 0.99 | 0.47 | 1.00 |
| | | 70 | 1.00 | 1.00 | 1.00 | 1.00 | 1.00 | 1.00 |
| | | 100 | 1.00 | 1.00 | 1.00 | 1.00 | 1.00 | 1.00 |

cially when the signal is moderate or strong. This is consistent with the strong performance of BIC-type criteria observed in the SBM experiments. Among cross-validation-based methods, the proposed DCSBM extension of INCV is competitive and outperforms NCV and NETCROP in many moderate and strong signal regimes. However, it does not dominate ECV. In particular, ECV performs better in several difficult settings with small communities or strong degree heterogeneity.

This limitation is not unexpected. The DCSBM extension considered here is a preliminary plug-in implementation rather than a theoretically optimized version of INCV. Compared with the SBM setting, degree heterogeneity introduces an additional source of variation in both the classification and validation steps. After node splitting, the smallest communities may contain only limited information in the training block, and the estimation of degree-adjusted connectivity patterns can become unstable. Edge-sampling methods such as ECV retain all nodes in the fitting and validation procedure, which may be advantageous in some degree-heterogeneous and imbalanced regimes.

Overall, these results suggest that the node-splitting and induction idea can be extended to degree-corrected models once the classification and validation steps are adjusted for node-level degree heterogeneity. A rigorous theoretical analysis for DCSBM is left for future work.

# References

R. J. Serfling. Probability inequalities for the sum in sampling without replacement. *The Annals of Statistics*, 2(1):39–48, 1974.