# OpenReview forum: "Inductive Node-split Cross-Validation in Networks"
_SLADS/Section_A — Accepted by SLADS_Section_A_

### Review · Reviewer_zTE9 · 2026-05-18

**Summary Of Contributions:**

This manuscript studies the problem of selecting the number of communities in stochastic block models using cross-validation. The authors propose an inductive node-splitting procedure (INCV), which uses $A^{(11)}$ for clustering training nodes, $A^{(12)}$ for assigning held-out nodes to the learned communities, and $A^{(22)}$ for validation. The theoretical properties have been carefully established.

**Audience:**

Yes

**Broader Impact Concerns:**

None.

**Claims And Evidence:**

Yes

**Requested Changes:**

# Major Comments

1. **Balanced estimated labels and post-processing.**

   Assumption 1 requires not only the true community structure but also the estimated labels $\hat\phi_K$ to be balanced for all candidate values of $K$. The manuscript proposes post-processing steps such as merging or trimming small clusters and reassigning test nodes from the largest estimated community to the smallest one.

   The authors assume that the estimated labels $\hat{\phi}_K$ are balanced. To enforce this when $K > K^*$, the authors propose an ad-hoc post-processing step that forcibly reclassifies nodes from the largest community to the smallest one. From a statistical perspective, this forced "peak-shaving" may be risky. It alters the natural clustering outcome and may distort the likelihood evaluation on the validation block $A^{(22)}$.
   Could the authors formally justify how this artificial intervention bounds the perturbation on the final validation loss. Or at least, in the simulation section, the authors could consider to report whether this post-processing was actually triggered and compare the empirical performance of INCV with and without this step.

2. **Separation condition on the block probability matrix.**

   In the general SBM case, the theory uses Assumption 4. This assumption requires that, for each true community $v$, the $v$-th column of the block probability matrix $P$ has a unique largest entry:

   $$\zeta(v)=\arg\max_{u\in[K^*]} p_{uv}$$

   Theorem 2 then uses the gap:

   $$\beta_n = \min_{v\in[K^*]} \left( p_{\zeta(v)v} - \max_{u\ne \zeta(v)}p_{uv} \right)$$

   This seems to be a slightly strict separation condition on the block probability matrix. The authors may consider clarifying whether Assumption 4 is needed for the INCV idea itself, or mainly for the current one-vs-all classification step. Could the condition be weakened? Compared with the classical identifiability condition requiring only distinct rows of the block matrix (Chen and Lei, 2018), Assumption 4 seems considerably stronger. The authors may consider adding a short discussion on whether such weaker conditions are possible, and whether the strict separation in Assumption 4 is a technical condition for the proof or a necessary condition for the current INCV procedure.

   There is also a typo in Corollary 2. It states that

   $$\Pr(\widehat K_{\mathrm{INCV}} < K^*) \to 1$$

   which contradicts Theorem 2 and the claimed half-consistency result. This should presumably be

   $$\Pr(\widehat K_{\mathrm{INCV}} < K^*) \to 0$$

   This should be corrected.

3. **Clarification of the Mechanism Behind the Consistency Improvement of INCV.**
   I would encourage the authors to provide a more explicit explanation, either theoretically or empirically, of why INCV is able to distinguish overfitted community configurations more effectively than existing methods. At present, the manuscript explains in detail why conventional CV fails, whereas the corresponding explanation of why INCV succeeds is comparatively less transparent. Although the intuition may be relatively natural for researchers working on network analysis, a more explicit explanation would likely help general readers better understand the main advantage of the proposed framework.

4. **Further extensions to DCSBM.**
   The authors may consider discussing how the proposed method could be extended to the degree-corrected stochastic block model (DCSBM). In addition, adding simulation experiments under DCSBM settings would help demonstrate the performance of the proposed approach in the presence of degree heterogeneity.

5. **More Simulation Studies.**
   In the experimental comparisons, the authors may consider to include likelihood-based methods (Wang and Bickel, 2017; Hu et al., 2020), in addition to existing network cross-validation approaches.

6. **Quantitative Evaluation Metrics in Applications.**
   The authors may include additional evaluation metrics that directly reflect the quality of the selected number of communities. For example, measures such as modularity, Ncut, and other clustering performance metrics could be reported to provide a more rigorous justification for the chosen community structure.

# Minor Comments

1. Community labels are identifiable only up to permutation. Assumption 3 could be written using an optimal label permutation, or using a permutation-invariant misclassification metric.

2. The notation alternates between $V$-fold and $F$-fold cross-validation. Please use one notation consistently.

3. Several typographical errors should be corrected, including "necessarity," "Firsty," "calculatedon," and inconsistent spellings such as "Forley/Fowler."

# References

- Chen, K. and Lei, J. (2018), "Network cross-validation for determining the number of communities in network data," *Journal of the American Statistical Association*, 113, 241–251.

- Hu, J., Qin, H., Yan, T., and Zhao, Y. (2020), "Corrected Bayesian information criterion for stochastic block models," *Journal of the American Statistical Association*, 115, 1771–1783.

- Wang, Y. R. and Bickel, P. J. (2017), "Likelihood-based model selection for stochastic block models," *The Annals of Statistics*, 45, 500–528.

**Strengths And Weaknesses:**

**Strengths.**
The paper studies an important problem on selecting the number of communities in stochastic block models using cross-validation. The proposed INCV framework is intuitive and statistically appealing, and the separation of clustering, induction, and validation through $(A^{(11)},A^{(12)},A^{(22)})$ is elegant. A major strength of the paper is the theoretical result showing that pure cross-validation, without additional penalization, can achieve full consistency under the affiliation SBM setting.  Overall, both the theoretical analysis and the numerical results suggest that the proposed method is promising and practically useful.

**Weaknesses.**
Some assumptions used in the theoretical analysis appear relatively strong, especially the balanced post-processing assumption and the separation condition in Assumption 4. In addition, while the manuscript explains clearly why conventional CV methods may fail, the mechanism explaining why INCV improves over existing methods is currently less transparent. The current theory and experiments also mainly focus on SBM settings, and it would be helpful to discuss more realistic scenarios such as degree heterogeneity or DCSBM settings.

---

> ### Author Response · Authors · 2026-06-02
> **Response to Reviewer zTE9, Part 1**
>
> The authors would like to thank the reviewers for their constructive comments and suggestions that have helped further improve the quality of this manuscript substantially. Please see below our detailed responses.
>
> >**Comment 1**: Assumption 1 requires not only the true community structure but also the estimated labels $\hat{\phi}_K$ to be balanced for all candidate values of $K$. The manuscript proposes post-processing steps such as merging or trimming small clusters and reassigning test nodes from the largest estimated community to the smallest one.
>
> >The authors assume that the estimated labels $\hat{\phi}_K$ are balanced. To enforce this when $K>K^*$, the authors propose an ad-hoc post-processing step that forcibly reclassifies nodes from the largest community to the smallest one. From a statistical perspective, this forced "peak-shaving" may be risky. It alters the natural clustering outcome and may distort the likelihood evaluation on the validation block $A^{(22)}$. Could the authors formally justify how this artificial intervention bounds the perturbation on the final validation loss. Or at least, in the simulation section, the authors could consider to report whether this post-processing was actually triggered and compare the empirical performance of INCV with and without this step.
>
> **Reply**: We thank the reviewer for raising this important issue. We now realized that the original wording could give the impression that the balancing post-processing step is part of the empirical INCV implementation, and that such a forced “peak-shaving” operation could be statistically questionable if it were actually applied to the validation procedure.
>
> We have revised the discussion following Assumption 1 and Remark 1 to clarify this point. The post-processing step is not used in our numerical experiments or real-data analyses. Instead, it is meant as a theoretical device for excluding degenerate estimated partitions in the proof. In the affiliation SBM analysis, when $K>K^*$, the proof needs to rule out cases where almost all test nodes are assigned to few estimated communities. The balanced-label condition provides a sensible way to ensure that the relevant within- and between-type validation edge sets have sufficiently large cardinalities, which is essential for controlling the validation loss under overfitting. We now explicitly state in Remark 1 that all empirical results are based on the natural outputs of the clustering and classification steps, without forced reclassification. Therefore, no perturbation of the validation loss is introduced by such post-processing in our reported experiments. Moreover, for the general SBM case, since we only consider the underfitting case, such balance requirement is not required.
>
> We also conducted an additional numerical analysis in Supplementary Material Section B.2 to diagnose that whether the natural output of INCV procedure satisfy the balanced condition. Specifically, for each candidate $K$, we compute the empirical balance ratio $$\widehat r_K = \frac{K\min_{1\le k\le K}|\hat{\mathcal N}_k|} {|\hat{\mathcal N}|},$$ where $\hat{\mathcal N}_k$ denotes the $k$-th estimated community under the candidate value $K>K^*$. For each replication, we summarize this ratio over all folds and all overfitted candidate values by its mean and 5\% quantile, and then report the averages of these summaries over 100 replications, both on the evaluation set and on the full estimated label vector. The results show that the empirical balance ratios remain bounded away from zero in the reported settings. This indicates that severe estimated-label imbalance is not a systematic issue in our simulations, and that the forced balancing step is not needed in practice.

---

> ### Author Response · Authors · 2026-06-02
> **Response to Reviewer zTE9, Part 2**
>
> >**Comment 2**: In the general SBM case, the theory uses Assumption 4. This assumption requires that, for each true community $v$, the
> $v$-th column of the block probability matrix $P$ has a unique largest entry:
> $$\zeta(v) := \arg\max_{u \in [K^{\ast}]} p_{uv}.$$
> Theorem 2 then uses the gap:
> $$\beta_n
> 	:= \min_{v \in [K^{\ast}]}
> 	\left(
> 	p_{\zeta(v)v} - \max_{u \neq \zeta(v)} p_{uv}
> 	\right).$$
> This seems to be a slightly strict separation condition on the block probability matrix. The authors may consider clarifying whether Assumption 4 is needed for the INCV idea itself, or mainly for the current one-vs-all classification step. Could the condition be weakened? Compared with the classical identifiability condition requiring only distinct rows of the block matrix (Chen and Lei, 2018), Assumption 4 seems considerably stronger. The authors may consider adding a short discussion on whether such weaker conditions are possible, and whether the strict separation in Assumption 4 is a technical condition for the proof or a necessary condition for the current INCV procedure. There is also a typo in Corollary 2. It states that
> $$
> \Pr(\hat{K}<K^{\ast})\to 1,
> $$
> which contradicts Theorem 2 and the claimed half-consistency result. This should presumably be
> $$
> \Pr(\hat{K}_{\mathrm{INCV}}<K^{\ast})\to 0.
> $$
> This should be corrected.
>
> **Reply**: We thank the reviewer for this insightful comment. We agree that Assumption 4 is stronger than the classical identifiability condition requiring distinct rows of the block probability matrix. We have revised the discussion in Remark 5 after Assumption 4 in the manuscript to clarify that this assumption is mainly imposed for the current one-vs-all induction step analyzed in the theory, rather than being a necessary condition for the INCV framework itself.
>
> The reason for this condition is as follows. For the one-vs-all induction rule, Assumption 4 has a direct population-level interpretation: conditional on accurate training labels, a test node from community $v$ is assigned according to the largest entry in the $v$-th column of $P$. Uniqueness makes this assignment well-defined, while the bijectivity of $v\mapsto\zeta(v)$ prevents distinct true communities from being merged. Without this condition, the induced labels on $\mathcal N^{(2)}$ may be systematically mixed, so the condition is difficult to substantially weaken for this simple classifier. Thus, Assumption 4 should be understood as a technical condition for this particular induction rule.
>
> At the same time, this condition does not imply an inherent limitation of the INCV framework. To illustrate this point, we added a new experiment in the Supplementary Material, Section B.4. We consider a three-community SBM with
> $$
> B=\rho B_0,\qquad
> B_0=
> \begin{pmatrix}
> 1 & 1/2 & 1\\\\
> 1/2 & 1 & 1/3\\\\
> 1 & 1/3 & 2/3
> \end{pmatrix}.
> $$
> This block matrix has distinct rows, so it satisfies the usual SBM identifiability condition, but it violates the column-wise separation condition in Assumption 4. We compare the affinity-based one-vs-all induction rule used in the main theoretical analysis with an alternative likelihood-based induction rule. The results show that the affinity-based rule indeed performs poorly in this setting, whereas the likelihood-based induction rule substantially improves the selection accuracy when the signal is moderate or strong. This supports our interpretation that Assumption 4 is tied to the simple induction rule used in the proof, and that weaker identifiability conditions may be possible with more informative induction procedures.
>
> We also corrected the typo in Corollary 2: the statement now reads
> $$
> \Pr(\widehat K_{\mathrm{INCV}}<K^{\ast})\to 0,
> $$
> which is consistent with Theorem 2 and the half-consistency result.

---

> ### Author Response · Authors · 2026-06-02
> **Response to Reviewer zTE9, Part 3**
>
> >**Comment 3**: I would encourage the authors to provide a more explicit explanation, either theoretically or empirically, of why INCV is able to distinguish overfitted community configurations more effectively than existing methods. At present, the manuscript explains in detail why conventional CV fails, whereas the corresponding explanation of why INCV succeeds is comparatively less transparent. Although the intuition may be relatively natural for researchers working on network analysis, a more explicit explanation would likely help general readers better understand the main advantage of the proposed framework.
>
> **Reply**: We thank the reviewer for this helpful suggestion. We have added an explicit explanation after Remark 2 on page 13 and further clarified the mechanism in Remark 8 and the Discussion. The key point is that INCV does not eliminate overfitting in fully general SBMs by itself. Rather, under the affiliation structure, overfitting by splitting a true community necessarily turns many truly within-community validation edges into estimated between-community edges. Because the affiliation model pools all within-community probabilities and all between-community probabilities, this creates a detectable validation loss gap. In unrestricted general SBMs, such a gap may vanish because the split blocks can each be assigned their own probability parameters. We hope this is helpful for a general reader.
>
> >**Comment 4**: The authors may consider discussing how the proposed method could be extended to the degree-corrected stochastic block model (DCSBM). In addition, adding simulation experiments under DCSBM settings would help demonstrate the performance of the proposed approach in the presence of degree heterogeneity.
>
> **Reply**: We thank the reviewer for this thoughtful comment. We agree that degree heterogeneity is an important feature of many real networks, and that it is useful to discuss how the proposed INCV framework may be extended beyond the standard SBM.
>
> In the revised manuscript, we added a new subsection in the Supplementary Material, Section B.5, entitled “A Preliminary Extension to Degree-Corrected Stochastic Block Models.” In this section, we propose an ad-hoc degree-adjusted implementation of INCV for DCSBMs. Specifically, under the DCSBM model
> $$
> \mathbb P(A_{ij}=1)=\theta_i\theta_j p_{\phi(i)\phi(j)},
> $$
> the clustering step is modified using a degree-normalized spectral clustering procedure on $A^{(11)}$. The classification step is also adjusted by normalizing the affinity of each test node to the estimated training communities using the estimated degree parameters. Finally, the validation loss on $A^{(22)}$ is computed using the predicted probabilities
> $$
> \widehat \theta_i \widehat p_{\hat\phi(i)\hat\phi(j)} \widehat \theta_j,
> $$
> where $\widehat{P}$ is estimated from $A^{(11)}$ using the clustering output on the training subgraph. Thus, the extension keeps the same INCV structure, that is, clustering on $A^{(11)}$, induction using $A^{(12)}$, and validation on $A^{(22)}$, while modifying each step to account for node-level degree heterogeneity.
>
> We also added simulation experiments under DCSBM settings in the same supplementary section. We compare the proposed degree-adjusted INCV implementation with NCV, ECV, NETCROP, and LRBIC. The results show that the DCSBM version of INCV is competitive with existing cross-validation-based methods in many moderate and strong signal regimes. Overall, these experiments suggest that the node-splitting and induction idea has potential beyond the standard SBM setting, although a rigorous theoretical treatment for DCSBMs is left for future work.
>
> >**Comment 5**: In the experimental comparisons, the authors may consider to include likelihood-based methods (Hu et al., 2020; Wang and Bickel, 2017), in addition to existing network cross-validation approaches.
>
> **Reply**: We thank the reviewer for this suggestion. Following this comment, we have added likelihood/BIC-type model selection methods to the experimental comparisons. Specifically, in the affiliation SBM simulation, we now compare INCV not only with existing network cross-validation methods, including NCV, ECV, and NETCROP, but also with two BIC-type likelihood criteria: LRBIC (Wang and Bickel, 2017) and CLBIC (Saldana et al., 2017). The updated results are reported in Table 1, together with the corresponding discussion in the simulation section.
>
> We were not able to locate a readily usable public implementation of the corrected BIC method in Hu et al. (2020). Therefore, we included CLBIC (Saldana et al., 2017) as an additional BIC-type benchmark in the affiliation SBM experiment. Since CLBIC is substantially more computationally demanding and does not change the qualitative conclusions in the affiliation SBM experiment, we use LRBIC as the representative BIC-type likelihood method in the additional supplementary experiments under general SBMs and DCSBMs.

---

> ### Author Response · Authors · 2026-06-02
> **Response to Reviewer zTE9, Part 4**
>
> >**Comment 6**: The authors may include additional evaluation metrics that directly reflect the quality of the selected number of communities. For example, measures such as modularity, Ncut, and other clustering performance metrics could be reported to provide a more rigorous justification for the chosen community structure.
>
> **Reply**: We thank the reviewer for this helpful suggestion. Following this comment, we have added additional clustering quality metrics for the two real-data applications. Specifically, for each candidate community number $K$, we apply the same spectral clustering method to the full adjacency matrix and compute the corresponding modularity and normalized cut. Since the standard multiway Ncut is a sum over all communities and therefore tends to increase mechanically with $K$, we report the averaged normalized cut, defined as
> $$
> \frac{1}{K}
> \sum_{k=1}^K
> \frac{\mathrm{cut}(\widehat{\mathcal N}_k,\widehat{\mathcal N}_k^c)}
> {\mathrm{vol}(\widehat{\mathcal N}_k)}.
> $$
> This quantity can be interpreted as the average normalized cut contribution per estimated community and is more comparable across different candidate values of $K$. The additional results are reported in Table 2, immediately after the real-data network visualizations, together with corresponding discussion. These additional diagnostics provide complementary support for the community structures selected by INCV.
>
> >**Comment 7**: 1. Community labels are identifiable only up to permutation. Assumption 3 could be written using an optimal label permutation, or using a permutation-invariant misclassification metric.
> 2. The notation alternates between $V$-fold and $F$-fold cross-validation. Please use one notation consistently.
> 3. Several typographical errors should be corrected, including "necessarity," "Firsty," "calculatedon," and inconsistent spellings such as "Forley/Fowler."
>
> **Reply**: Revised as suggested. Thank you.

---

### Review · Reviewer_Ujiw · 2026-05-26

**Summary Of Contributions:**

The authors propose INCV (Inductive Node-split Cross-Validation), a three-step node-splitting cross-validation procedure for selecting the number of communities $K$ in stochastic block models (SBMs). The procedure clusters the training subgraph $A^{(11)}$, inductively classifies test nodes via a one-vs-all rule based on $A^{(12)}$, and validates on the held-out subgraph $A^{(22)}$.

**Audience:**

Yes

**Broader Impact Concerns:**

Not applied

**Claims And Evidence:**

Yes

**Requested Changes:**

See my questions above

**Strengths And Weaknesses:**

The motivation is clear, and  the proposed method is novel.



However, the current manuscript needs substantial revision. I will suggest major revision.

1. Table 1 contains an impossible setting. With n=500 and K*=6, nmin=90 cannot be the smallest community size because 6*90 = 540 > 500.
2. Simulations focus on affiliation SBM. That is understandable given Theorem 1, but the manuscript also discusses general SBMs and robustness. There are no simulations for general non-affiliation SBMs
3.  The algorithm should use arg min for MSE.
4. The paper cites PNN-CV as a method achieving full consistency, but does not compare against it.
5. Assumption 1 requires that both the true labels $\phi^{\ast}$ and the recovered labels $\hat{\phi}_K$ be balanced with respect to a constant $\pi_0$. The second half is unusual: for $K > K^{\ast}$, requiring the over-fitted clustering to spread mass evenly across $K$ groups is not a property of the data but an artifact imposed by the clustering method.
6. The authors should also report the computational cost when comparing different methods. Assumption~3 requires exact recovery on the training subgraph, $\Pr(\hat{\phi}_{K^\ast}\neq \phi^\ast) \le g(n\tau), g(x)\to 0.$ This is a strong requirement and depends heavily on the clustering algorithm, the sparsity level, the signal gap, and the community balance. The manuscript states that likelihood-based methods and spectral clustering with exact $k$-means satisfy this assumption under mild conditions, but it does not spell out those conditions or verify that they are compatible with the assumptions in Theorem 1 and Corollary 1. This is particularly important because many spectral-clustering results provide misclassification consistency rather than exact recovery unless stronger signal-to-noise conditions hold. The authors should state concrete sufficient conditions under which Assumption 3 holds for the clustering method used in the simulations and explain how these conditions combine with the theorem's requirements on $\tau$, $p-q$, $\delta_n$, and $\rho_n$.
7. A small figure showing $ A = \begin{pmatrix} A^{(11)} & A^{(12)} \\ A^{(21)} & A^{(22)} \end{pmatrix} $
with the three roles (Clustering, Classification, Validation) labeled would clarify the conceptual core of the procedure.

---

> ### Author Response · Authors · 2026-06-02
> **Response to Reviewer Ujiw, Part 1**
>
> The authors would like to thank the reviewers for their constructive comments and suggestions that have helped further improve the quality of this manuscript substantially. Please see below our detailed responses.
> >**Comment 1**: Table 1 contains an impossible setting. With $n=500$ and $K^*=6$, $n_{\min}=90$ cannot be the smallest community size because $6^*90 = 540 > 500$.
>
> **Reply**: We thank the reviewer for catching the error. The largest value of $n_{\min}$ should be 80 and we have corrected it in Table 1.
>
> >**Comment 2**: Simulations focus on affiliation SBM. That is understandable given Theorem 1, but the manuscript also discusses general SBMs and robustness. There are no simulations for general non-affiliation SBMs.
>
> **Reply**: We thank the reviewer for raising this point. We agree that, since the manuscript discusses general SBMs and robustness beyond the affiliation SBM, it is important to include simulations under non-affiliation SBM settings. In the revised manuscript, we added new simulations under structured general SBMs in the Supplementary Material, Section B.3.
>
> We also clarified an important addition for an alternative validation implementation. The main theoretical analysis for general SBMs focuses on the validation-MLE version, where the block probabilities are re-estimated on $A^{(22)}$. Theorem 2 shows that this version avoids underfitting, but our numerical experiments confirm that it may still over-select in general SBMs. This can be partly explained by the fact that, when each estimated block pair is assigned its own probability parameter, re-estimating these probabilities on $A^{(22)}$ can fit random fluctuations in the validation block.
> Motivated by this observation and following the spirit of existing network cross-validation methods such as Chen and Lei (2018) and Li et al. (2020), we introduced a practical plug-in alternative in Section 2.2 under the paragraph ``Alternative Probability Estimation''. In this version, the block probability matrix is estimated from $A^{(11)}$ and then plugged into the validation loss on $A^{(22)}$. The supplementary simulations compare this plug-in version of INCV with NCV, ECV, NETCROP, and LRBIC. The results show that the plug-in INCV implementation is broadly comparable to existing CV-based methods in many moderate and strong signal regimes.
>
> >**Comment 3**: The algorithm should use arg min for MSE.
>
> **Reply**: We thank the reviewer for this sharp observation. For consistency of notation, we have rewritten the likelihood criterion as a negative log-likelihood loss, so that the final selection rule uses $\arg\min$ for both the likelihood and MSE objectives. We have corrected this in the Validation paragraph in Section 2.2 and in Algorithm 1.
>
> >**Comment 4**: The paper cites PNN-CV as a method achieving full consistency, but does not compare against it.
>
> **Reply**: We thank the reviewer for this suggestion. PNN-CV is indeed an important related method, and we have cited it in the manuscript because it provides a full-consistency result by adding an explicit penalty to the cross-validation criterion.
>
> However, we did not include PNN-CV in the numerical comparison for two reasons. First, to the best of our knowledge, a publicly available implementation of PNN-CV is not currently available. Second, PNN-CV is conceptually different from the main focus of this paper: it achieves full consistency by adding an explicit complexity penalty, whereas our main theoretical question is whether and when cross-validation alone can guarantee full consistency. To address the reviewer’s broader concern about comparisons with penalized model selection procedures, we have added likelihood/BIC-type benchmarks, including LRBIC and CLBIC, to the affiliation SBM simulation. These methods also incorporate explicit complexity adjustment and therefore provide relevant non-CV benchmarks. We hope you agree this is a sensible solution.

---

> ### Author Response · Authors · 2026-06-02
> **Response to Reviewer Ujiw, Part 2**
>
> >**Comment 5**: Assumption 1 requires that both the true labels $\phi^{\ast}$ and the recovered labels $\hat{\phi}_K$ be balanced with respect to a constant $\pi_0$. The second half is unusual: for $K>K^*$, requiring the over-fitted clustering to spread mass evenly across $K$ groups is not a property of the data but an artifact imposed by the clustering method.
>
> **Reply**: We thank the reviewer for this helpful comment. We agree that the balancedness of the recovered labels $\hat\phi_K$, especially when $K>K^*$, should not be interpreted as an intrinsic property of the data-generating network. Rather, it is a technical regularity condition on the candidate partitions used in the theoretical proof. The original manuscript did not make this distinction sufficiently clear, and we have revised the discussion following Assumption 1 accordingly.
>
> In the affiliation SBM analysis, when $K>K^*$, the proof needs to rule out cases where almost all test nodes are assigned to few estimated communities. The balanced-label condition provides a sensible way to ensure that the relevant within- and between-type validation edge sets have sufficiently large cardinalities, which is essential for controlling the validation loss under overfitting. We now explicitly state in Remark 1 that all empirical results are based on the natural outputs of the clustering and classification steps, without forced reclassification. Therefore, no perturbation of the validation loss is introduced by such post-processing in our reported experiments. Moreover, for the general SBM case, since we only consider the underfitting case, such balance requirement is not required.
>
> We also conducted an additional numerical analysis in Supplementary Material Section B.2 to diagnose that whether the natural output of INCV procedure satisfy the balanced condition. Specifically, for each candidate $K$, we compute the empirical balance ratio $$\widehat r_K = \frac{K\min_{1\le k\le K}|\hat{\mathcal N}_k|} {|\hat{\mathcal N}|},$$ where $\hat{\mathcal N}_k$ denotes the $k$-th estimated community under the candidate value $K>K^*$. For each replication, we summarize this ratio over all folds and all overfitted candidate values by its mean and 5\% quantile, and then report the averages of these summaries over 100 replications, both on the evaluation set and on the full estimated label vector. The results show that the empirical balance ratios remain bounded away from zero in the reported settings. This indicates that severe estimated-label imbalance is not a systematic issue in our simulations, and that the forced balancing step is not needed in practice.
>
> >**Comment 7**: A small figure showing $A=(A^{(11)},A^{(12)};A^{(21)},A^{(22)})$ with the three roles (Clustering, Classification, Validation) labeled would clarify the conceptual core of the procedure.
>
> **Reply**: Thank you for this nice suggestion. We add a small figure (Figure 1) in Section 2.2 before the Clustering paragraph to clarify the roles more explicitly.

---

> ### Author Response · Authors · 2026-06-02
> **Response to Reviewer Ujiw, Part 3**
>
> >**Comment 6**: The authors should also report the computational cost when comparing different methods. Assumption~3 requires exact recovery on the training subgraph, $\Pr\left( \hat{\phi}_{K^{\ast}} \ne \phi^{\ast} \right) \le g(n\tau),g(x)\to 0$. This is a strong requirement and depends heavily on the clustering algorithm, the sparsity level, the signal gap, and the community balance. The manuscript states that likelihood-based methods and spectral clustering with exact $k$-means satisfy this assumption under mild conditions, but it does not spell out those conditions or verify that they are compatible with the assumptions in Theorem 1 and Corollary 1. This is particularly important because many spectral-clustering results provide misclassification consistency rather than exact recovery unless stronger signal-to-noise conditions hold. The authors should state concrete sufficient conditions under which Assumption 3 holds for the clustering method used in the simulations and explain how these conditions combine with the theorem's requirements on $\tau$, $p-q$, $\delta_n$, and $\rho_n$.
>
> **Reply**: We thank the reviewer for this important comment. We agree that Assumption 3 is a strong clustering requirement and that its compatibility with the signal and sparsity conditions in Theorem 1 and Corollary 1 should be made explicit.
>
> In the revised manuscript, we added a detailed discussion after Corollary 1. For spectral clustering with exact $k$-means, we state concrete sufficient conditions under which Assumption 3 holds. Let $m=|\mathcal N^{(1)}|\asymp n\tau$ and define
> $$
> \Delta_{n,\tau}=\frac{m(p-q)^2}{p}.
> $$
> Using row-wise eigenvector perturbation results together with the nonsplitting property argument (Cape et al., 2019; Jin et al., 2023), Assumption 3 is satisfied when
> $$
> n\tau p \gg \log n,\qquad \Delta_{n,\tau}\gg \log n.
> $$
> These conditions are compatible with Theorem 1. Moreover, under the sparse scaling $p=\rho_n p_0$, $q=\rho_n q_0$ with fixed $p_0>q_0>0$, the above conditions reduce to $n\tau\rho_n\gg \log n$, which is exactly the order required by condition (i) in Corollary 1. We also explain that likelihood-based training methods have compatible requirements; for example, exact consistency of likelihood modularity requires the expected degree parameter to dominate $\log n$ (Bickel and Chen, 2009), which again corresponds to $n\tau\rho_n\gg \log n$ on the training subgraph.
>
> We have also clarified the distinction between the idealized clustering step used in the theory and the practical implementation used in the simulations. Since exact $k$-means is computationally intractable in general, our numerical experiments use a multi-start approximate $k$-means algorithm. This is consistent with common practice in spectral-clustering-based network analysis, where theoretical analyses often formulate guarantees under exact $k$-means while implementations use practical approximate solvers (Jin et al., 2023).
>
> Finally, following your suggestion, we added a computational cost comparison in the Supplementary Material, Section B.1. We report the running times of INCV and the competing methods under representative affiliation SBM settings. The results show that INCV is slightly slower than some existing CV methods because it uses 10-fold node splitting and computes both likelihood and MSE validation objectives, but its computational cost remains moderate.

---

> > ### Comment · Reviewer_Ujiw · 2026-06-15
> > **Comments**
> >
> > I thank the authors for a careful and responsive revision. I have three  additional questions.
> >
> > 1.  The paper's self-described ``key novelty,'' the node-split-plus-induction step, is the same as the \emph{predictive assignment} procedure of Bhadra, Pensky, and Sengupta (Scalable community detection in massive networks via predictive assignment), which is neither cited nor discussed.
> > 2. The introduction's survey of cross-validation methods for networks would be more complete by also discussing Cheng, Chen, Ma, and Zhong (2026), ``Graphon Cross-Validation: Assessing Models on Network Data'' (ICLR 2026), a recent cross-validation procedure for network and graphon models.
> > 3. In the DCSBM experiment, the proposed INCV does not beat ECV, please explain why and add discussions of limitations.

---

> > > ### Author Response · Authors · 2026-06-16
> > > **Response to Reviewer Ujiw, Additional Comments**
> > >
> > > The authors would like to thank the reviewer for your additional constructive comments and suggestions that have helped further improve the quality of this manuscript. Please see below our detailed responses.
> > > > **Comment 1**: The paper's self-described ``key novelty,'' the node-split-plus-induction step, is the same as the predictive assignment procedure of Bhadra, Pensky, and Sengupta (Scalable community detection in massive networks via predictive assignment), which is neither cited nor discussed.
> > >
> > > **Reply**: We thank the reviewer for pointing this out. We were previously unaware of the closely related work by Bhadra, Pensky, and Sengupta (2026), and we appreciate the reviewer bringing it to our attention. We agree that the induction step in INCV is closely related to their predictive assignment idea. We have now added the reference and included a discussion of the connection and differences in the Introduction and in Section 2.2.
> > >
> > > The main distinction is the statistical goal. Predictive assignment is designed for scalable community detection when the number of communities $K$ is given. It reduces computation by performing community detection on a smaller subgraph and then assigning the remaining nodes using model-based structural information. By contrast, our goal is model selection: $K$ is unknown, and INCV repeats the clustering--induction--validation procedure over candidate values of $K$ and selects $K$ by cross-validation loss.
> > >
> > > We also note that the concrete induction rules are different. Predictive assignment constructs estimated structural-link quantities or prototypes from the training subgraph and assigns each remaining node by comparing its observed vector with these targets. In INCV, the main implementation uses a direct affinity-based induction rule, and the supplement also considers a likelihood-based induction rule. These differences in the induction rule are secondary to our main distinction: INCV uses induction as an intermediate step in a cross-validation procedure for selecting unknown K, with the induced test subgraph reserved for validation.
> > >
> > > > **Comment 2**: The introduction's survey of cross-validation methods for networks would be more complete by also discussing Cheng, Chen, Ma, and Zhong (2026), ``Graphon Cross-Validation: Assessing Models on Network Data'' (ICLR 2026), a recent cross-validation procedure for network and graphon models.
> > >
> > > **Reply**: We thank the reviewer for pointing out this recent related work. We have added Cheng, Chen, Ma, and Zhong (2026) to the introduction when discussing cross-validation methods for network data.
> > >
> > > > **Comment 3**: In the DCSBM experiment, the proposed INCV does not beat ECV, please explain why and add discussions of limitations.
> > >
> > > **Reply**: We thank the reviewer for this comment. We agree that the DCSBM experiment should not be interpreted as showing that INCV uniformly outperforms ECV. We have revised the discussion after Table B.5 to make this limitation explicit.
> > >
> > > At the same time, the table suggests that INCV is not uniformly dominated by ECV. The proposed DCSBM extension achieves high or perfect accuracy in most moderate- and strong-signal settings, and it is comparable to ECV in most cases considered in this simulation. For example, when $K^*=3$, $\rho=0.1$, and $n_{\min}=120$, INCV slightly outperforms ECV. The main underperformance occurs in a few difficult regimes, especially when the smallest community is small or the degree heterogeneity makes the induction step unstable.
> > >
> > > This behavior is understandable because the DCSBM experiment is only a preliminary plug-in extension beyond the affiliation SBM, which is the main setting covered by our full-consistency theory. Under DCSBM, node-specific degree parameters introduce additional variability into both the induction and validation steps. After node splitting, small communities may have limited information in the training block, making degree-adjusted classification less stable. In contrast, ECV is based on edge sampling and retains all nodes in the fitting and validation procedure, which can be advantageous in degree-heterogeneous and imbalanced regimes. A more systematic study of INCV under DCSBMs is left for future work.

---

### Decision · Action_Editor_wBoL · 2026-06-24

**Recommendation:** Accept as is

**Audience:**

Yes

**Claims And Evidence:**

Yes